# STRUCTURED COVARIANCE ESTIMATION VIA TENSOR-TRAIN DECOMPOSITION

## ABSTRACT

We consider a problem of covariance estimation from a sample of i.i.d. high-dimensional random vectors. To avoid the curse of dimensionality, we impose an additional assumption on the structure of the covariance matrix $\Sigma$. To be more precise, we study the case when $\Sigma$ can be approximated by a sum of double Kronecker products of smaller matrices in a tensor train (TT) format. Our setup naturally extends widely known Kronecker sum and CANDECOMP/PARAFAC models but admits richer interaction across modes. We suggest an iterative polynomial time algorithm based on TT-SVD and higher-order orthogonal iteration (HOOI) adapted to Tucker-2 hybrid structure. We derive non-asymptotic dimension-free bounds on the accuracy of covariance estimation taking into account hidden Kronecker product and tensor train structures. The efficiency of our approach is illustrated with numerical experiments.

## 1 INTRODUCTION

Given $\mathbf{X}, \mathbf{X}_1, \ldots, \mathbf{X}_n \in \mathbb{R}^d$ i.i.d. centered random vectors, we are interested in estimation of their covariance matrix $\Sigma = \mathbb{E}\mathbf{X}\mathbf{X}^\top \in \mathbb{R}^{d \times d}$. Despite its long history, this classical problem still gets considerable attention of statistical and machine learning communities. The reason is that in modern data mining tasks researchers often have to deal with high-dimensional observations. In such scenarios they cannot rely on classical estimates, for instance, sample covariance

$$\widehat{\Sigma} = \frac{1}{n} \sum_{i=1}^{n} \mathbf{X}_i \mathbf{X}_i^\top,$$

suffering from the curse of dimensionality. To overcome this issue, statisticians impose additional assumptions on $\Sigma$ in order to exploit the data structure and reduce the total number of unknown parameters. Some recent methodological and theoretical advances in covariance estimation are related with Kronecker product models, which are particularly useful for analysis of multiway or tensor-valued data (Werner et al., 2008; Allen and Tibshirani, 2010; Greenewald et al., 2013; Sun et al., 2018; Guggenberger et al., 2023). For example, motivated by multiple input multiple output (MIMO) wireless communications channels, Werner, Jansson, and Stoica (2008) assumed that $\Sigma$ can be represented as a Kronecker product of two smaller matrices $\Phi \in \mathbb{R}^{p \times p}$ and $\Psi \in \mathbb{R}^{q \times q}$, such that $pq = d$:

$$\Sigma = \Phi \otimes \Psi = \begin{pmatrix} \varphi_{11}\Psi & \ldots & \varphi_{1p}\Psi \\ \vdots & \ddots & \vdots \\ \varphi_{p1}\Psi & \ldots & \varphi_{pp}\Psi \end{pmatrix}. \tag{1}$$

It is known that (see, for instance, the proof of Theorem 1 in (Van Loan and Pitsianis, 1993)) $\Sigma$ of form (1) can be reshaped into a rank-one matrix using an isometric rearrangement (or permutation) operator $\mathcal{P} : \mathbb{R}^{pq \times pq} \to \mathbb{R}^{p^2 \times q^2}$ (see (Puchkin and Rakhuba, 2024, Definition 2.1)). Based on this fact, Werner, Jansson, and Stoica suggested to estimate $\mathcal{P}(\Sigma)$ applying singular value decomposition to $\mathcal{P}(\widehat{\Sigma})$ and showed that this estimate is asymptotically efficient in the Gaussian case. They called this approach covariance matching. This idea was further developed by (Tsiligkaridis and Hero, 2013; Masak et al., 2022; Puchkin and Rakhuba, 2024), who considered the sum of Kronecker products model

$$\Sigma = \sum_{k=1}^{K} \Phi_k \otimes \Psi_k, \tag{2}$$

where $\Phi_1, \Psi_1, \ldots, \Phi_K, \Psi_K$ are symmetric positive semidefinite matrices, such that $\Phi_j \in \mathbb{R}^{p \times p}$, $\Psi_j \in \mathbb{R}^{q \times q}$ for all $j \in \{1, \ldots, K\}$ and $pq = d$. They studied properties of the permuted regularized least squares (PRLS) estimates. In (Tsiligkaridis and Hero, 2013; Puchkin and Rakhuba, 2024), the authors regularized the loss function using the nuclear norm

$$\widehat{\Sigma}^{\circ} = \mathcal{P}^{-1}(\widetilde{R}), \quad \text{where} \quad \widetilde{R} \in \operatorname*{argmin}_{R \in \mathbb{R}^{p^2 \times q^2}} \left\{ \left\| R - \mathcal{P}(\widehat{\Sigma}) \right\|_{\mathrm{F}}^2 + \lambda \|R\|_* \right\}, \tag{3}$$

while Masak et al. (2022) considered a rank-penalized estimate

$$\check{\Sigma} = \mathcal{P}^{-1}(\check{R}), \quad \check{R} \in \operatorname*{argmin}_{R \in \mathbb{R}^{p^2 \times q^2}} \left\| R - \mathcal{P}(\widehat{\Sigma}) \right\|_{\mathrm{F}}^2 + \lambda \operatorname{rank}(R). \tag{4}$$

Following the covariance matching approach of Werner et al. (2008), both (3) and (4) reduce the problem of covariance estimation to recovering of a low-rank matrix $\mathcal{P}(\widehat{\Sigma})$ from noisy observations. We would like to note that the estimates $\widehat{\Sigma}^{\circ}$ and $\check{\Sigma}$ admit explicit expressions based on the singular value decomposition of $\mathcal{P}(\widehat{\Sigma})$. For this reason, they can be computed in polynomial time.

In the present paper, we consider a covariance model combining Kronecker product and tensor train (TT) structure. To be more precise, we consider $\Sigma$ of the form

$$\Sigma = \sum_{j=1}^{J} \sum_{k=1}^{K} U_j \otimes W_{jk} \otimes V_k, \tag{5}$$

where $U_j \in \mathbb{R}^{p \times p}$, $V_{jk} \in \mathbb{R}^{q \times q}$, and $W_k \in \mathbb{R}^{r \times r}$ for any $j \in \{1, \ldots, J\}$ and $k \in \{1, \ldots, K\}$. The numbers $p$, $q$, and $r$ are assumed to be such that $pqr = d$. Let us note that (5) naturally extends (2) to the case of three-way data and coincides with it when $J = 1$ and $U_1 = 1$. The rationale for selecting our model is that the TT decomposition (Oseledets, 2011) is recognized for its computational efficiency compared to the canonical polyadic (CP) decomposition, while providing a robust framework for representing higher-order tensors. Notice that the CANDECOMP/PARAFAC model

$$\Sigma = \sum_{k=1}^{K} \Phi_k \otimes \Psi_k \otimes \Omega_k, \tag{6}$$

which has recently got considerable attention in the literature (see, for example, (Pouryazdian et al., 2016; Greenewald et al., 2019; Yu et al., 2025) and the references therein), is a particular case of (5) with $J = K$, $W_{jk} = \Psi_k \mathbb{1}(j = k)$, and $U_j = \Phi_j$. Following the covariance matching approach, we can reshape a matrix $\Sigma$ of the form (5) into a third-order tensor with low canonical rank. Indeed, given a matrix $A \in \mathbb{R}^{pqr \times pqr}$, let us define a rearrangement operator $\mathcal{R} : \mathbb{R}^{pqr \times pqr} \to \mathbb{R}^{p^2 \times q^2 \times r^2}$ componentwise: for any $1 \leqslant a \leqslant p^2$, $1 \leqslant b \leqslant q^2$, and $1 \leqslant c \leqslant r^2$

$$\mathcal{R}(\Sigma)_{a,b,c} = \Sigma_{(\lceil a/p \rceil - 1) \cdot qr + (\lceil b/q \rceil - 1) \cdot r + \lceil c/r \rceil, ((a-1)\%p) \cdot qr + ((b-1)\%q) \cdot r + (c-1)\%r + 1}, \tag{7}$$

where $y\%x \in \{0, \ldots, x-1\}$ stands for the residual of $y$ modulo $x$. Then it is easy to check that

$$\mathcal{R}(\Sigma) = \sum_{j=1}^{J} \sum_{k=1}^{K} \mathbf{vec}(U_j) \otimes \mathbf{vec}(W_{jk}) \otimes \mathbf{vec}(V_k), \tag{8}$$

where, for any matrix $A$, $\mathbf{vec}(A)$ is a vector obtained by stacking the columns of $A$ together. Unfortunately, a formal extension of the approach suggested by Tsiligkaridis and Hero (2013) to the CANDECOMP/PARAFAC model will not result in a practical algorithm. The main obstacle is that approximation of the nuclear norm of a tensor is an NP-hard problem Hillar and Lim (2013). The statistical-computational gap was discussed in several papers including (Barak and Moitra, 2016; Zhang and Xia, 2018; Han et al., 2022a; Luo and Zhang, 2022; 2024). For this reason, when developing an algorithm for estimation of the covariance matrix (5), we must take into account both its computational and sample complexities. In the present paper, we extend the approach of Zhang and Xia (2018) and suggest an iterative procedure similar to the higher-order orthogonal iteration (HOOI) with the notable distinction of utilizing the Tucker-2 representation of the tensor. Our algorithm successfully adapts to the structure (5) but requires less time, than Tucker decomposition and HOOI.

While statisticians (see, for example, (Tsiligkaridis and Hero, 2013; Puchkin and Rakhuba, 2024)) established rates of convergence of the PRLS estimate (3), the CANDECOMP/PARAFAC model (6) and the more general tensor train model (5) remain underexplored. In Section 2 (see (9) below), we discuss that the tensor train model (5) can be represented in a way, which is very similar to the low Tucker rank tensor model (see, for instance, (Han et al., 2022a, Definition 2.1)). The only difference is that (9) includes two factors with orthogonal columns while in Tucker decomposition one has three such factors. For this reason, some bounds on the estimation accuracy of $\Sigma$ of the form (5) with respect to the Frobenius norm follow from the results on tensor estimation Zhang and Xia (2018); Han et al. (2022b); Kumar et al. (2025), scalar-on-tensor regression Khavari and Rabusseau (2021); Wang et al. (2025), and tensor-on-tensor regression Raskutti et al. (2019); Luo and Zhang (2024) with constraints on Tucker ranks. However, these bounds are dimension dependent, while many recent results in covariance estimation establish dimension-free bounds (see, for instance, Koltchinskii and Lounici (2017); Bunea and Xiao (2015); Abdalla and Zhivotovskiy (2022); Zhivotovskiy (2024); Puchkin and Rakhuba (2024); Puchkin et al. (2025)). To our knowledge, the existing dimension-free results on tensor estimation only cover the case of simple rank-one tensors (Vershynin, 2020; Zhivotovskiy, 2024; Al-Ghattas et al., 2025; Chen and Sanz-Alonso, 2025). In the present paper, we derive high-probability dimension-free bounds on the accuracy of estimation of third-order tensors with low TT-ranks and of the covariance matrices, which can be well approximated by (5).

**Contribution.** Our main contribution is a comprehensive non-asymptotic analysis of this estimation procedure. We first derive a general deterministic perturbation bound for our TT-SVD-like algorithm, which may be of independent interest. We then leverage this result to establish a high-probability error bound for our covariance estimator. The final bound clearly decomposes the error into a bias term, related to how well the true $\Sigma$ can be approximated by our model, and a variance term. This variance term scales gracefully with the sample size n, the TT-ranks $(J, K)$, and data-dependent effective dimensions that capture the intrinsic complexity of the covariance structure. To our knowledge, this is the first work to provide a computationally efficient and theoretically guaranteed method for covariance estimation with this flexible TT-based structure.

**Paper structure.** The rest of the paper is organized as follows. In Section 2, we present our algorithm and main theoretical guarantees. We provide some practical analysis in Section 3 and conclude with a discussion in Section 4. All proofs are deferred to the Appendix.

**Notation.** Given a matrix $M \in \mathbb{R}^{d_1 \times d_2}$, we define its vectorization as
$$\mathbf{vec}(M)_{(a-1) \cdot d_2 + b} = M_{a,b}, \quad a \leqslant d_1, b \leqslant d_2.$$
For a tensor $\mathcal{T}$ of order $k$ with dimensions $d_1, \ldots, d_k$, we define a multiplication $\times_i$ on mode $i$ by a matrix $M \in \mathbb{R}^{d' \times d_i}$ as follows:
$$(\mathcal{T} \times_i M)_{a_1 a_2 \ldots a_i a_{i+1} \ldots a_k} = \sum_{a_i'=1}^{d_i} \mathcal{T}_{a_1 a_2 \ldots a_{i-1} a_i' a_{i+1} \ldots a_k} M_{a_i a_i'},$$
where $a_j, j \neq i$, takes values in $\{1, \ldots, d_j\}$ and $a_i$ takes values in $\{1, \ldots, d'\}$.

It will be convenient to assume that random vectors $\mathbf{X}, \mathbf{X}_1, \ldots, \mathbf{X}_n$ lie in a tensor product space $\mathbb{R}^p \otimes \mathbb{R}^q \otimes \mathbb{R}^q$, so $\Sigma = \mathbb{E} \mathbf{X} \mathbf{X}^\top$ belongs to the space of SDP Hermitian operators $\mathcal{H}_+(\mathbb{R}^p \otimes \mathbb{R}^q \otimes \mathbb{R}^r)$ from $\mathbb{R}^p \otimes \mathbb{R}^q \otimes \mathbb{R}^q$ to itself. Then, we will define partial traces of $\Sigma$ as follows. Given linear spaces $L_1, L_2$ and linear operators $X : L_1 \to L_1, Y : L_2 \to L_2$, we define the partial trace $\mathrm{Tr}_{L_i}, i = 1, 2$, w.r.t. $L_i$ as follows:
$$\mathrm{Tr}_{L_1}(X \otimes Y) = \mathrm{Tr}(X) \cdot Y, \quad \mathrm{Tr}_{L_2}(X \otimes Y) = X \cdot \mathrm{Tr}(Y).$$
We extend $\mathrm{Tr}_{L_i}(\cdot)$ to all operators from $L_1 \otimes L_2 \to L_1 \otimes L_2$ by linearity. In our case, for operators from $\mathcal{H}_+(\mathbb{R}^p \otimes \mathbb{R}^q \otimes \mathbb{R}^r)$, we define $\mathrm{Tr}_1(\cdot)$ as a partial trace w.r.t. $\mathbb{R}^p$, $\mathrm{Tr}_2(\cdot)$ as a partial trace w.r.t. $\mathbb{R}^q$ and $\mathrm{Tr}_3(\cdot)$ as a partial trace w.r.t. $\mathbb{R}^r$. Partial traces will play in important role in our theoretical analysis. We define
$$\mathtt{r}_1(\Sigma) = \max\left\{\frac{\|\mathrm{Tr}_1(\Sigma)\|}{\|\Sigma\|}, \frac{\|\mathrm{Tr}_{1,2}(\Sigma)\|}{\|\mathrm{Tr}_2(\Sigma)\|}\right\}, \qquad \mathtt{r}_2(\Sigma) = \max\left\{\frac{\|\mathrm{Tr}_2(\Sigma)\|}{\|\Sigma\|}, \frac{\|\mathrm{Tr}_{2,3}(\Sigma)\|}{\|\mathrm{Tr}_3(\Sigma)\|}\right\},$$
$$\mathtt{r}_3(\Sigma) = \max\left\{\frac{\|\mathrm{Tr}_3(\Sigma)\|}{\|\Sigma\|}, \frac{\|\mathrm{Tr}_{1,3}(\Sigma)\|}{\|\mathrm{Tr}_1(\Sigma)\|}, \frac{\|\mathrm{Tr}_{1,2,3}(\Sigma)\|}{\|\mathrm{Tr}_{1,2}(\Sigma)\|}\right\},$$

where $\text{Tr}_{i_1 i_2 \dots i_k}$ stands for the composition of the traces $\text{Tr}_{i_1}, \text{Tr}_{i_2}, \dots, \text{Tr}_{i_k}$. Quantities $\mathbf{r}_1(\Sigma), \mathbf{r}_2(\Sigma), \mathbf{r}_3(\Sigma)$ play the role of effective dimensions. From (Rastegin, 2012, display (23)), we know that $\mathbf{r}_1(\Sigma) \leqslant p, \mathbf{r}_2(\Sigma) \leqslant q, \mathbf{r}_3(\Sigma) \leqslant r$. We define them as maxima over ratios of some partial traces to ensure that for any non-empty set $S \subset \{1, 2, 3\}$ we have

$$\frac{\|\text{Tr}_S(\Sigma)\|}{\|\Sigma\|} \leqslant \prod_{s \in S} \mathbf{r}_s(\Sigma).$$

For a tensor $\mathcal{T} \in \mathbb{R}^{p^2 \times q^2 \times r^2}$, we introduce the unfolding operator with respect to the first mode as

$$\mathbf{m}_1(\mathcal{T})_{x,y} = \mathcal{T}_{x, \lceil y/r^2 \rceil, (y-1)\%r^2+1}.$$

Similarly, the unfolding operators with respect to the second and the third modes are define as follows:

$$\mathbf{m}_2(\mathcal{T})_{x,y} = \mathcal{T}_{(y-1)\%p^2+1, x, \lceil y/p^2 \rceil}, \quad \mathbf{m}_3(\mathcal{T})_{x,y} = \mathcal{T}_{\lceil y/q^2 \rceil, (y-1)\%q^2+1, x}.$$

We denote the output of SVD algorithm with hard thresholding via rank $J$ as $SVD_J$. We denote matrices with orthonormal columns of size $\mathbb{R}^{d \times r}$ by $\mathbb{O}_{d,r}$. In what follows, $[m]$ stands for the set of integers from 1 to $m$.

## 2 MAIN RESULTS

Let us return to the estimation of the covariance matrix $\Sigma$ of the form (5). As discussed in the introduction, we can reshape $\Sigma$ into a third-order tensor $\mathcal{R}(\Sigma)$ using the rearrangement operator (7):

$$\mathcal{R}(\Sigma) = \sum_{j=1}^{J} \sum_{k=1}^{K} \mathbf{vec}(U_j) \otimes \mathbf{vec}(W_{jk}) \otimes \mathbf{vec}(V_k) \in \mathbb{R}^{p^2 \times q^2 \times r^2},$$

where vectors $\mathbf{vec}(U_j)$ are assumed to be linearly independent, as well as vectors $\mathbf{vec}(V_k)$. Stacking together vectors $\mathbf{vec}(U_j)$, $j = 1, \dots, J$ into a matrix $U \in \mathbb{R}^{p^2 \times J}$, vectors $\mathbf{vec}(V_k)$, $k = 1, \dots, K$ into a matrix $V \in \mathbb{R}^{r^2 \times K}$ and matrices $W_{jk}$, $j = 1, \dots, J$, $k = 1, \dots, K$ into a three-dimensional tensor $\mathcal{W} \in \mathbb{R}^{J \times q^2 \times K}$, we can rewrite the above decomposition in the following compact form:

$$\mathcal{R}(\Sigma) = \mathcal{W} \times_3 V \times_1 U. \tag{9}$$

Note that this decomposition is not unique. In particular, multiplying $U$ by an invertible matrix $Q_U \in \mathbb{R}_{J,J}$ from the right and $\mathcal{W}$ by $Q_U^{-1}$ from the first mode does not change the right-hand side of (9). The same true for the factor $V$. Hence, one can assume that the columns of $U$ and $V$ are orthonormal, i.e. $U \in \mathbb{O}_{p^2,J}$ and $V \in \mathbb{O}_{r^2,K}$. In what follows, we always assume that this is the case. For brevity, we set $d_1 = p^2$, $d_2 = q^2$, and $d_3 = r^2$.

We extend the model (5) to the case when $\Sigma$ can be approximated by decomposition (5) up to some error. Then, it is naturally to consider the best $(J, K)$-TT-rank approximation of $\mathcal{R}(\Sigma)$, which we denote by $\mathcal{T}^*$. We denote the misspecification shift $\mathcal{R}(\Sigma) - \mathcal{T}^*$ by $\overline{\mathcal{E}}$. To approximate $\Sigma$, we aim to recover its structured part $\mathcal{T}^*$ from the noisy tensor $\mathcal{Y} = \mathcal{R}(\widehat{\Sigma})$, which can be represented as

$$\mathcal{Y} = \mathcal{T}^* + \mathcal{E} \in \mathbb{R}^{d_1 \times d_2 \times d_3},$$

where the error tensor $\mathcal{E}$ consists of the approximation part $\overline{\mathcal{E}}$ and the noise part $\widehat{\mathcal{E}} = \mathcal{R}(\widehat{\Sigma}) - \mathcal{R}(\Sigma)$.

Since $\mathcal{T}^*$ has TT-ranks $(J, K)$, it can be decomposed as $\mathcal{T}^* = \mathcal{W}^* \times_3 V^* \times_1 U^*$, where $U^* \in \mathbb{O}_{p^2,J}$, $V^* \in \mathbb{O}_{r^2,K}$ and $\mathcal{W}^* \in \mathbb{R}^{J \times q^2 \times K}$. This decomposition suggests the following natural algorithm for estimating $\mathcal{T}^*$ from $\mathcal{Y}$. Using truncated SVD, one estimates the image of $U^*$ which coincides with $\text{Im}\,\mathbf{m}_1(\mathcal{T}^*)$, then estimates the image of $V^*$ which coincides with $\text{Im}\,\mathbf{m}_3(\mathcal{T}^*)$, and then project $\mathcal{Y}$ onto the estimated spaces. However, this estimation is not straightforward, and one should apply truncated SVD iteratively to reach reasonable accuracy. In Section 3, we conduct numerical experiments illustrating that additional iterations indeed improve the estimation. We summarized the resulting procedure as Algorithm 1. We refer to it as the Hartth algorithm where the abbreviation HardTTh stands for **Hard Tensor Train Thresholding**.

---

**Algorithm 1:** HardTTh

---

**Input:** Tensor $\mathcal{Y} \in \mathbb{R}^{d_1 \times d_2 \times d_3}$, TT-ranks $(J, K)$, number of steps $T$

**Output:** TT-approximation $\widehat{\mathcal{T}} = \widehat{\mathcal{W}} \times_3 \widehat{V} \times_1 \widehat{U}$, where $\widehat{U} \in \mathbb{O}_{d_1, J}, \widehat{V} \in \mathbb{O}_{d_2, K}$,
$\qquad \widehat{\mathcal{W}} \in \mathbb{R}^{J \times d_2 \times K}$;

Find SVD of $\mathtt{m}_1(\mathcal{Y})$ truncated on the first $J$ singular values: $\widehat{U}_0, \Sigma_{0,1}, \widetilde{U}_0 = \mathrm{SVD}_J(\mathtt{m}_1(\mathcal{Y}))$

Find truncated SVD of $\mathtt{m}_3(\mathcal{Y} \times_1 \widehat{U}_0^\top)$: $\widehat{V}_0, \Sigma_{0,2}, \widetilde{V}_0 = \mathrm{SVD}_K(\mathtt{m}_3(\mathcal{Y} \times_1 \widehat{U}_0^\top))$

**for** $t = 1, \ldots, T$ **do**
$\qquad$ Set $\widehat{U}_t, \Sigma_{t,1}, \widetilde{U}_t = \mathrm{SVD}_J(\mathtt{m}_1(\mathcal{Y} \times_3 \widehat{V}_{t-1}^\top))$
$\qquad$ Set $\widehat{V}_t, \Sigma_{t,2}, \widetilde{V}_t = \mathrm{SVD}_K(\mathtt{m}_3(\mathcal{Y} \times_1 \widehat{U}_t^\top))$

Set $\widehat{U} = \widehat{U}_T, \widehat{V} = \widehat{V}_T$ and $\widehat{\mathcal{W}} = \mathcal{Y} \times_3 \widehat{V}^\top \times_1 \widehat{U}^\top$.

---

Notice that computational complexity of Algorithm 1 is determined by the complexity of truncated SVD applied to the matricizations. The truncated $SVD_J$ at the first step of HardTTh takes $O(d_1 d_2 d_3 \cdot \min\{d_1, d_2 d_3\})$. Other steps require either $O(J d_3 d_2 \cdot \min\{d_3, J d_2\} + J d_1 d_2 d_3)$ or $O(K d_1 d_2 \min\{d_1, K d_2\} + K d_1 d_2 d_3)$ flops, so the overall complexity of the algorithm is

$$O((J + K)T d_1 d_2 d_3 + T K d_1 d_2 \cdot \min\{d_1, K d_2\} + T J d_3 d_2 \cdot \min\{d_3, J d_2\}$$
$$+ d_1 d_2 d_3 \cdot \min\{d_1, d_2 d_3\}).$$

If the misspecification is not too large, the number $T$ of iterations can be taken logarithmical in the ambient dimensions, see discussion below after Theorem 2.2.

In practice, randomized truncated SVD could be used (Halko et al., 2011) or other approximate algorithms (Baglama and Reichel, 2005).

Given the output $\widehat{\mathcal{T}}$ of Algorithm 1 applied to $\mathcal{Y} = \mathcal{R}(\widehat{\Sigma})$, define the estimator $\widetilde{\Sigma}$ of $\Sigma$ as $\widetilde{\Sigma} = \mathcal{R}^{-1}(\widehat{\mathcal{T}})$. To analyze rates of convergence for this estimator, we impose some assumption on the distribution of $\mathbf{X}_i$.

**Assumption 2.1.** *There exists $\omega > 0$, such that the standardized random vector $\Sigma^{-1/2}\mathbf{X}$ satisfies the inequality*

$$\log \mathbb{E} \exp \left\{ (\Sigma^{-1/2}\mathbf{X})^\top V (\Sigma^{-1/2}\mathbf{X}) - \mathrm{Tr}(V) \right\} \leqslant \omega^2 \|V\|_\mathrm{F}^2 \tag{10}$$

*for all $V \in \mathbb{R}^{d \times d}$, such that $\|V\|_\mathrm{F} \leqslant 1/\omega$.*

In (Puchkin et al., 2025), the authors showed that Assumption 2.1 holds for a large class of distribution. Indeed, Assumption 2.1 is a weaker version of the Hanson–Wright inequality. In particular, if the Hanson–Wright inequality is fulfilled for $\Sigma^{-1/2}\mathbf{X}$, then $\mathbf{X}$ satisfies Assumption 2.1. Therefore, Assumption 2.1 can be used when $\Sigma^{-1/2}\mathbf{X}$ is multivariate standard Gaussian, consists of i.i.d. sub-Gaussian random variables, satisfies the logarithmic Sobolev inequality or the convex concentration property (Adamczak, 2015).

Under Assumption 2.1, we establish the following theorem. We give its proof in Appendix E. The proof sketch is given in Appendix D.

**Theorem 2.2.** *Fix $\delta \in (0, 1)$. Grant Assumption 2.1. Suppose that singular values $\sigma_J(\mathtt{m}_1(\mathcal{R}(\Sigma)), \sigma_K(\mathtt{m}_3(\mathcal{R}(\Sigma))$ satisfy*

$$\sigma_J(\mathtt{m}_1(\mathcal{R}(\Sigma))) \geqslant 25\|\mathtt{m}_1(\overline{\mathcal{E}})\| + 768\omega\|\Sigma\|\sqrt{\frac{\mathtt{r}_1^2(\Sigma) + \mathtt{r}_2^2(\Sigma)\mathtt{r}_3^2(\Sigma) + \log(6/\delta)}{n}},$$

$$\sigma_K(\mathtt{m}_3(\mathcal{R}(\Sigma))) \geqslant 25\|\mathtt{m}_3(\overline{\mathcal{E}})\| + 768\omega\|\Sigma\|\sqrt{\frac{J\mathtt{r}_1^2(\Sigma) + J\mathtt{r}_2^2(\Sigma) + \mathtt{r}_3^2(\Sigma) + \log(48/\delta)}{n}}.$$

*Then, we have*

$$\|\widetilde{\Sigma} - \Sigma\|_\mathrm{F} \leqslant \overline{\mathbf{b}} + 96\omega\|\Sigma\|\sqrt{\frac{J\mathtt{r}_1^2(\Sigma) + JK\mathtt{r}_2^2(\Sigma) + K\mathtt{r}_3^2(\Sigma) + \log(48/\delta)}{n}} + \widetilde{\diamondsuit}_2 + \widetilde{r}_T$$

*with probability at least $1 - \delta$, provided $n \geqslant \mathtt{R}_\delta$, where*

$$\overline{\mathbf{b}} = \|\overline{\mathcal{E}}\|_{\mathrm{F}} + 5\sqrt{J}\|\mathtt{m}_1(\overline{\mathcal{E}})\| + 5\sqrt{K}\|\mathtt{m}_3(\overline{\mathcal{E}})\|,$$

*and $\mathtt{R}_\delta$ and remainder terms $\widetilde{\lozenge}_2, \widetilde{r}_T$ are defined in Table 1.*

| Variable | Expression |
|---|---|
| $\widetilde{\alpha}_U$ | $\|\mathtt{m}_1(\overline{\mathcal{E}} \times_3 (V^*)^\top)\| + 32\omega\|\Sigma\|\sqrt{\frac{\mathtt{r}_1^2(\Sigma) + K\mathtt{r}_2^2(\Sigma) + \log(48/\delta)}{n}}$ |
| $\widetilde{\beta}_U$ | $\sup_{\substack{V \in \mathbb{R}^{d_2 \times K} \\ \|V\| \leqslant 1}} \|\mathtt{m}_1(\overline{\mathcal{E}} \times_3 V^\top)\| + 32\omega\|\Sigma\|\sqrt{\frac{\mathtt{r}_1^2(\Sigma) + K\mathtt{r}_2^2(\Sigma) + K\mathtt{r}_3^2(\Sigma) + \log(48/\delta)}{n}}$ |
| $\widetilde{\alpha}_V$ | $\|\mathtt{m}_3(\overline{\mathcal{E}} \times_1 (U^*)^\top)\| + 32\omega\|\Sigma\|\sqrt{\frac{\mathtt{r}_3^2(\Sigma) + J\mathtt{r}_2^2(\Sigma) + \log(48/\delta)}{n}}$ |
| $\widetilde{\beta}_V$ | $\sup_{\substack{U \in \mathbb{R}^{d_1 \times J} \\ \|U\| \leqslant 1}} \|\mathtt{m}_3(\overline{\mathcal{E}} \times_1 U^\top)\| + 32\omega\|\Sigma\|\sqrt{\frac{\mathtt{r}_2^2(\Sigma) + J\mathtt{r}_1^2(\Sigma) + J\mathtt{r}_2^2(\Sigma) + \log(48/\delta)}{n}}$ |
| $\widetilde{\lozenge}_2$ | $96\left(\frac{\sqrt{K}\widetilde{\beta}_V\widetilde{\alpha}_U}{\sigma_J(\mathtt{m}_1(\mathcal{R}(\Sigma)))} + \frac{\sqrt{J}\widetilde{\beta}_U\widetilde{\alpha}_V}{\sigma_K(\mathtt{m}_3(\mathcal{R}(\Sigma)))}\right)$ |
| $\widetilde{r}_T$ | $(\sqrt{J} + \sqrt{K}) \cdot \left(\frac{200\widetilde{\beta}_V\widetilde{\beta}_U}{\sigma_J(\mathtt{m}_1(\mathcal{R}(\Sigma)))\sigma_K(\mathtt{m}_3(\mathcal{R}(\Sigma)))}\right)^T \times$ $\times \left(\|\mathtt{m}_1(\overline{\mathcal{E}})\| + 32\omega\sqrt{\frac{\mathtt{r}_1^2(\Sigma) + \mathtt{r}_2^2(\Sigma)\mathtt{r}_3^2(\Sigma) + \log(6/\delta)}{n}}\right)$ |
| $\mathtt{R}_\delta$ | $J\mathtt{r}_1^2(\Sigma) + JK\mathtt{r}_2^2(\Sigma) + K\mathtt{r}_3^2(\Sigma) + \mathtt{r}_2^2(\Sigma)\mathtt{r}_3^2(\Sigma) + \log(48/\delta)$ |

Table 1: List of ancillary variables

The upper bound on $\|\widetilde{\Sigma} - \Sigma\|_{\mathrm{F}}$ provided by the above theorem can be decomposed into the bias term $\overline{\mathbf{b}}$ due to model misspecification, the leading variance term

$$\widehat{\mathbf{v}} = 96\omega\|\Sigma\|\sqrt{\frac{J\mathtt{r}_1^2(\Sigma) + JK\mathtt{r}_2^2(\Sigma) + K\mathtt{r}_3^2(\Sigma) + \log(48/\delta)}{n}},$$

and remainder terms $\widetilde{\lozenge}_2, \widetilde{r}_T$. Note that after $T = O(\log(JK\mathtt{r}_2(\Sigma)))$ iterations, the variance part

$$\widetilde{r}_T^v = (\sqrt{J} + \sqrt{K}) \cdot \left(\frac{200\widetilde{\beta}_V\widetilde{\beta}_U}{\sigma_J(\mathtt{m}_1(\mathcal{R}(\Sigma)))\sigma_K(\mathtt{m}_3(\mathcal{R}(\Sigma)))}\right)^T$$

$$\times 32\omega\sqrt{\frac{\mathtt{r}_1^2(\Sigma) + \mathtt{r}_2^2(\Sigma)\mathtt{r}_3^2(\Sigma) + \log(6/\delta)}{n}},$$

of $\widetilde{r}_T$ will be dominated by $\widehat{\mathbf{v}}$.

Compared to the known results in the literature, Theorem 2.2 has several advantages. First, it provides dimension-free bounds based on the effective dimensions $\mathtt{r}_i(\Sigma) \leqslant d_i$ instead of bounds involving ambient dimensions $d_1, d_2, d_3$ as in vast of literature on high-dimensional tensor estimation (cf. (Zhang and Xia, 2018; Qin et al., 2025; Han et al., 2022b; Tang et al., 2025; Luo and Zhang, 2024)). Second, we point out the following. Set $\mathtt{r}(\Sigma) = \mathrm{Tr}(\Sigma)/\|\Sigma\|$. It is known that, under some assumptions, the sample covariance matrix $\widehat{\Sigma}$ satisfies concentration inequalities

$$\|\widehat{\Sigma} - \Sigma\| \lesssim \|\Sigma\|\sqrt{\frac{\mathtt{r}(\Sigma) + \log(1/\delta)}{n}}, \qquad \|\widehat{\Sigma} - \Sigma\|_{\mathrm{F}} \lesssim \|\Sigma\|\sqrt{\frac{\mathtt{r}^2(\Sigma) + \log(1/\delta)}{n}}$$

with probability at least $1 - \delta$ (see (Zhivotovskiy, 2024; Bunea and Xiao, 2015; Hsu et al., 2012; Puchkin et al., 2025)), where $\lesssim$ hides some distribution-dependent constant. Hence, our effective dimensions $\mathtt{r}_i(\Sigma)$ naturally extends the effective dimension $\mathtt{r}(\Sigma)$ of sample covariance concentration in the unstructured case. Third, while Puchkin and Rakhuba (2024) prove dimension-free bounds for the model (2) and the estimator $\widehat{\Sigma}^\circ = \mathcal{P}^{-1}(\widetilde{R})$ defined by (3), they do not analyze the misspecification case and bound the variance term with probability at least $1 - \delta$ as follows:

$$\|\widehat{\Sigma}^\circ - \Sigma\|_{\mathrm{F}} \lesssim \sqrt{K}\omega\sum_{k=1}^K \|\Phi_k\|\|\Psi_k\|\sqrt{\frac{\max_k \mathtt{r}^2(\Psi_k) + \max_k \mathtt{r}^2(\Phi_k) + \log(1/\delta)}{n}},$$

so they have rough variance proxy factor $\sum_{k=1}^{K} \|\Phi_k\| \|\Psi_k\|$ instead of $\|\Sigma\| = \|\sum_{k=1}^{K} \Phi_k \otimes \Psi_k\|$. We improve their analysis to establish bounds on the variance involving variance proxy factor $\|\Sigma\|$ which seems to be tight.

To highlight the advances of Theorem 2.2, let us discuss how effective dimensions could be small compared to the ambient dimensions. In Appendix C, we prove the following proposition.

**Proposition 2.3.** *Suppose that a covariance matrix* $\Sigma \in \mathcal{H}_+(\mathbb{R}^p \otimes \mathbb{R}^q \otimes \mathbb{R}^r)$ *can be represented in the form*

$$\Sigma = \sum_{j=1}^{J} \sum_{k=1}^{K} U_j \otimes V_{jk} \otimes W_k$$

*for some symmetric positive semidefinite matrices* $U_j, W_{jk}, V_k$. *Then, we have*

$$\mathbf{r}_1(\Sigma) \leqslant J \cdot \max_j \mathbf{r}(U_j), \quad \mathbf{r}_2(\Sigma) \leqslant JK \cdot \max_{jk} \mathbf{r}(W_{jk}), \quad \mathbf{r}_3(\Sigma) \leqslant K \cdot \max_k \mathbf{r}(V_k).$$

For example, Proposition 2.3 implies that if the spectra of matrices $U_j, W_{jk}$ and $V_k$ decay quadratically, i.e. if $\max_{jk}\{\sigma_i(U_j)/\|U_j\|, \sigma_i(W_{jk})/\|W_{jk}\|, \sigma_i(V_k)/\|V_k\|\} \leqslant C_\sigma i^{-2}$, then $\mathbf{r}_1(\Sigma) \leqslant C_\sigma \pi^2/6 \cdot J$, $\mathbf{r}_2(\Sigma) \leqslant C_\sigma \pi^2/6 \cdot JK$ and $\mathbf{r}_3(\Sigma) \leqslant C_\sigma \pi^2/6 \cdot K$.

The main drawback of Theorem 2.2 is the requirements $\sigma_J(\mathbf{m}_1(\mathcal{R}(\Sigma))) \gtrsim \|\Sigma\|\sqrt{\mathbf{r}_2^2(\Sigma)\mathbf{r}_3^2(\Sigma)/n}$ and $n \gtrsim \mathbf{r}_2^2(\Sigma)\mathbf{r}_3^2(\Sigma)$. Indeed, the theory of tensor estimation by SVD-based algorithms developed in (Zhang and Xia, 2018; Tang et al., 2025) suggests that the minimax error can be achieved under condition

$$\sigma_J(\mathbf{m}_1(\mathcal{R}(\Sigma))) \gtrsim \|\Sigma\|/n^{1/2} \cdot (d_2 d_3)^{3/8}, \tag{11}$$

and there is strong evidence that the power $3/8$ in the above inequality can not be taken smaller for any polynomial-time algorithm (Barak and Moitra, 2016; Hopkins et al., 2015; Zhang and Xia, 2018; Luo and Zhang, 2024; Diakonikolas et al., 2023). However, minimax bounds under conditions of the type (11) were established when entries of $\widehat{\mathcal{E}}$ are i.i.d. Roughly speaking, the estimation error of the singular subspaces corresponds to the impact of the term $\mathbf{m}_1(\mathcal{E})\mathbf{m}_1(\mathcal{E})^\top$ in the decomposition

$$\mathbf{m}_1(\mathcal{Y})\mathbf{m}_1(\mathcal{Y})^\top = \mathbf{m}_1(\mathcal{T}^*)\mathbf{m}_1(\mathcal{T}^*)^\top + \mathbf{m}_1(\mathcal{T}^*)\mathbf{m}_1(\mathcal{E})^\top + \mathbf{m}_1(\mathcal{E})\mathbf{m}_1(\mathcal{T}^*)^\top + \mathbf{m}_1(\mathcal{E})\mathbf{m}_1(\mathcal{E})^\top$$

on the perturbation of eigenspace of $\mathbf{m}_1(\mathcal{T}^*)\mathbf{m}_1(\mathcal{T}^*)^\top$, see (Cai and Zhang, 2018). When entries of $\widehat{\mathcal{E}}$ are i.i.d., we have $\mathbb{E}\mathbf{m}_1(\widehat{\mathcal{E}})\mathbf{m}_1(\widehat{\mathcal{E}})^\top = \alpha I_{d_1}$ for some scalar $\alpha$, so the error of singular subspaces estimation is determined by deviations of $\mathbf{m}_1(\widehat{\mathcal{E}})^\top\mathbf{m}_1(\widehat{\mathcal{E}})^\top$ from its mean, which can be controlled under conditions like (11). This is clearly not the case of our setup, so Algorithm 1 requires debiasing before applying SVD, which needs extra assumptions on the distribution of $\mathbf{X}_i$ and is left for future work.

Comparing Theorem 2.2 with results of Zhang and Xia (2018), one can note that, in their paper, upper bounds on the tensor estimation error do not involve second-order terms like $\widetilde{\Diamond}_2$. The reason is that their work imposes an assumption $\max\{d_1, d_2, d_3\} \leqslant C \min\{d_1, d_2, d_3\}$ for some absolute constant $C$. Translated to our setup, it means that, assuming $\max_i \mathbf{r}_i(\Sigma) \leqslant C \min_i \mathbf{r}_i(\Sigma)$, the term $\widetilde{\Diamond}_2$ is dominated by the leading variance term $\widehat{\mathbf{v}}$, which is exactly the case.

Finally, we briefly comment on the choice of $J$ and $K$. If $\Sigma$ can be represented by (5) for some $J, K$, such that

$$\sigma_J(\mathbf{m}_1(\mathcal{R}(\Sigma))) \geqslant C\omega\|\Sigma\|\sqrt{\frac{\mathbf{r}_1^2(\Sigma) + \mathbf{r}_2^2(\Sigma)\mathbf{r}_3^2(\Sigma) + \log(6/\delta)}{n}},$$

$$\sigma_K(\mathbf{m}_3(\mathcal{R}(\Sigma))) \geqslant C\omega\|\Sigma\|\sqrt{\frac{J\mathbf{r}_2^2(\Sigma) + J\mathbf{r}_2^2(\Sigma) + \mathbf{r}_3^2(\Sigma) + \log(48/\delta)}{n}}$$

for some large enough absolute constant $C$, and the following bounds hold

$$\|\Sigma\|/2 \leqslant \|\widehat{\Sigma}\| \leqslant 3\|\Sigma\|/2,$$

$$\|\mathrm{Tr}_S(\widehat{\Sigma}) - \mathrm{Tr}_S(\Sigma)\| \leqslant \frac{1}{2}\|\mathrm{Tr}_S(\Sigma)\| \text{ for all non-empty } S \subset [3] \tag{12}$$

with probability at least $1 - \delta/6$, then one can define estimators $\widehat{J}, \widehat{K}$ of $J, K$ as

$$\widehat{J} = \max \left\{ J' \mid \sigma_{J'}(\mathtt{m}_1(\mathcal{R}(\widehat{\Sigma}))) \geqslant C'\omega\|\widehat{\Sigma}\|\sqrt{\frac{\mathtt{r}_1^2(\widehat{\Sigma}) + \mathtt{r}_2^2(\widehat{\Sigma})\mathtt{r}_3^2(\widehat{\Sigma}) + \log(6/\delta)}{n}} \right\}, \qquad (13)$$

$$\widehat{K} = \max \left\{ K' \mid \sigma_{K'}(\mathtt{m}_3(\mathcal{R}(\widehat{\Sigma}))) \geqslant C'\omega\|\widehat{\Sigma}\|\sqrt{\frac{\widehat{J}\mathtt{r}_1^2(\widehat{\Sigma}) + \widehat{J}\mathtt{r}_2^2(\widehat{\Sigma}) + \mathtt{r}_3^2(\widehat{\Sigma}) + \log(48/\delta)}{n}} \right\},$$

where $C'$ is some other absolute constant and $\omega$ is assumed to be known. For example, one can compute $\omega$ explicitly when $\mathbf{X}_i$ are linear transform of Gaussian random variables. For such $\widehat{J}$, we will have

$$\sigma_{\widehat{J}}(\mathtt{m}_1(\mathcal{R}(\Sigma))) > 768\omega\|\Sigma\|\sqrt{\frac{\mathtt{r}_1^2(\Sigma) + \mathtt{r}_2^2(\Sigma)\mathtt{r}_3^2(\Sigma) + \log(6/\delta)}{n}} \geqslant \|\mathtt{m}_1(\widehat{\mathcal{E}})\|,$$

with probability $1 - \delta/6$ (see Lemma E.1 in Appendix), implying $\widehat{J} \leqslant J$. If $C$ is significantly larger than $C'$, then the singular value $\sigma_J(\mathtt{m}_1(\mathcal{R}(\widehat{\Sigma}))) \geqslant \sigma_J(\mathtt{m}_1(\mathcal{R}(\Sigma))) - \|\mathtt{m}_1(\widehat{\mathcal{E}})\|$ satisfies the inequality of the definition (13) with probability at least $1 - \delta/6$, so $J \leqslant \widehat{J}$, and we conclude $J = \widehat{J}$ with probability at least $1 - \delta/2$. Analogously, one can show that $K = \widehat{K}$ for suitable choice of $C, C'$ with probability at least $1 - \delta/2$, yielding $J = \widehat{J}$ and $K = \widehat{K}$ with probability at least $1 - \delta$.

Then, while applying Algorithm 1 with $J \leqslant \widehat{J}, K \leqslant \widehat{K}$ could lead to better bias-variance tradeoff, using $J > \widehat{J}$ will result in much worse convergence rate in our model.

However, this holds assuming that (12) is fulfilled, so concentration bounds should be established for the norms of partial traces, which we left for future research.

## 3 EXPERIMENTS

In the present section, we illustrate that additional iterations $T$ of HardTTh indeed improve the estimation of the covariance matrix $\Sigma$ provided singular values of matricizations satisfy conditions of Theorem 2.2 up to some constant. We also compare HardTTh with several other algorithms.

To illustrate our theory, we construct a sampling model with the covariance matrix $\Sigma$ satisfying (5) as follows. Set $J = 7, K = 9$ and $p = q = r = 10$. Let $\mathcal{E}^{ijk}, i \in [n], j \in [J], k \in [K]$ be $n \cdot JK$ tensors of shape $(p, q, r)$ consisting of i.i.d. standard Gaussian entries. Let $A_j \in \mathbb{R}^{p \times p}, B_{jk} \in \mathbb{R}^{q \times q}, C_k \in \mathbb{R}^{r \times r}$ be random symmetric matrices with diagonal and upper diagonal entries being i.i.d. Gaussian as well. Then, random vectors $\mathbf{X}_1, \ldots, \mathbf{X}_n$ are defined as vectorized tensors

$$\sum_{j=1}^{J} \sum_{k=1}^{K} \mathcal{E}^{ijk} \times_3 C_k \times_2 B_{jk} \times_1 A_j \in \mathbb{R}^{p \times q \times r},$$

conditioned on $A_j, B_{jk}, C_k$. The covariance matrix $\Sigma$ of $\mathbf{X}_i$ satisfies (see Puchkin and Rakhuba (2024))

$$\Sigma = \sum_{j=1}^{J} \sum_{k=1}^{K} A_j^2 \otimes B_{jk}^2 \otimes C_k^2.$$

We propose several algorithms for comparative analysis with HardTTh. Specifically, we consider a version of Algorithm 1 with $T = 0$ additional steps, to which we refer as TT-HOSVD. This algorithm computes an approximate Tucker-2 decomposition of a noisy tensor $\mathcal{R}(\widehat{\Sigma}) \approx \widehat{\mathcal{W}} \times_3 \widehat{V}_0 \times_1 \widehat{U}_0$, and output the estimatior $\widehat{\mathcal{W}} \times_3 \widehat{V}_0 \times_1 \widehat{U}_0$ of $\mathcal{R}(\Sigma)$. We use this comparison to justify whether additional iterations are indeed necessary.

Furthermore, we modify the algorithm proposed in Tsiligkaridis and Hero (2013) for use in our context. Instead of a single parameter $\lambda$ to control soft-thresholding, two distinct parameters are passed for each of the first and third matricizations of $\mathcal{R}(\widehat{\Sigma})$. Using the first one, soft-thresholding upon first matricization is applied, then tensor is reshaped and soft-thresholding with another parameter upon

third matricization is used. Then, we reshape the obtained tensor $\widehat{\mathcal{X}}$ back into a matrix $\mathcal{R}^{-1}(\widehat{\mathcal{X}})$ of size $pqr \times pqr$. The pseudocode is given in Algorithm 2 in Appendix H.1.

Finally, we compare HardTTh with the approximate Tucker decomposition with the Tucker ranks $(J, JK, K)$ using HOOI (Higher Order Orthogonal Iterations) algorithm of Zhang and Xia (2018). If no additional iterations in this algorithm were applied, we refer to it as "Tucker" in our tables. Otherwise, we refer to it as "Tucker+HOOI".

We also include the sample covariance estimator into our comparative analysis.

We conduct several experiments varying the number of samples $n$. For $n = 500$, the result is given in Table 2. For $n = 2000$, the result is given in Table 3. Other values of $n$ are studied in Appendix H. For each estimator $\widehat{S}$ of $\Sigma$, we compute the relative error $\|\widehat{S} - \Sigma\|_{\mathrm{F}}/\|\Sigma\|_{\mathrm{F}}$ in the Frobenius norm. For each $n$, we tune parameters $\lambda_1, \lambda_2$ of the PRLS algorithm over a log-scale grid. We fix the number of iterations $T$ of HardTTh to 10.

Table 2: Performance comparison of tensor decomposition algorithms for $n = 500$. Relative errors were averaged over 32 repeats of the experiment, empirical standard deviation is given after $\pm$ sign. The best results are boldfaced.

| Metric | Algorithm | | |
|---|---|---|---|
| | Sample Mean | TT-HOSVD | HardTTh |
| Relative Error | $1.22 \pm 0.02$ | $0.269 \pm 0.008$ | $\mathbf{0.238 \pm 0.013}$ |
| Time (seconds) | $0.007 \pm 0.003$ | $1.9 \pm 0.8$ | $2.7 \pm 0.8$ |

| Metric | Algorithm | | |
|---|---|---|---|
| | Tucker | Tucker+HOOI | PRLS |
| Relative Error | $0.252 \pm 0.007$ | $0.240 \pm 0.013$ | $\mathbf{0.238 \pm 0.017}$ |
| Time (seconds) | $41.3 \pm 1.7$ | $81.6 \pm 3.5$ | $0.7 \pm 0.3$ |

Table 3: Performance comparison of tensor decomposition algorithms for $n = 2000$. Relative errors were averaged over 16 repeats of the experiment, empirical standard deviation is given after $\pm$ sign. The best results are boldfaced.

| Metric | Algorithm | | |
|---|---|---|---|
| | Sample Mean | TT-HOSVD | HardTTh |
| Relative Error | $0.611 \pm 0.009$ | $0.154 \pm 0.006$ | $\mathbf{0.082 \pm 0.005}$ |
| Time (seconds) | $0.010 \pm 0.007$ | $1.7 \pm 0.6$ | $4.1 \pm 1.1$ |

| Metric | Algorithm | | |
|---|---|---|---|
| | Tucker | Tucker+HOOI | PRLS |
| Relative Error | $0.150 \pm 0.005$ | $\mathbf{0.082 \pm 0.005}$ | $0.216 \pm 0.012$ |
| Time (seconds) | $39.9 \pm 5.2$ | $74.2 \pm 8.1$ | $0.6 \pm 0.3$ |

Note that while the sample size increases by $4$, the relative error of HardTTh decreases by $3$, contradicting the $1/\sqrt{n}$ dependence between estimation error and the sample size. The reason is that for $n = 500$ neither TT-HOSVD nor HardTTh is able to reconstruct bases of $\mathrm{Im}\,\mathtt{m}_1(\mathcal{R}(\Sigma))$ and $\mathrm{Im}\,\mathtt{m}_3(\mathcal{R}(\Sigma))$, so the leading error is determined by the lost components of these bases. Hence, one indeed needs some condition on the least singular values of matricizations of $\mathcal{R}(\Sigma)$. When $n = 2000$, HardTTh is able to approximate these bases, yielding a much better performance, while TT-HOSVD cannot approximate them. It is instructive to look at $\sin\Theta$-distance between $\mathrm{Im}\,\widehat{U}_0, \mathrm{Im}\,\widehat{U}_T$ and $\mathrm{Im}\,U^*$. If $n = 500$, then both $\mathrm{Im}\,\widehat{U}_0, \mathrm{Im}\,\widehat{U}_T$ have $\sin\Theta$-distance to $\mathrm{Im}\,U^*$ around $1$. But for $n = 2000$, while $\sin\Theta(\mathrm{Im}\,\widehat{U}_0, \mathrm{Im}\,U^*)$ is still around $1$, we have $\sin\Theta(\mathrm{Im}\,\widehat{U}_T, \mathrm{Im}\,U^*) = 0.33 \pm 0.08$. Therefore, additional iterations of HardTTh indeed help.

The fact that noise in singular values is larger than the estimation error is illustrated by the fact that PRLS performs worse than TT-HOSVD. Indeed, to remove noise in singular values, PRLS applies soft-thresholding with $\lambda_1, \lambda_2$ being around the noise level in singular values of matricizations. Then, soft-thresholded SVD has each singular value decreased by either $\lambda_1/2$ or $\lambda_2/2$. This yields the estimation error around the maximum of $\lambda_1$ and $\lambda_2$, which dramatically affects the algorithm performance. This highlights the difference between low-rank tensor estimation problem and low-rank matrix estimation problem, since for the latter there is no significant difference between soft-thresholding and hard-thresholding estimation.

We conduct experiments on image denoising task between mentioned tensor methods. The idea behind such comparison is the following: comparing covariance estimation through long pipelines is unfair, since other blocks might need additional tuning and it is hard to solve credit assignment between such changes. So we have decided to estimate the denoising abilities of our algorithm across one-shot methods (neural nets are out of scope, due to the training process in which they interact with tons of data). One can see results in Figure 1. We chosen $p, q, r$ as $(8, 4, 4)$ to match the dimension 256 of a given picture. Then we apply gaussian noise to the picture and pass it as sample covariance to the denoising algorithms. We search best hyperparameters to minimize the error and obtain $J, K = 32, 32$.

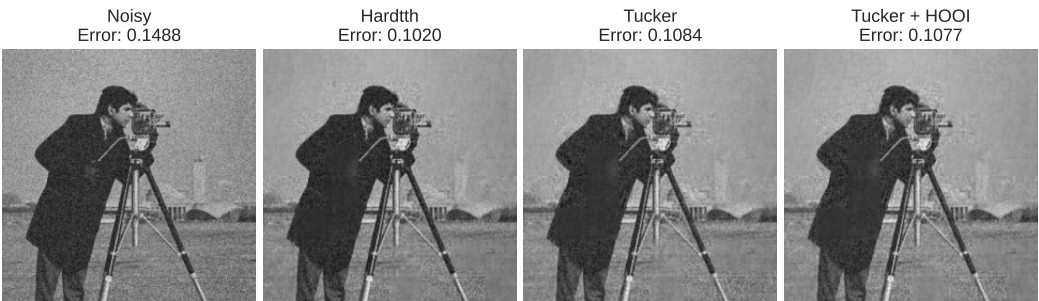

Figure 1: Performance of tensor decomposition algorithms on image denoising task.

## 4 CONCLUSION

In the present paper, we suggest a computationally efficient algorithm for estimation of high-dimensional covariance matrix based on HOOI algorithm of De Lathauwer et al. (2000). We provide a comprehensive theoretical analysis of this algorithm, establishing sufficient conditions for its application and rigorous guarantees that take into account both bias and variance of the proposed estimator. Our analysis is non-asymptotic and relies on the intrinsic dimensions of the covariance matrix associated to our algorithm, without involving the ambient dimension. We illustrate our theory with numerical experiments.

## 5 REPRODUCIBILITY STATEMENT

We provide the code in Supplementary Material. We give a proof sketch of Theorem 2.2 in Appendix D. The proof of Theorem 2.2 is given in Appendix E.

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

## A USAGE OF LLM

We used DeepSeek to polish and aid writing. All mathematical derivations and numerical experiments were performed solely by the authors.

## B ADDITIONAL NOTATIONS AND BASIC TOOLS

For proofs, we need some extra notation. First, we adapt the Einstein notation for tensors, omitting the summation symbol and assuming that the summation holds across repeated indices, e.g. for the matrix product

$$(AB)_{ab} = \sum_c A_{ac}B_{cb},$$

we will write as

$$(AB)_{ab} = A_{ac}B_{cb}.$$

Second, we will widely use the following identities for a tensor $\mathcal{T} \in \mathbb{R}^{d_1 \times d_2 \times d_3}$ and a matrix $X$ of suitable shape

$$
\begin{aligned}
\mathtt{m}_1(\mathcal{T} \times_3 X) &= \mathtt{m}_1(\mathcal{T})(I_{d_2} \otimes X^\top), \\
\mathtt{m}_1(\mathcal{T} \times_1 X) &= X \cdot \mathtt{m}_1(\mathcal{T}), \\
\mathtt{m}_3(\mathcal{T}X \times_1 X) &= \mathtt{m}_3(\mathcal{T})(X^\top \otimes I_{d_2}), \\
\mathtt{m}_3(\mathcal{T} \times_3 X) &= X \cdot \mathtt{m}_3(\mathcal{T}).
\end{aligned}
\tag{14}
$$

While the second and the fourth identities are straightforward, the first and the last one should be verified. Let us prove the first identity for $X \in \mathbb{R}^{d' \times d_3}$. Choosing indices $a \in [d_1], b \in [d_2], c \in [d']$, we obtain

$$
\begin{aligned}
(\mathtt{m}_1(\mathcal{T} \times_3 X))_{a,(b-1)\cdot d_3+c} &= (\mathcal{T} \times_3 X)_{abc} = X_{cc'}\mathcal{T}_{abc'} \\
&= \mathtt{m}_1(\mathcal{T})_{a,(b'-1)d_3+c'}(I_{d_2} \otimes X^\top)_{(b'-1)d_3+c',(b-1)d_3+c}.
\end{aligned}
$$

The third idenitty of (14) can be checked analogously.

For a matrix $U \in \mathbb{O}_{d,r}$, we denote the projector $UU^\top$ on $\operatorname{Im} U$ by $\Pi_U$.

## C PROOF OF PROPOSITION 2.3

*Proof.* The proposition follows from the following bound on the partial trace. Let $\Psi_g : L_1 \to L_1, \Phi_g : L_2 \to L_2, g = 1, \ldots, G$, be positive semidefinite operators. Define

$$H = \sum_{g=1}^G \Psi_g \otimes \Phi_g.$$

Then, we have

$$
\begin{aligned}
\|\operatorname{Tr}_{L_1}(H)\| = \|\sum_{g=1}^G \operatorname{Tr}(\Psi_g)\Phi_g\| &\leqslant \sum_{g=1}^G \frac{\operatorname{Tr}(\Psi_g)}{\|\Psi_g\|}\|\Psi_g\|\|\Phi_g\| \leqslant \max_g \mathtt{r}(\Psi_g) \sum_{g=1}^G \|\Psi_g\|\|\Phi_g\| \\
&\leqslant G \cdot \max_g \mathtt{r}(\Psi_g) \cdot \max_g \|\Psi_g\|\|\Phi_g\| \leqslant G \cdot \max_g \mathtt{r}(\Psi_g) \cdot \|H\|.
\end{aligned}
$$

The result follows by applying the above to each partial trace $\operatorname{Tr}_S(\Sigma), S \subset [3]$, with a proper choice of $L_1, \Psi_g$ and $\Phi_g$. $\qquad\square$

## D PROOF SKETCH FOR THEOREM 2.2

In this section, we provide the sketch of the proof of Theorem 2.2. The proof develops the ideas of Zhang and Xia (2018) and Puchkin and Rakhuba (2024). First, we consider the problem of

estimating a tensor $\mathcal{T}^* = \mathcal{W}^* \times_3 V^* \times_1 U^*$ from a noisy observations $\mathcal{Y} = \mathcal{T}^* + \mathcal{E}$, without any assumptions on the error term $\mathcal{E}$. Let $\widehat{\mathcal{T}}$ be the estimator obtained by Algorithm 1 on the input $\mathcal{Y}$. The noise $\mathcal{E}$ influence the estimation of $\widehat{\mathcal{T}}$ in several ways. First, one need to impose some assumptions depending on the norms of $\mathtt{m}_1(\mathcal{E})$ and $\mathtt{m}_3(\mathcal{E} \times_1 \widehat{U}_0)$ on the singular values of matricizations $\mathtt{m}_1(\mathcal{T}^*), \mathtt{m}_3(\mathcal{T}^*)$ to be able to recover left singular subspaces of these matricizations up to a $\sin \Theta$-error at most $1/4$. Second, we show by induction on $t = 1, \ldots, T$ that $\operatorname{Im} \widehat{U}_t, \operatorname{Im} \widehat{V}_t$ improves the estimation of singular subspaces and establish the dependence of the estimation error on $\mathcal{E}$ at step $T$. Finally, we decompose the error $\|\widehat{\mathcal{T}} - \mathcal{T}^*\|_{\mathrm{F}}$ into terms depending on the singular subspaces estimation and the error of estimating $\mathcal{W}^*$. Combining all types of errors, we obtain the following theorem. Its proof if postponed to Section F.

**Theorem D.1.** *Given model* (16)*, suppose that singular values* $\sigma_J(\mathtt{m}_1(\mathcal{T}^*)), \sigma_K(\mathtt{m}_3(\mathcal{T}^*))$ *satisfy*

$$\sigma_J(\mathtt{m}_1(\mathcal{T}^*)) \geqslant 24 \|\mathtt{m}_1(\mathcal{E})\| \quad and \quad \sigma_K(\mathtt{m}_3(\mathcal{T}^*)) \geqslant 24 \sup_{\substack{U \in \mathbb{R}^{d_1 \times J} \\ \|U\| \leqslant 1}} \|\mathtt{m}_3(\mathcal{E})(U \otimes I_{d_2})\|. \tag{15}$$

*Put*

$$\alpha_U = \|\mathtt{m}_1(\mathcal{E} \times_3 (V^*)^\top)\|, \qquad\qquad \beta_U = \sup_{\substack{V \in \mathbb{R}^{d_2 \times K} \\ \|V\| \leqslant 1}} \|\mathtt{m}_1(\mathcal{E} \times_3 V^\top)\|,$$

$$\alpha_V = \|\mathtt{m}_3(\mathcal{E} \times_1 (U^*)^\top)\|, \qquad\qquad \beta_V = \sup_{\substack{U \in \mathbb{R}^{d_1 \times J} \\ \|U\| \leqslant 1}} \|\mathtt{m}_3(\mathcal{E} \times_1 U^\top)\|.$$

*Then, we have*

$$\|\widehat{\mathcal{T}} - \mathcal{T}^*\|_{\mathrm{F}} \leqslant \sup_{U \in \mathbb{O}_{d_1, J}, V \in \mathbb{O}_{d_2, K}} \|\mathcal{E} \times_3 V^\top \times_1 U^\top\|_{\mathrm{F}} + 4\sqrt{K}\alpha_V + 4\sqrt{J}\alpha_U + \diamondsuit_2 + r_T,$$

*where*

$$\diamondsuit_2 = 48 \cdot \left( \frac{\sqrt{K}\beta_V \alpha_U}{\sigma_J(\mathtt{m}_1(\mathcal{T}^*))} + \frac{\sqrt{J}\beta_U \alpha_V}{\sigma_K(\mathtt{m}_3(\mathcal{T}^*))} \right),$$

$$r_T = 3(\sqrt{J} + \sqrt{K}) \cdot \left( \frac{64\beta_V \beta_U}{\sigma_J(\mathtt{m}_1(\mathcal{T}^*))\sigma_K(\mathtt{m}_3(\mathcal{T}^*))} \right)^T \|\mathtt{m}_1(\mathcal{E})\|.$$

Then, we decompose the error $\mathcal{E}$ into the bias part $\overline{\mathcal{E}}$ and the varaince part $\widehat{\mathcal{E}}$. Using the triangle inequality, we bound each error term appearing in Theorem D.1 into the bias and variance parts, and bound the variance parts with high probability using the variational PAC–Bayes approach (see (Catoni and Giulini, 2017; Zhivotovskiy, 2024; Abdalla and Zhivotovskiy, 2022; Puchkin and Rakhuba, 2024) for other applications of this technique).

## E  PROOF OF THEOREM 2.2

*Proof of Theorem 2.2.* For clarity, we divide the proof into several steps. For brevity, we denote $\mathcal{R}(\mathtt{m}_i(\cdot)), i = 1, 3$, by $\mathcal{R}_i(\cdot)$.

**Step 1. Sensititivty analysis of Algorithm 1.** First, we establish deterministic bounds on the reconstruction of the tensor $\mathcal{T}^*$ from a noisy observation $\mathcal{Y}$ by Algorithm 1, denoting

$$\mathcal{Y} = \mathcal{T}^* + \mathcal{E}, \tag{16}$$

where $\mathcal{T}^* = \mathcal{W}^* \times_3 V^* \times_1 U^*$ is the best $(J, K)$-TT-rank approximation of $\mathcal{R}(\Sigma)$, $U^* \in \mathbb{O}_{d_1, J}$, $V^* \in \mathbb{O}_{d_3, K}$, $\mathcal{W}^* \in \mathbb{R}^{J \times d_2 \times K}$, and $\mathcal{Y} = \mathcal{R}(\widehat{\Sigma})$. Let $\widehat{\mathcal{T}}$ be the output of Algorithm 1 with input $\mathcal{Y}$. Then, Theorem D.1 is applicable. But we need first to check its conditions.

**Step 2. Checking conditions of Theorem D.1.** We deduce Theorem 2.2 from Theorem D.1. Let us start with conditions of Theorem D.1, and bound right-hand sides of inequalities (15) from above. Consider the lower bound on $\sigma_J(\mathtt{m}_1(\mathcal{T}^*))$. By the triangle inequality, we have

$$\|\mathtt{m}_1(\mathcal{E})\| \leqslant \|\mathtt{m}_1(\overline{\mathcal{E}})\| + \|\mathtt{m}_1(\widehat{\mathcal{E}})\|.$$

The second term of the above can be upper bounded using the following lemma.

**Lemma E.1.** *Fix $\delta \in (0, 1)$. Suppose that $n \geqslant \mathtt{r}_1^2(\Sigma) + \mathtt{r}_2^2(\Sigma)\mathtt{r}_3^2(\Sigma) + \log(4/\delta)$. Then, under Assumption 2.1, we have*

$$\|\mathtt{m}_1(\widehat{\mathcal{E}})\| \leqslant 32\omega\|\Sigma\|\sqrt{\frac{\mathtt{r}_1^2(\Sigma) + \mathtt{r}_2^2(\Sigma)\mathtt{r}_3^2(\Sigma) + \log(1/\delta)}{n}}$$

*with probability at least $1 - \delta$.*

Define the event

$$\boldsymbol{\mathcal{E}}_1 = \left\{ \|\mathtt{m}_1(\widehat{\mathcal{E}})\| \leqslant 32\omega\|\Sigma\|\sqrt{\frac{\mathtt{r}_1^2(\Sigma) + \mathtt{r}_2^2(\Sigma)\mathtt{r}_3^2(\Sigma) + \log(6/\delta)}{n}} \right\}. \tag{17}$$

Since $n \geqslant \mathtt{R}_\delta \geqslant \mathtt{r}_1^2(\Sigma) + \mathtt{r}_2^2(\Sigma)\mathtt{r}_3^2(\Sigma) + \log(24/\delta)$, due to Lemma E.1, we have $\Pr(\boldsymbol{\mathcal{E}}_1) \geqslant 1 - \delta/6$. Hence, if

$$\sigma_J(\mathtt{m}_1(\mathcal{T}^*)) \geqslant 24\|\mathtt{m}_1(\overline{\mathcal{E}})\| + 768\omega\|\Sigma\|\sqrt{\frac{\mathtt{r}_1^2(\Sigma) + \mathtt{r}_2^2(\Sigma)\mathtt{r}_3^2(\Sigma) + \log(6/\delta)}{n}},$$

the first inequality of (15) is fulfilled on the event $\boldsymbol{\mathcal{E}}_1$. Since $\sigma_J(\mathtt{m}_1(\mathcal{T}^*)) \geqslant \sigma_J(\mathcal{R}_1(\Sigma)) - \|\mathtt{m}_1(\overline{\mathcal{E}})\|$, on $\boldsymbol{\mathcal{E}}_1$, to fulfill the first inequality of (15), it is enough to ensure that

$$\sigma_J(\mathcal{R}_1(\Sigma)) \geqslant 25\|\mathtt{m}_1(\overline{\mathcal{E}})\| + 768\omega\|\Sigma\|\sqrt{\frac{\mathtt{r}_1^2(\Sigma) + \mathtt{r}_2^2(\Sigma)\mathtt{r}_3^2(\Sigma) + \log(6/\delta)}{n}},$$

as guaranteed by the conditions of the theorem.

To satisfy the second inequality of (15), we use the triangle inequality again and obtain

$$\sup_{\substack{U \in \mathbb{R}^{d_1 \times J} \\ \|U\| \leqslant 1}} \|\mathtt{m}_3(\mathcal{E})(U \otimes I_{d_2})\| \leqslant \sup_{\substack{U \in \mathbb{R}^{d_1 \times J} \\ \|U\| \leqslant 1}} \|\mathtt{m}_3(\overline{\mathcal{E}})(U \otimes I_{d_2})\| + \sup_{\substack{U \in \mathbb{R}^{d_1 \times J} \\ \|U\| \leqslant 1}} \|\mathtt{m}_3(\widehat{\mathcal{E}})(U \otimes I_{d_2})\|.$$

We bound the second term, using the following lemma. Its proof is given in Section E.2.

**Lemma E.2.** *Fix $\delta \in (0, 1)$. Suppose that $n \geqslant J\mathtt{r}_1^2(\Sigma) + J\mathtt{r}_2^2(\Sigma) + \mathtt{r}_3^2(\Sigma) + \log(8/\delta)$. Then, with probability at least $1 - \delta$, we have*

$$\sup_{\substack{U \in \mathbb{R}^{d_1 \times J} \\ \|U\| \leqslant 1}} \|\mathtt{m}_3(\widehat{\mathcal{E}})(U \otimes I_{d_2})\| \leqslant 32\omega\|\Sigma\|\sqrt{\frac{J\mathtt{r}_1^2(\Sigma) + J\mathtt{r}_2^2(\Sigma) + \mathtt{r}_3^2(\Sigma) + \log(8/\delta)}{n}}.$$

*Analogously, if $n \geqslant \mathtt{r}_1^2(\Sigma) + K\mathtt{r}_2^2(\Sigma) + K\mathtt{r}_3^2(\Sigma) + \log(8/\delta)$, then, with probability at least $1 - \delta$, it holds that*

$$\sup_{V \in \mathbb{R}^{d_3 \times K}, \|V\| \leqslant 1} \|\mathtt{m}_1(\mathcal{E})(I_{d_2} \otimes V)\| \leqslant 32\omega\|\Sigma\|\sqrt{\frac{\mathtt{r}_1^2(\Sigma) + K\mathtt{r}_2^2(\Sigma) + K\mathtt{r}_3^2(\Sigma) + \log(8/\delta)}{n}}.$$

Define the event

$$\boldsymbol{\mathcal{E}}_2 = \left\{ \sup_{\substack{U \in \mathbb{R}^{d_1 \times J} \\ \|U\| \leqslant 1}} \|\mathtt{m}_3(\widehat{\mathcal{E}})(U \otimes I_{d_2})\| \leqslant 32\omega\|\Sigma\|\sqrt{\frac{\mathtt{r}_3^2(\Sigma) + J\mathtt{r}_1^2(\Sigma) + J\mathtt{r}_2^2(\Sigma) + \log(48/\delta)}{n}} \right\}.$$

It has probability $\Pr(\boldsymbol{\mathcal{E}}_2) \geqslant 1 - \delta/6$, since $n \geqslant \mathtt{R}_\delta$ satisfies conditions of Lemma E.2 with $\delta/6$ in place of $\delta$. Due to conditions of the theorem, we have

$$\sigma_K(\mathcal{R}_3(\Sigma)) \geqslant 25\|\mathtt{m}_3(\overline{\mathcal{E}})\| + 768\omega\|\Sigma\|\sqrt{\frac{\mathtt{r}_3^2(\Sigma) + J\mathtt{r}_1^2(\Sigma) + J\mathtt{r}_2^2(\Sigma) + \log(48/\delta)}{n}},$$

so conditions of Theorem D.1 is satisfied on $\boldsymbol{\mathcal{E}}_1 \cap \boldsymbol{\mathcal{E}}_2$.

**Step 3. Bounding $\alpha_U, \alpha_V, \beta_U, \beta_V$.** Then, we bound $\alpha_U, \alpha_V, \beta_U, \beta_V$. We start by the former two quantities. By the triangle inequality, we have

$$\alpha_U \leqslant \|\mathtt{m}_1(\overline{\mathcal{E}} \times_3 (V^*)^\top)\| + \|\mathtt{m}_1(\widehat{\mathcal{E}} \times_3 (V^*)^\top)\|,$$

$$\alpha_V \leqslant \|\mathtt{m}_3(\overline{\mathcal{E}} \times_1 (U^*)^\top)\| + \|\mathtt{m}_3(\widehat{\mathcal{E}} \times_3 (U^*)^\top)\|.$$

To bound the second terms of the right-hand sides of the above, we use the following lemma. Its proof is given in Section E.3.

**Lemma E.3.** *Fix $\delta \in (0,1)$. Suppose that $n \geqslant \mathrm{r}_1^2(\Sigma) + K\mathrm{r}_2^2(\Sigma) + \log(8/\delta)$. Then, with probability at least $1 - \delta$, we have*

$$\|\mathrm{m}_1(\widehat{\mathcal{E}} \times_3 (V^*)^\top)\| \leqslant 32\omega\|\Sigma\|\sqrt{\frac{\mathrm{r}_1^2(\Sigma) + K\mathrm{r}_2^2(\Sigma) + \log(8/\delta)}{n}}.$$

*Analogously, if $n \geqslant \mathrm{r}_3^2(\Sigma) + J\mathrm{r}_2^2(\Sigma) + \log(8/\delta)$, then, with probability at least $1 - \delta$, we have*

$$\|\mathrm{m}_3(\widehat{\mathcal{E}} \times_3 (U^*)^\top)\| \leqslant 32\omega\|\Sigma\|\sqrt{\frac{\mathrm{r}_3^2(\Sigma) + J\mathrm{r}_2^2(\Sigma) + \log(8/\delta)}{n}}.$$

Define events

$$\mathcal{E}_3 = \left\{ \|\mathrm{m}_1(\widehat{\mathcal{E}} \times_3 (V^*)^\top)\| \lesssim \omega\|\Sigma\|\sqrt{\frac{\mathrm{r}_1^2(\Sigma) + K\mathrm{r}_2^2(\Sigma) + \log(6/\delta)}{n}} \right\},$$

$$\mathcal{E}_4 = \left\{ \|\mathrm{m}_3(\widehat{\mathcal{E}} \times_3 (U^*)^\top)\| \lesssim \omega\|\Sigma\|\sqrt{\frac{\mathrm{r}_3^2(\Sigma) + J\mathrm{r}_2^2(\Sigma) + \log(6/\delta)}{n}} \right\}.$$

Since $n \geqslant \mathrm{R}_\delta$ satisfies the conditions of Lemma E.3 with $\delta/6$ in place of $\delta$, the lemma and the union bound imply $\Pr(\mathcal{E}_3 \cap \mathcal{E}_4) \geqslant 1 - \delta/3$. On the event $\mathcal{E}_3 \cap \mathcal{E}_4$, we have

$$\alpha_U \leqslant \widetilde{\alpha}_U \quad \text{and} \quad \alpha_V \leqslant \widetilde{\alpha}_V,$$

where $\widetilde{\alpha}_U, \widetilde{\alpha}_V$ are defined in Table 1.

Next, we bound $\beta_U, \beta_V$. Applying the triangle inequality, we get

$$\beta_U \leqslant \sup_{\substack{V \in \mathbb{R}^{d_2 \times K} \\ \|V\| \leqslant 1}} \|\mathrm{m}_1(\overline{\mathcal{E}} \times_3 V^\top)\| + \sup_{\substack{V \in \mathbb{R}^{d_2 \times K} \\ \|V\| \leqslant 1}} \|\mathrm{m}_1(\widehat{\mathcal{E}} \times_3 V^\top)\|,$$

$$\beta_V \leqslant \sup_{\substack{U \in \mathbb{R}^{d_1 \times J} \\ \|U\| \leqslant 1}} \|\mathrm{m}_3(\overline{\mathcal{E}})(U \otimes I_{d_2})\| + \sup_{\substack{U \in \mathbb{R}^{d_1 \times J} \\ \|U\| \leqslant 1}} \|\mathrm{m}_3(\widehat{\mathcal{E}})(U \otimes I_{d_2})\|.$$

Note that on the event $\mathcal{E}_2$, we have $\beta_V \leqslant \widetilde{\beta}_V$, where $\widetilde{\beta}_V$ is defined in Table 1. To bound $\beta_U$, we use Lemma E.2 again. Define an event

$$\mathcal{E}_5 = \left\{ \sup_{\substack{V \in \mathbb{R}^{d_2 \times K} \\ \|V\| \leqslant 1}} \|\mathrm{m}_1(\widehat{\mathcal{E}} \times_3 V^\top)\| \leqslant 32\omega\|\Sigma\|\sqrt{\frac{\mathrm{r}_1^2(\Sigma) + K\mathrm{r}_2^2(\Sigma) + K\mathrm{r}_3^2(\Sigma) + \log(48/\delta)}{n}} \right\}.$$

Since $n \geqslant \mathrm{R}_\delta$ satisfies the conditions of the lemma with $\delta/6$ in place of $\delta$, we have $\Pr(\mathcal{E}_5) \geqslant 1 - \delta/6$, and on this event $\beta_U \leqslant \widetilde{\beta}_U$.

**Step 4. Bounding** $\sup_{U \in \mathbb{O}_{d_1,J}, V \in \mathbb{O}_{d_2,K}} \|\mathcal{E} \times_3 V^\top \times_1 U^\top\|_\mathrm{F}$. Using the triangle inequality again, we get

$$\sup_{U \in \mathbb{O}_{d_1,J}, V \in \mathbb{O}_{d_2,K}} \|\mathcal{E} \times_3 V^\top \times_1 U^\top\|_\mathrm{F} \leqslant \sup_{U \in \mathbb{O}_{d_1,J}, V \in \mathbb{O}_{d_2,K}} \|\overline{\mathcal{E}} \times_3 V^\top \times_1 U^\top\|_\mathrm{F}$$

$$+ \sup_{U \in \mathbb{O}_{d_1,J}, V \in \mathbb{O}_{d_2,K}} \|\widehat{\mathcal{E}} \times_3 V^\top \times_1 U^\top\|_\mathrm{F}.$$

We bound the second term of the right-hand side using the following lemma. Its proof is given in Section E.4.

**Lemma E.4.** *Fix $\delta \in (0,1)$. Suppose that $n \geqslant J\mathrm{r}_1^2(\Sigma) + JK\mathrm{r}_2^2(\Sigma) + K\mathrm{r}_3^2(\Sigma) + \log(8/\delta)$. Then, with probability at least $1 - \delta$, we have*

$$\sup_{U \in \mathbb{O}_{d_1,J}, V \in \mathbb{O}_{d_2,K}} \|\widehat{\mathcal{E}} \times_3 V^\top \times_1 U^\top\|_\mathrm{F} \leqslant 32\omega\|\Sigma\|\sqrt{\frac{J\mathrm{r}_1^2(\Sigma) + JK\mathrm{r}_2^2(\Sigma) + K\mathrm{r}_3^2(\Sigma) + \log(8/\delta)}{n}}.$$

Define the event

$$\boldsymbol{\mathcal{E}}_6 = \left\{ \sup_{U \in \mathbb{O}_{d_1, J}, V \in \mathbb{O}_{d_2, K}} \|\widehat{\mathcal{E}} \times_3 V^\top \times_1 U^\top\|_{\mathrm{F}} \right.$$

$$\left. \leqslant 32\|\Sigma\|\sqrt{\frac{J\mathbf{r}_1^2(\Sigma) + JK\mathbf{r}_2^2(\Sigma) + K\mathbf{r}_3^2(\Sigma) + \log(48/\delta)}{n}} \right\}.$$

Since $n \geqslant \mathtt{R}_\delta$ satisfies the conditions of Lemma E.4 with $\delta/6$ in place of $\delta$, it implies $\Pr(\boldsymbol{\mathcal{E}}_6) \geqslant 1 - \delta/6$.

**Step 5. Establishing bias and variance leading terms.** The event $\boldsymbol{\mathcal{E}}_0 = \bigcap_{i=1}^6 \boldsymbol{\mathcal{E}}_i$ has probability at least $1 - \delta$ due to the union bound. On the event $\boldsymbol{\mathcal{E}}_0$, conditions of Theorem D.1 are satisfied, so we have

$$\alpha_U \leqslant \widetilde{\alpha}_U, \quad \alpha_V \leqslant \widetilde{\alpha}_V, \quad \beta_U \leqslant \widetilde{\beta}_U, \quad \beta_V \leqslant \widetilde{\beta}_V$$

and

$$\sup_{U \in \mathbb{O}_{d_1, J}, V \in \mathbb{O}_{d_2, K}} \|\widehat{\mathcal{E}} \times_3 V^\top \times_1 U^\top\|_{\mathrm{F}} \leqslant 32\omega\|\Sigma\|\sqrt{\frac{J\mathbf{r}_1^2(\Sigma) + JK\mathbf{r}_2^2(\Sigma) + K\mathbf{r}_3^2(\Sigma) + \log(48/\delta)}{n}}.$$

The conclusion of Theorem D.1 yields

$$\|\widehat{\mathcal{T}} - \mathcal{T}^*\|_{\mathrm{F}} \leqslant \sup_{U \in \mathbb{O}_{d_1, J}, V \in \mathbb{O}_{d_2, K}} \|\overline{\mathcal{E}} \times_3 V^\top \times_1 U^\top\|_{\mathrm{F}}$$

$$+ \omega\|\Sigma\|\sqrt{\frac{J\mathbf{r}_1^2(\Sigma) + JK\mathbf{r}_2^2(\Sigma) + K\mathbf{r}_3^2(\Sigma) + \log(6/\delta)}{n}}$$

$$+ 4\sqrt{K}\widetilde{\alpha}_U + 4\sqrt{J}\widetilde{\alpha}_U + \Diamond_2 + r_T.$$

Substituting expressions for $\widetilde{\alpha}_U, \widetilde{\alpha}_V$ from Table 1, we obtain

$$\|\widehat{\mathcal{T}} - \mathcal{T}^*\|_{\mathrm{F}} \leqslant \sup_{U \in \mathbb{O}_{d_1, J}, V \in \mathbb{O}_{d_2, K}} \|\overline{\mathcal{E}} \times_3 V^\top \times_1 U^\top\|_{\mathrm{F}} + 4\sqrt{K}\|\mathtt{m}_1(\overline{\mathcal{E}} \times_3 (V^*)^\top)\|$$

$$+ 4\sqrt{J}\|\mathtt{m}_3(\overline{\mathcal{E}} \times_1 (U^*)^\top)\|$$

$$+ 32\omega\|\Sigma\|\sqrt{\frac{J\mathbf{r}_1^2(\Sigma) + JK\mathbf{r}_2^2(\Sigma) + K\mathbf{r}_3^2(\Sigma) + \log(48/\delta)}{n}}$$

$$+ 32\sqrt{J}\omega\|\Sigma\|\sqrt{\frac{\mathbf{r}_1^2(\Sigma) + K\mathbf{r}_2^2(\Sigma) + \log(48/\delta)}{n}}$$

$$+ 32\sqrt{K}\omega\|\Sigma\|\sqrt{\frac{\mathbf{r}_3^2(\Sigma) + J\mathbf{r}_2^2(\Sigma) + \log(48/\delta)}{n}} + \Diamond_2 + r_T.$$

Note that the fifth and sixth terms of the right-hand side are dominated by the fourth term. Using

$$\|\widetilde{\Sigma} - \Sigma\|_{\mathrm{F}} = \|\widehat{\mathcal{T}} - \mathcal{T}^* + \mathcal{T}^* - \mathcal{R}^{-1}(\Sigma)\|_{\mathrm{F}} \leqslant \|\widehat{\mathcal{T}} - \mathcal{T}^*\|_{\mathrm{F}} + \|\overline{\mathcal{E}}\|_{\mathrm{F}},$$

$$\sup_{U \in \mathbb{O}_{d_1, J}, V \in \mathbb{O}_{d_2, K}} \|\overline{\mathcal{E}} \times_3 V^\top \times_1 U^\top\|_{\mathrm{F}} \leqslant \sup_{U \in \mathbb{O}_{d_1, J}} \sup_{V \in \mathbb{O}_{d_3, K}} \|U^\top \mathtt{m}_1(\overline{\mathcal{E}})(I_{d_2} \otimes V)\|_{\mathrm{F}}$$

$$\leqslant \sqrt{J} \sup_{V \in \mathbb{O}_{d_3, K}} \|\mathtt{m}_1(\overline{\mathcal{E}})(I_{d_2} \otimes V)\| \leqslant \sqrt{J}\|\mathtt{m}_1(\overline{\mathcal{E}})\|,$$

$$\|\mathtt{m}_3(\overline{\mathcal{E}} \times_1 (U^*)^\top)\| \leqslant \|\mathtt{m}_3(\overline{\mathcal{E}})\|,$$

$$\|\mathtt{m}_1(\overline{\mathcal{E}} \times_3 (V^*)^\top)\| \leqslant \|\mathtt{m}_1(\overline{\mathcal{E}})\|,$$

we derive

$$\|\widetilde{\Sigma} - \Sigma\|_{\mathrm{F}} \leqslant \overline{\mathbf{b}} + 96\omega\|\Sigma\|\sqrt{\frac{J\mathbf{r}_1^2(\Sigma) + JK\mathbf{r}_2^2(\Sigma) + K\mathbf{r}_3^2(\Sigma) + \log(48/\delta)}{n}} + \Diamond_2 + r_T \quad (18)$$

on $\boldsymbol{\mathcal{E}}_0$.

**Step 6.** **Bounding the remainder terms.** Since $\Diamond_2, r_T$ depend on $1/\sigma_J(\mathtt{m}_1(\mathcal{T}^*))$ and $1/\sigma_K(\mathtt{m}_3(\mathcal{T}^*))$, we will bound singular values $\sigma_J(\mathtt{m}_1(\mathcal{T}^*)), \sigma_K(\mathtt{m}_3(\mathcal{T}^*))$ below using $\sigma_J(\mathcal{R}_1(\Sigma)), \sigma_K(\mathcal{R}_3(\Sigma))$. By the conditions of the theorem, we have $\sigma_J(\mathcal{R}_1(\Sigma)) \geqslant 25\|\mathtt{m}_1(\overline{\mathcal{E}})\|$ and $\sigma_K(\mathcal{R}_3(\Sigma)) \geqslant \|\mathtt{m}_3(\overline{\mathcal{E}})\|$, so, by the Weyl inequality, we deduce

$$\sigma_J(\mathtt{m}_1(\mathcal{T}^*)) \geqslant \sigma_J(\mathcal{R}_1(\Sigma)) - \|\mathtt{m}_1(\overline{\mathcal{E}})\| \geqslant \frac{24}{25} \cdot \sigma_J(\mathcal{R}_1(\Sigma)),$$

$$\sigma_K(\mathtt{m}_3(\mathcal{T}^*)) \geqslant \sigma_K(\mathcal{R}_3(\Sigma)) - \|\mathtt{m}_3(\overline{\mathcal{E}})\| \geqslant \frac{24}{25} \cdot \sigma_K(\mathcal{R}_3(\Sigma)).$$

On the event $\mathcal{E}_0$, it implies

$$\Diamond_2 = 48 \cdot \left( \frac{\sqrt{K}\beta_V\alpha_U}{\sigma_J(\mathtt{m}_1(\mathcal{T}^*))} + \frac{\sqrt{J}\beta_U\alpha_V}{\sigma_K(\mathtt{m}_3(\mathcal{T}^*))} \right)$$

$$\leqslant 50 \cdot \left( \frac{\sqrt{K}\widetilde{\beta}_V\widetilde{\alpha}_U}{\sigma_J(\mathcal{R}_1(\mathcal{T}^*))} + \frac{\sqrt{J}\widetilde{\beta}_U\widetilde{\alpha}_V}{\sigma_K(\mathcal{R}_3(\Sigma))} \right) = \widetilde{\Diamond}_2,$$

and

$$r_T = 3(\sqrt{J} + \sqrt{K}) \cdot \left( \frac{64\beta_V\beta_U}{\sigma_J(\mathtt{m}_1(\mathcal{T}^*))\sigma_K(\mathtt{m}_3(\mathcal{T}^*))} \right)^T \|\mathtt{m}_1(\mathcal{E})\|$$

$$\leqslant (\sqrt{J} + \sqrt{K}) \left( \frac{200\widetilde{\beta}_V\widetilde{\beta}_U}{\sigma_J(\mathcal{R}_1(\Sigma))\sigma_K(\mathcal{R}_3(\Sigma))} \right)^T \|\mathtt{m}_1(\mathcal{E})\|.$$

Using definition (17) of the event $\mathcal{E}_1$, $\mathcal{E}_0 \subset \mathcal{E}_1$, and the trinagle inequality $\|\mathtt{m}_1(\mathcal{E})\| \leqslant \|\mathtt{m}_1(\overline{\mathcal{E}})\| + \|\mathtt{m}_1(\widehat{\mathcal{E}})\|$, we obtain

$$r_T \leqslant \widetilde{r}_T,$$

where $\widetilde{r}_T$ is defined in Table 1. Substituting the above bounds on $\Diamond_2, r_T$ into (18) finishes the proof. $\qquad\square$

### E.1 PROOF OF LEMMA E.1

*Proof.* **Step 1. Reduction to the PAC-bayes inequality.** The analysis will be based the following lemma, which is known as the PAC-Bayes inequality (see, e.g., Catoni and Giulini (2017)).

**Lemma E.5.** *Let $\mathbf{X}, \mathbf{X}_1, \ldots, \mathbf{X}_n$ be i.i.d. random elements on a measurable space $\mathcal{X}$. Let $\Theta$ be a parameter space equipped with a measure $\mu$ (which is also referred to as prior). Let $f : \mathcal{X} \times \Theta \to \mathbb{R}$. Then, with probability at least $1 - \delta$, it holds that*

$$\mathbb{E}_{\boldsymbol{\theta} \sim \rho} \frac{1}{n} \sum_{i=1}^{n} f(\mathbf{X}_i, \boldsymbol{\theta}) \leqslant \mathbb{E}_{\boldsymbol{\theta} \sim \rho} \log \mathbb{E}_{\mathbf{X}} e^{f(\mathbf{X}, \boldsymbol{\theta})} + \frac{\mathcal{KL}(\rho, \mu) + \log(1/\delta)}{n}$$

*simultaneously for all $\rho \ll \mu$.*

Let us rewrite $\|\mathtt{m}_1(\widehat{\mathcal{E}})\|$ as the supremum of a certain empirical process. We have

$$\|\mathtt{m}_1(\widehat{\mathcal{E}})\| = \sup_{\mathbf{x} \in \mathbb{S}^{d_1-1}, \mathbf{y} \in \mathbb{S}^{d_2 d_3-1}} \mathbf{x}^\top \mathtt{m}_1(\widehat{\mathcal{E}})\mathbf{y} = \sup_{\mathbf{x} \in \mathbb{S}^{d_1-1}, \mathbf{y} \in \mathbb{S}^{d_2 d_3-1}} \langle \mathtt{m}_1(\widehat{\mathcal{E}}), \mathbf{x}\mathbf{y}^\top \rangle$$

$$= \sup_{\mathbf{x} \in \mathbb{S}^{d_1-1}, \mathbf{y} \in \mathbb{S}^{d_2 d_3-1}} \langle \widehat{\Sigma} - \Sigma, \mathcal{R}_1^{-1}(\mathbf{x}\mathbf{y}^\top) \rangle$$

$$= \sup_{\mathbf{x} \in \mathbb{S}^{d_1-1}, \mathbf{y} \in \mathbb{S}^{d_2 d_3-1}} \frac{1}{n} \sum_{i=1}^{n} \langle \mathbf{X}_i \mathbf{X}_i^\top, \mathcal{R}_1^{-1}(\mathbf{x}\mathbf{y}^\top) \rangle - \mathbb{E}\langle \mathbf{X}_i \mathbf{X}_i^\top, \mathcal{R}_1^{-1}(\mathbf{x}by^\top) \rangle$$

$$= \sup_{\mathbf{x} \in \mathbb{S}^{d_1-1}, \mathbf{y} \in \mathbb{S}^{d_2 d_3-1}} \frac{1}{n} \sum_{i=1}^{n} \mathbf{X}_i^\top \mathcal{R}_1^{-1}(\mathbf{x}\mathbf{y}^\top)\mathbf{X}_i - \mathbb{E}\mathbf{X}_i^\top \mathcal{R}_1^{-1}(\mathbf{x}\mathbf{y}^\top)\mathbf{X}_i.$$

Define the following functions:

$$f_i(\mathbf{x}, \mathbf{y}) = \lambda \left\{ \mathbf{X}_i^\top \mathcal{R}_1^{-1}(\mathbf{x}\mathbf{y}^\top)\mathbf{X}_i - \mathbb{E}\mathbf{X}_i^\top \mathcal{R}_1^{-1}(\mathbf{x}\mathbf{y}^\top)\mathbf{X}_i \right\},$$

$$f_{\mathbf{X}}(\mathbf{x}, \mathbf{y}) = \lambda \left\{ \mathbf{X}^\top \mathcal{R}_1^{-1}(\mathbf{x}\mathbf{y}^\top)\mathbf{X} - \mathbb{E}\mathbf{X}^\top \mathcal{R}_1^{-1}(\mathbf{x}\mathbf{y}^\top)\mathbf{X} \right\},$$

where the positive factor $\lambda$ to be chosen later. We will apply Lemma E.5 to the empirical process

$$\lambda \|\mathtt{m}_1(\widehat{\mathcal{E}})\| = \sup_{\mathbf{x}\in\mathbb{S}^{d_1-1}, \mathbf{y}\in\mathbb{S}^{d_2 d_3-1}} \frac{1}{n} \sum_{i=1}^n f_i(\mathbf{x}, \mathbf{y})$$

with $\mathbb{R}^{d_1} \otimes \mathbb{R}^{d_2 d_3}$ as the parameter space and the centered Gaussian distribution $\mathcal{N}(0, \sigma_1^2 I_{d_1}) \otimes \mathcal{N}(0, \sigma_2^2 I_{d_2 d_3})$ as the prior $\mu$, where $\sigma_1, \sigma_2$ will be defined in the sequel. Consider random vectors $\boldsymbol{\xi}, \boldsymbol{\eta}$ with mutual distribution $\rho_{\mathbf{x},\mathbf{y}}$ such that $\mathbb{E}\boldsymbol{\xi}\boldsymbol{\eta}^\top = \mathbf{x}\mathbf{y}^\top$. Since $f_i(\mathbf{x}, \mathbf{y}), f_{\mathbf{X}}(\mathbf{x}, \mathbf{y})$ are linear in $\mathbf{x}\mathbf{y}^\top$, we have $\mathbb{E}_{\rho_{\mathbf{x},\mathbf{y}}} f_i(\boldsymbol{\xi}, \boldsymbol{\eta}) = f_i(\mathbf{x}, \mathbf{y})$, so Lemma E.5 yields

$$\sup_{\substack{\mathbf{x}\in\mathbb{S}^{d_1-1} \\ \mathbf{y}\in\mathbb{S}^{d_2 d_3-1}}} \frac{1}{n} \sum_{i=1}^n f_i(\mathbf{x}, \mathbf{y}) \leqslant \sup_{\substack{\mathbf{x}\in\mathbb{S}^{d_1-1} \\ \mathbf{y}\in\mathbb{S}^{d_2 d_3-1}}} \left\{ \mathbb{E}_{\rho_{\mathbf{x},\mathbf{y}}} \log \mathbb{E}_{\mathbf{X}} \exp f_{\mathbf{X}}(\boldsymbol{\xi}, \boldsymbol{\eta}) \right.$$

$$\left. + \frac{\mathcal{KL}(\rho_{\mathbf{x},\mathbf{y}}, \mu) + \log(1/\delta)}{n} \right\} \tag{19}$$

with probability at least $1 - \delta$. Then, we construct $\rho_{\mathbf{x},\mathbf{y}}$ such that the right-hand side of the above inequality can be controlled efficiently.

**Step 2. Constructing $\rho_{\mathbf{x},\mathbf{y}}$.** Suppose for a while that $\rho_{\mathbf{x},\mathbf{y}}$-almost surely we have

$$\lambda \|\Sigma^{1/2} \mathcal{R}_1^{-1}(\boldsymbol{\xi}\boldsymbol{\eta}^\top)\Sigma^{1/2}\|_{\mathrm{F}} \leqslant 1/\omega. \tag{20}$$

Then, Assumption 2.1 implies

$$\mathbb{E}_{\rho_{\mathbf{x},\mathbf{y}}} \log \mathbb{E}_{\mathbf{X}} \exp f_{\mathbf{X}}(\boldsymbol{\xi}, \boldsymbol{\eta}) = \mathbb{E}_{\rho_{\mathbf{x},\mathbf{y}}} \log \mathbb{E}_{\mathbf{X}} \exp \left\{ \lambda \left( \mathbf{X}^\top \mathcal{R}_1^{-1}(\mathbf{x}\mathbf{y}^\top)\mathbf{X} - \mathbb{E}\mathbf{X}^\top \mathcal{R}_1^{-1}(\mathbf{x}\mathbf{y}^\top)\mathbf{X} \right) \right\}$$

$$\leqslant \lambda^2 \omega^2 \mathbb{E}_{\rho_{\mathbf{x},\mathbf{y}}} \|\Sigma^{1/2} \mathcal{R}_1^{-1}(\boldsymbol{\xi}\boldsymbol{\eta}^\top)\Sigma^{1/2}\|_{\mathrm{F}}^2. \tag{21}$$

So, to control the above and keep the left-hand side of (20) bounded, we do the following. Define independent random vectors $G_1 \sim \mathcal{N}(0, \sigma_1^2 I_{d_1}), G_2 \sim \mathcal{N}(0, \sigma_2^2 I_{d_2 d_3})$, and consider a function

$$g(\mathbf{x}', \mathbf{y}') = \|\Sigma^{1/2} \mathcal{R}_1^{-1}(\mathbf{x}'(\mathbf{y}')^\top)\Sigma^{1/2}\|_{\mathrm{F}}. \tag{22}$$

By the triangle inequality, we have

$$g(\mathbf{x} + G_1, \mathbf{y} + G_2) \leqslant g(\mathbf{x}, \mathbf{y}) + g(\mathbf{x}, G_2) + g(G_1, \mathbf{y}) + g(G_1, G_2),$$

so

$$g^2(\mathbf{x} + G_1, \mathbf{y} + G_2) \leqslant 4g^2(\mathbf{x}, \mathbf{y}) + 4g^2(\mathbf{x}, G_2) + 4g^2(G_1, \mathbf{y}) + 4g^2(G_1, G_2).$$

Then, the distribution $\rho_{\mathbf{x},\mathbf{y}}$ of the random vector $(\boldsymbol{\xi}, \boldsymbol{\eta})$ is equal to the distribution of $(\mathbf{x}+G_1, \mathbf{y}+G_2)$ subject to the condition

$$(G_1, G_2) \in \Upsilon = \left\{ g^2(a, b) \leqslant 4\mathbb{E}g^2(a, b) \mid (a, b) \in (\{\mathbf{x}, G_1\} \times \{\mathbf{y}, G_2\})\backslash\{(\mathbf{x}, \mathbf{y})\} \right\}.$$

Note that by the union bound and the Markov inequality, we have

$$\Pr\left( (G_1, G_2) \notin \Upsilon \right) \leqslant \sum_{(a,b)\in(\{\mathbf{x}, G_1\}\times\{\mathbf{y}, G_2\})\backslash\{(\mathbf{x},\mathbf{y})\}} \Pr\left( g^2(a, b) > 4\mathbb{E}g^2(a, b) \right)$$

$$\leqslant \sum_{(a,b)\in(\{\mathbf{x}, G_1\}\times\{\mathbf{y}, G_2\})\backslash\{(\mathbf{x},\mathbf{y})\}} \frac{1}{4} = \frac{3}{4}. \tag{23}$$

Let us check, that $\mathbb{E}_{\rho_{\mathbf{x},\mathbf{y}}} \boldsymbol{\xi}\boldsymbol{\eta}^\top = \mathbf{x}\mathbf{y}^\top$. Since the Gaussian distribution is centrally symmetric and the function $g$ does not change its value when multiplying any of its argument by $-1$, we have

$$(\boldsymbol{\xi}, \boldsymbol{\eta}) \overset{d}{=} (\mathbf{x} + \varepsilon_1(\boldsymbol{\xi} - \mathbf{x}), \mathbf{y} + \varepsilon_2(\boldsymbol{\eta} - \mathbf{y})), \tag{24}$$

where $\varepsilon_1, \varepsilon_2$ are i.i.d. Rademacher ramdom variables independent of $(\boldsymbol{\xi}, \boldsymbol{\eta})$. Then, we obtain

$$\mathbb{E}\boldsymbol{\xi}\boldsymbol{\eta}^\top = \mathbf{x}\mathbf{y}^\top + \mathbb{E}\varepsilon_1\mathbb{E}(\boldsymbol{\xi}-\mathbf{x})\mathbf{y}^\top + \mathbb{E}\varepsilon_2\mathbb{E}\mathbf{x}(\boldsymbol{\eta}-\mathbf{y})^\top + \mathbb{E}\varepsilon_1\mathbb{E}\varepsilon_2\mathbb{E}(\boldsymbol{\xi}-\mathbf{x})(\boldsymbol{\eta}-\mathbf{y})^\top = \mathbf{x}\mathbf{y}^\top.$$

Hence, to satisfy the assumption (20) and use (21), it is enough to bound expectations $\mathbb{E}g^2(a, b)$ for $(a, b) \in \{\mathbf{x}, G_1\} \times \{\mathbf{y}, G_2\}$.

**Step 3. Bounding expectations $\mathbb{E}g^2(\cdot, \cdot)$.** Let us start with $g^2(\mathbf{x}, \mathbf{y})$. From the definition (22), we have

$$g^2(\mathbf{x}, \mathbf{y}) = \|\Sigma^{1/2}\mathcal{R}_1^{-1}(\mathbf{x}\mathbf{y}^\top)\Sigma^{1/2}\|_F^2 = \mathrm{Tr}(\Sigma^{1/2}\mathcal{R}_1^{-1}(\mathbf{x}\mathbf{y}^\top)\Sigma\mathcal{R}_1^{-\top}(\mathbf{x}\mathbf{y}^\top)\Sigma^{1/2})$$
$$= \mathrm{Tr}(\Sigma\mathcal{R}_1^{-1}(\mathbf{x}\mathbf{y}^\top)\Sigma\mathcal{R}_1^{-\top}(\mathbf{x}\mathbf{y}^\top)) \tag{25}$$

Since $\mathrm{Tr}(AB) \leqslant \|A\|_F\|B\|_F$ for any matrices $A, B$, we have

$$g^2(\mathbf{x}, \mathbf{y}) \leqslant \|\Sigma\mathcal{R}_1^{-1}(\mathbf{x}\mathbf{y}^\top)\|_F\|\Sigma\mathcal{R}_1^{-\top}(\mathbf{x}\mathbf{y}^\top)\| \leqslant \|\Sigma\|^2\|\mathbf{x}\mathbf{y}^\top\|_F^2 = \|\Sigma\|,$$

where we used the fact that $\mathcal{R}_1^{-1}(\cdot)$ does not change the Frobenius norm and that $\|\mathbf{x}\mathbf{y}^\top\|_F = \|\mathbf{x}\|\|\mathbf{y}\| = 1$.

It will be convenient for future purposes to rewrite (25) in a slightly different form. We introduce the following tensors, that are reshapings of the matrix $\Sigma$ and vectors $\mathbf{x}, \mathbf{y}, G_1, G_2$:

$$\mathcal{S}_{p_1q_1r_1p_2q_2r_2} = \Sigma_{(p_1-1)qr+(q_1-1)r+r_1, (p_2-1)qr+(q_2-1)r+r_2},$$
$$\mathcal{G}^{(1)}_{p_2p_3} = (G_1)_{(p_2-1)\cdot p+p_3}, \quad \mathcal{G}^{(2)}_{q_2q_3r_2r_3} = (G_2)_{(q_2-1)qr^2+(q_3-1)r^2+(r_2-1)r+r_3},$$
$$\mathbf{x}_{p_2p_3} = \mathbf{x}_{(p_2-1)p+p_3}, \quad \mathbf{y}_{q_2q_3r_2r_3} = \mathbf{y}_{(q_2-1)qr^2+(q_3-1)r^2+(r_2-1)r+r_3}.$$

Following the Einstein notation, we obtain

$$g^2(\mathbf{x}, \mathbf{y}) = \mathrm{Tr}(\Sigma\mathcal{R}_1^{-1}(G_1\mathbf{y}^\top)\Sigma\mathcal{R}_1^{-\top}(\mathbf{x}\mathbf{y}^\top))$$
$$= \Sigma_{(p_1-1)qr+(r_1-1)r+r_1, (p_2-1)qr+(q_2-1)r+r_2}$$
$$\times (\mathbf{x}\mathbf{y})^\top_{(p_2-1)p+p_3, (q_2-1)qr^2+(q_3-1)r^2+(r_2-1)r+r_3}$$
$$\times \Sigma_{(p_3-1)qr+(q_3-1)r+r_3, (p_4-1)qr+(q_4-1)r+r_4}$$
$$\times (\mathbf{x}\mathbf{y})^\top_{(p_1-1)p+p_4, (q_1-1)qr^2+(q_4-1)r^2+(r_1-1)r+r_4}.$$
$$= \mathcal{S}_{p_1q_1r_1p_2q_2r_2}\mathbf{x}_{p_2p_3}\mathbf{y}_{q_2q_3r_2r_3}\mathcal{S}_{p_3q_3r_3p_4q_4r_4}\mathbf{x}_{p_1p_4}\mathbf{y}_{q_1q_4r_1r_4} \tag{26}$$

Note that the above holds for any $\mathbf{x} \in \mathbb{R}^{d_1}, \mathbf{y} \in \mathbb{R}^{d_2d_3}$.

Then, we bound $\mathbb{E}g^2(G_1, \mathbf{y})$. Following (26), we get

$$\mathbb{E}g^2(G_1, \mathbf{y}) = \mathbb{E}\mathcal{S}_{p_1q_1r_1p_2q_2r_2}\mathcal{G}^{(1)}_{p_2p_3}\mathbf{y}_{q_2q_3r_2r_3}\mathcal{S}_{p_3q_3r_3p_4q_4r_4}\mathcal{G}^{(1)}_{p_1p_4}\mathbf{y}_{q_1q_4r_1r_4}$$
$$= \sigma_1^2\delta_{p_2p_1}\delta_{p_3p_4}\mathcal{S}_{p_1q_1r_1p_2q_2r_2}\mathbf{y}_{q_2q_3r_2r_3}\mathcal{S}_{p_3q_3r_3p_4q_4r_4}\mathbf{y}_{q_1q_4r_1r_4}$$
$$= \sigma_1^2\mathcal{S}_{p_1q_1r_1p_1q_2r_2}\mathbf{y}_{q_2q_3r_2r_3}\mathcal{S}_{p_3q_3r_3p_3q_4r_4}\mathbf{y}_{q_1q_4r_1r_4}$$

where $\delta$ is the Kronecker delta symbol. The above can be rewritten as the following trace:

$$\mathbb{E}g^2(G_1, \mathbf{y}) = \sigma_1^2 \cdot \mathrm{Tr}(\mathrm{Tr}_1(\Sigma)Y\mathrm{Tr}_1(\Sigma)Y^\top),$$

where entries of the matrix $Y$ are defined by $Y_{(q_2-1)r+r_2, (q_3-1)r+r_3} = \mathbf{y}_{q_2q_3r_2r_3}$. Then, we have

$$\mathbb{E}g^2(G_1, \mathbf{y}) \leqslant \sigma_1^2\|\mathrm{Tr}_1(\Sigma)Y\|_F \cdot \|\mathrm{Tr}_1(\Sigma)Y^\top\|_F \leqslant \sigma_1^2\|\mathrm{Tr}_1(\Sigma)\|^2 \cdot \|Y\|_F^2 = \sigma_1^2\|\mathrm{Tr}_1(\Sigma)\|.$$

Next, we bound $\mathbb{E}g^2(\mathbf{x}, G_2)$. Using (26), we derive

$$\mathbb{E}g^2(\mathbf{x}, G_2) = \mathbb{E}\mathcal{S}_{p_1q_1r_1p_2q_2r_2}\mathbf{x}_{p_2p_3}\mathcal{G}^{(2)}_{q_2q_3r_2r_3}\mathcal{S}_{p_3q_3r_3p_4q_4r_4}\mathbf{x}_{p_1p_4}\mathcal{G}^{(2)}_{q_1q_4r_1r_4}$$
$$= \sigma_2^2\delta_{q_2q_1}\delta_{q_3q_4}\delta_{r_2r_1}\delta_{r_3r_4}\mathcal{S}_{p_1q_1r_1p_2q_2r_2}\mathbf{x}_{p_2p_3}\mathcal{S}_{p_3q_3r_3p_4q_4r_4}\mathbf{x}_{p_1p_4}$$
$$= \sigma_2^2 \cdot \mathrm{Tr}(\mathrm{Tr}_{2,3}(\Sigma)X\mathrm{Tr}_{2,3}(\Sigma)X^\top),$$

where entries of the matrix $X$ are defined by $X_{p_2, p_3} = \mathbf{x}_{p_2p_3}$. Then, we have

$$\mathbb{E}g^2(\mathbf{x}, G_2) \leqslant \sigma_2^2\|\mathrm{Tr}_{2,3}(\Sigma)X\|_F \cdot \|\mathrm{Tr}_{2,3}(\Sigma)X^\top\|_F \leqslant \sigma_2^2\|\mathrm{Tr}_{2,3}(\Sigma)\| \cdot \|X\|_F^2 = \sigma_2^2 \cdot \|\mathrm{Tr}_{2,3}(\Sigma)\|^2.$$

Finally, we bound $\mathbb{E}g^2(G_1, G_2)$. Using (26), we get

$$
\begin{aligned}
\mathbb{E}g^2(G_1, G_2) &= \mathbb{E}\mathcal{S}_{p_1q_1r_1p_2q_2r_2}\mathcal{G}^{(1)}_{p_2p_3}\mathcal{G}^{(2)}_{q_2q_3r_2r_3}\mathcal{S}_{p_3q_3r_3p_4q_4r_4}\mathcal{G}^{(1)}_{p_1p_4}\mathcal{G}^{(2)}_{q_1q_4r_1r_4} \\
&= \sigma_1^2\sigma_2^2\delta_{p_1p_2}\delta_{p_3p_4}\delta_{q_1q_2}\delta_{q_3q_4}\delta_{r_1r_2}\delta_{r_3r_4}\mathcal{S}_{p_1q_1r_1p_2q_2r_2}\mathcal{S}_{p_3q_3r_3p_4q_4r_4} \\
&= \sigma_1^2\sigma_2^2 \cdot \mathrm{Tr}^2(\Sigma).
\end{aligned}
$$

Hence, we have $\rho_{\mathbf{x},\mathbf{y}}$-almost surely:

$$
g(\boldsymbol{\xi}, \boldsymbol{\eta}) \leqslant 2\sqrt{\|\Sigma\|^2 + \sigma_1^2\|\mathrm{Tr}_1(\Sigma)\|^2 + \sigma_2^2\|\mathrm{Tr}_{2,3}(\Sigma)\|^2 + \sigma_1^2\sigma_2^2\mathrm{Tr}^2(\Sigma)}.
$$

Set $\sigma_1^2 = \mathbf{r}_1^{-2}(\Sigma)$ and $\sigma_2^2 = \mathbf{r}_2^{-2}(\Sigma)\mathbf{r}_3^{-2}(\Sigma)$. By the definition of $\mathbf{r}_i(\Sigma)$, for this choice of $\sigma_1, \sigma_2$, the function $g(\boldsymbol{\xi}, \boldsymbol{\eta})$ is bounded by $4\|\Sigma\|$ almost surely. Thus, using (20) and (21), we deduce that for any $\lambda$ satisfying

$$
\lambda \leqslant (4\omega\|\Sigma\|)^{-1},
$$

we have

$$
\mathbb{E}_{\rho_{\mathbf{x},\mathbf{y}}} \log \mathbb{E}_{\mathbf{X}} \exp f_{\mathbf{X}}(\boldsymbol{\xi}, \boldsymbol{\eta}) \leqslant \lambda^2\omega^2 \cdot \mathbb{E}_{\rho_{\mathbf{x},\mathbf{y}}}g^2(\boldsymbol{\xi}, \boldsymbol{\eta}) \leqslant 16\lambda^2\omega^2\|\Sigma\|^2. \tag{27}
$$

Due to (19), it remains to bound the Kullback-Leibler divergence $\mathcal{KL}(\rho_{\mathbf{x},\mathbf{y}}, \mu)$.

**Step 4. Bounding the Kullback-Leibler divergence.** The density of $\rho_{\mathbf{x},\mathbf{y}}$ is given by

$$
\begin{aligned}
\rho_{\mathbf{x},\mathbf{y}}(x, y) &= \frac{(2\pi)^{-(d_1+d_2d_3)/2}\sigma_1^{-d_1}\sigma_2^{-d_2d_3}}{\Pr((G_1, G_2 \in \Upsilon)} \exp\left\{-\frac{1}{2\sigma_1^2}\|x - \mathbf{x}\|^2 - \frac{1}{2\sigma_2^2}\|y - \mathbf{y}\|^2\right\} \\
&\quad \times \mathbb{1}\{(x - \mathbf{x}, y - \mathbf{y}) \in \Upsilon\}.
\end{aligned}
$$

The density of the prior $\mu$ is given by

$$
\mu(x, y) = \frac{(2\pi)^{-(d_1+d_2d_3)/2}}{\sigma_1^{d_1}\sigma_2^{d_2d_3}} \exp\left\{-\frac{1}{2\sigma_1^2}\|x\|^2 - \frac{1}{2\sigma_2^2}\|y\|^2\right\}.
$$

Then, the KL-divergence can be computed as follows:

$$
\begin{aligned}
\mathcal{KL}(\rho_{\mathbf{x},\mathbf{y}}, \mu) &= \int_{\mathbb{R}^{d_1 \times d_2d_3}} \rho_{\mathbf{x},\mathbf{y}}(x, y) \log \frac{\rho_{\mathbf{x},\mathbf{y}}(x, y)}{\mu(x, y)} dxdy \\
&= \log \frac{1}{\Pr((G_1, G_2) \in \Upsilon)} \\
&\quad + \int_{\mathbb{R}^{d_1 \times d_2d_3}} \rho_{\mathbf{x},\mathbf{y}}(x, y) \left\{-\frac{1}{2\sigma_1^2}(\|x - \mathbf{x}\|^2 - \|x\|^2) - \frac{1}{2\sigma_2^2}(\|y - \mathbf{y}\|^2 - \|y\|^2)\right\} dxdy.
\end{aligned}
$$

Due to (23), the first term is bounded by $\log 4$. Note that the second term is equal to:

$$
-\frac{\|\mathbf{x}\|^2}{2\sigma_1^2} + \frac{2}{2\sigma_1^2}\langle \mathbb{E}_{\rho_{\mathbf{x},\mathbf{y}}}\boldsymbol{\xi}, \mathbf{x}\rangle - \frac{\|\mathbf{y}\|^2}{2\sigma_2^2} + \frac{2}{2\sigma_2^2}\langle \mathbb{E}_{\rho_{\mathbf{x},\mathbf{y}}}\boldsymbol{\eta}, \mathbf{y}\rangle.
$$

Using (24), we get

$$
\begin{aligned}
\mathbb{E}_{\rho_{\mathbf{x},\mathbf{y}}}\boldsymbol{\xi} &= \mathbf{x} + \mathbb{E}\varepsilon_1\mathbb{E}(\boldsymbol{\xi} - \mathbf{x}) = \mathbf{x}, \\
\mathbb{E}_{\rho_{\mathbf{x},\mathbf{y}}}\boldsymbol{\eta} &= \mathbf{y} + \mathbb{E}\varepsilon_2\mathbb{E}(\boldsymbol{\eta} - \mathbf{y}) = \mathbf{y},
\end{aligned}
$$

so we have

$$
\mathcal{KL}(\rho_{\mathbf{x},\mathbf{y}}, \mu) \leqslant \log 4 + \frac{\|\mathbf{x}\|_2^2}{2\sigma_1^2} + \frac{\|\mathbf{y}\|_2^2}{2\sigma_2^2} = \log 4 + \mathbf{r}_1^2(\Sigma)/2 + \mathbf{r}_2^2(\Sigma)\mathbf{r}_3^2(\Sigma)/2.
$$

**Step 5. Final bound.** Substituting the above bound and bound (27) into (46) and using

$$
\|\mathtt{m}_1(\widehat{\mathcal{E}})\| = \frac{1}{\lambda} \sup_{\substack{\mathbf{x}\in\mathbb{S}^{d_1-1} \\ \mathbf{y}\in\mathbb{S}^{d_2d_3-1}}} \frac{1}{n}\sum_{i=1}^{n} f_i(\mathbf{x}, \mathbf{y}),
$$

we get

$$\|\mathtt{m}_1(\widehat{\mathcal{E}})\| \leqslant 16\lambda\omega^2\|\Sigma\|^2 + \frac{\mathtt{r}_1^2(\Sigma)/2 + \mathtt{r}_2^2(\Sigma)\mathtt{r}_3^2(\Sigma)/2 + \log(4/\delta)}{\lambda n}$$

for any positive $\lambda \leqslant (4\omega\|\Sigma\|)^{-1}$ with probability at least $1 - \delta$. Since $n \geqslant \mathtt{r}_1^2(\Sigma) + \mathtt{r}_2^2(\Sigma)\mathtt{r}_3^2(\Sigma) + \log(4/\delta)$, we choose

$$\lambda = (4\omega\|\Sigma\|)^{-1}\sqrt{\frac{\mathtt{r}_1^2(\Sigma)/2 + \mathtt{r}_2^2(\Sigma)\mathtt{r}_3^2(\Sigma)/2 + \log(4/\delta)}{n}},$$

and get

$$\|\mathtt{m}_1(\widehat{\mathcal{E}})\| \leqslant 8\omega\|\Sigma\|\sqrt{\frac{\mathtt{r}_1^2(\Sigma)/2 + \mathtt{r}_2^2(\Sigma)\mathtt{r}_3^2(\Sigma)/2 + \log(4/\delta)}{n}}$$

$$\leqslant 32\omega\|\Sigma\|\sqrt{\frac{\mathtt{r}_1^2(\Sigma) + \mathtt{r}_2^2(\Sigma)\mathtt{r}_3^2(\Sigma) + \log(1/\delta)}{n}}. \qquad \square$$

### E.2 PROOF OF LEMMA E.2

*Proof.* We deduce Lemma E.2 from the following theorem. Its proof is posteponed to Section G.

**Theorem E.6.** *Let $\mathbb{S}_1, \mathbb{S}_2, \mathbb{S}_3$ be sets of linear operators*

$$\mathbb{S}_i \subset \left\{ A_i : L_i \to \mathbb{R}^{d_i}, \text{ such that } \|A_i\| \leqslant 1 \right\}, i = 1, 3,$$

$$\mathbb{S}_2 \subset \left\{ A \in L_1 \otimes \mathbb{R}^{d_2} \otimes L_3 \text{ such that } \|A\|_{\mathrm{F}} \leqslant 1 \right\}.$$

*For brevity, put $L_2 = L_1 \otimes L_3$. Denote $\dim L_i$ as $l_i$. Then, we have*

$$\sup_{\substack{A_1 \in \mathbb{S}_1, \\ A_2 \in \mathbb{S}_2, A_3 \in \mathbb{S}_3}} \langle \widehat{\mathcal{E}} \times_3 A_3^\top \times_1 A_1^\top, A_2 \rangle \leqslant 2^7\omega\|\Sigma\|\sqrt{\frac{\sum_{i=1}^3 \min\{\mathtt{r}_i^2(\Sigma) \cdot l_i, \log|\mathbb{S}_i|\} + \log(8/\delta)}{n}}$$

*with probability at least $1 - \delta$, provided $n \geqslant \sum_{i=1}^3 \min\{\mathtt{r}_i^2(\Sigma) \cdot l_i, \log|\mathbb{S}_i|\} + \log(8/\delta)$. Here we assume that $\min\{\mathtt{r}_i(\Sigma) \cdot l_i, \log|\mathbb{S}_i|\} = \mathtt{r}_i(\Sigma) \cdot l_i$ if $\mathbb{S}_i$ is infinite.*

Note that

$$\sup_{\substack{U \in \mathbb{R}^{d_1 \times J} \\ \|U\| \leqslant 1}} \|\mathtt{m}_3(\widehat{\mathcal{E}})(U \otimes I_{d_2})\| = \sup_{\substack{U \in \mathbb{R}^{d_1 \times J} \\ \|U\| \leqslant 1}} \|\mathtt{m}_3(\widehat{\mathcal{E}} \times_1 U^\top)\|$$

$$= \sup_{\substack{\mathbf{x} \in \mathbb{R}^{d_3}, \mathbf{y} \in \mathbb{R}^{Jd_2}, U \in \mathbb{R}^{d_1 \times J} \\ \|\mathbf{x}\| \leqslant 1, \|\mathbf{y}\| \leqslant 1, \|U\| \leqslant 1}} \mathbf{x}^\top \mathtt{m}_3(\widehat{\mathcal{E}} \times_1 U^\top)\mathbf{y}.$$

can rewritten as the following supremum over scalar product:

$$\sup_{\substack{A_1 \in \mathbb{S}_1, \\ A_2 \in \mathbb{S}_2, A_3 \in \mathbb{S}_3}} \langle \widehat{\mathcal{E}} \times_3 A_3^\top \times_1 A_1^\top, A_2 \rangle,$$

where

$$\mathbb{S}_1 = \{A_1 : \mathbb{R}^J \to \mathbb{R}^{d_1} \mid \|A_1\| \leqslant 1\},$$

$$\mathbb{S}_2 = \{A_2 \in \mathbb{R}^{J \times d_2 \times 1} \mid \|A_2\|_{\mathrm{F}} \leqslant 1\},$$

$$\mathbb{S}_3 = \{A_3 : \mathbb{R} \to \mathbb{R}^{d_3} \mid \|A_3\| \leqslant 1\}.$$

Then, Theorem E.6 implies that for any $\delta \in (0, 1)$, with probability at least $1 - \delta$, we have

$$\sup_{\substack{U \in \mathbb{R}^{d_1 \times J} \\ \|U\| \leqslant 1}} \|\mathtt{m}_3(\widehat{\mathcal{E}})(U \otimes I_{d_2})\| \leqslant 2^7\omega\|\Sigma\|\sqrt{\frac{J\mathtt{r}_1^2(\Sigma) + J\mathtt{r}_2^2(\Sigma) + \mathtt{r}_3^2(\Sigma) + \log(8/\delta)}{n}},$$

if $n \geqslant J\mathtt{r}_1^2(\Sigma) + J\mathtt{r}_2^2(\Sigma) + \mathtt{r}_3^2(\Sigma) + \log(8/\delta)$.

Analogously, we have

$$\sup_{V \in \mathbb{R}^{d_3 \times K}, \|V\| \leqslant 1} \|\mathtt{m}_1(\mathcal{E})(I_{d_2} \otimes V)\| \leqslant 32\omega\|\Sigma\|\sqrt{\frac{\mathtt{r}_1^2(\Sigma) + K\mathtt{r}_2^2(\Sigma) + K\mathtt{r}_3^2(\Sigma) + \log(8/\delta)}{n}}$$

with probability at least $1 - \delta$, if $n \geqslant \mathtt{r}_1^2(\Sigma) + K\mathtt{r}_2^2(\Sigma) + K\mathtt{r}_3^2(\Sigma) + \log(8/\delta)$. This completes the proof. $\quad\square$

### E.3 PROOF OF LEMMA E.3

*Proof.* Note that the norm

$$\|\mathtt{m}_1(\widehat{\mathcal{E}} \times_3 (V^*)^\top)\| = \sup_{\substack{\mathbf{x} \in \mathbb{R}^{d_1}, \mathbf{y} \in \mathbb{R}^{Kd_2} \\ \|\mathbf{x}\| \leqslant 1, \|\mathbf{y}\| \leqslant 1}} \mathbf{x}^\top \mathtt{m}_1(\widehat{\mathcal{E}} \times_3 (V^*)^\top)\mathbf{y}$$

can be rewritten as the following supremum over scalar product:

$$\sup_{\substack{A_1 \in \mathbb{S}_1, \\ A_2 \in \mathbb{S}_2, A_3 \in \mathbb{S}_3}} \langle \widehat{\mathcal{E}} \times_3 A_3^\top \times_1 A_1^\top, A_2 \rangle,$$

where

$$\mathbb{S}_1 = \{A_1 : \mathbb{R} \to \mathbb{R}^{d_1} \mid \|A_1\| \leqslant 1\},$$
$$\mathbb{S}_2 = \{A_2 \in \mathbb{R}^{K \times d_2 \times 1} \mid \|A_2\|_{\mathrm{F}} \leqslant 1\},$$
$$\mathbb{S}_3 = \{V^*\}.$$

Hence, Theorem E.6 implies that for any $\delta \in (0, 1)$, with probability at least $1 - \delta$, we have

$$\|\mathtt{m}_1(\widehat{\mathcal{E}} \times_3 (V^*)^\top)\| \leqslant 32\omega\|\Sigma\|\sqrt{\frac{\mathtt{r}_1^2(\Sigma) + K\mathtt{r}_2^2(\Sigma) + \log(8/\delta)}{n}},$$

if $n \geqslant \mathtt{r}_1^2(\Sigma) + K\mathtt{r}_2^2(\Sigma) + \log(8/\delta)$. Analogously, we have

$$\|\mathtt{m}_3(\widehat{\mathcal{E}} \times_1 (U^*)^\top)\| \leqslant 32\omega\|\Sigma\|\sqrt{\frac{\mathtt{r}_3^2(\Sigma) + J\mathtt{r}_2^2(\Sigma) + \log(8/\delta)}{n}},$$

with probability at least $1 - \delta$, if $n \geqslant J\mathtt{r}_2^2(\Sigma) + \mathtt{r}_3^2(\Sigma) + \log(8/\delta)$. This completes the proof. $\quad\square$

### E.4 PROOF OF LEMMA E.4

*Proof.* Using the variational representation of the Frobenius norm, we observe that

$$\sup_{U \in \mathbb{O}_{d_1, J}, V \in \mathbb{O}_{d_2, K}} \|\widehat{\mathcal{E}} \times_3 V^\top \times_1 U^\top\|_{\mathrm{F}} = \sup_{\substack{U \in \mathbb{O}_{d_1, J}, V \in \mathbb{O}_{d_2, K} \\ W \in \mathbb{R}^{J \times d_2 \times K}, \|W\|_{\mathrm{F}} \leqslant 1}} \langle \widehat{\mathcal{E}} \times_3 V^\top \times_1 U^\top, W \rangle.$$

Then, we apply Theorem E.6 with $\mathbb{S}_1 = \mathbb{O}_{d_1, J}, \mathbb{S}_2 = \{W \in \mathbb{R}^{J \times d_2 \times K} : \|W\|_{\mathrm{F}} \leqslant 1\}, \mathbb{S}_3 = \mathbb{O}_{d_3, K}$ and get the desired result. $\quad\square$

## F PROOF OF THEOREM D.1

*Proof of Theorem D.1.* The proof follows that of Theorem 1 by Zhang and Xia (2018). For clarity, we divide it into several steps.

**Step 1. Reduction to spectral norm of random matrices.** We have

$$\begin{aligned}
\|\widehat{\mathcal{T}} - \mathcal{T}^*\|_{\mathrm{F}}^2 &= \|\widehat{\mathcal{W}} \times_3 \widehat{V} \times_1 \widehat{U} - \mathcal{W}^* \times_3 V^* \times_1 U^*\|_{\mathrm{F}}^2 \\
&= \|\widehat{\mathcal{W}} \times_3 \widehat{V} \times_1 \widehat{U} - \mathcal{W}^* \times_3 V^* \times_1 (\widehat{U}\widehat{U}^\top)U^*\|_{\mathrm{F}}^2 + \|\mathcal{W}^* \times_3 V^* \times_1 (I - \Pi_{\widehat{U}})U^*\|_{\mathrm{F}}^2 \\
&= \|\widehat{\mathcal{W}} \times_3 \widehat{V} - \mathcal{W}^* \times_3 V^* \times_1 (\widehat{U}^\top U^*)\|_{\mathrm{F}}^2 + \|\mathcal{W}^* \times_3 V^* \times_1 (I - \Pi_{\widehat{U}})U^*\|_{\mathrm{F}}^2 \\
&= \|\widehat{\mathcal{W}} - \mathcal{W}^* \times_3 (\widehat{V}^\top V^*) \times_1 (\widehat{U}^\top U^*)\|_{\mathrm{F}}^2 + \|\mathcal{W}^* \times_3 (I - \Pi_{\widehat{V}})V^* \times_1 (\widehat{U}^\top U^*)\|_{\mathrm{F}}^2 \\
&\quad + \|\mathcal{W}^* \times_3 V^* \times_1 (I - \Pi_{\widehat{U}})U^*\|_{\mathrm{F}}^2.
\end{aligned} \tag{28}$$

By the construction of $\widehat{\mathcal{W}}$, the first term is equal to

$$\|\mathcal{Y} \times_3 \widehat{V}^\top \times_1 \widehat{U}^\top - \mathcal{T}^* \times_3 \widehat{V}^\top \times_1 \widehat{U}^\top\|_{\mathrm{F}}^2 = \|\mathcal{E} \times_3 \widehat{V}^\top \times_1 \widehat{U}^\top\|_{\mathrm{F}}^2. \qquad (29)$$

We rewrite the second term as follows:

$$\|\mathcal{W}^* \times_3 (I - \Pi_{\widehat{V}}) V^* \times_1 (\widehat{U}^\top U^*)\|_{\mathrm{F}} = \|(I - \Pi_{\widehat{V}}) \mathtt{m}_3(\mathcal{T}^* \times_1 \widehat{U}^\top)\|_{\mathrm{F}}.$$

Due to (14), we have $\mathtt{m}_3(\mathcal{T}^* \times_1 \widehat{U}^\top) = \mathtt{m}_3(\mathcal{T}^*)(\widehat{U} \otimes I_{d_2})$, so $\mathtt{m}_3(\mathcal{T}^* \times_1 \widehat{U}^\top)$ has rank at most $K$ and

$$\begin{aligned}
\|(I - \Pi_{\widehat{V}}) \mathtt{m}_3(\mathcal{T}^*)(\widehat{U} \otimes I_{d_2})\|_{\mathrm{F}} &\leqslant \sqrt{K} \|(I - \Pi_{\widehat{V}}) \mathtt{m}_3(\mathcal{T}^*)(\widehat{U} \otimes I_{d_2})\| \\
&= \sqrt{K} \|(I - \Pi_{\widehat{V}}) \mathtt{m}_3(\mathcal{T}^* \times_1 \widehat{U}^\top)\| \\
&\leqslant \sqrt{K} \|(I - \Pi_{\widehat{V}}) \mathtt{m}_3(\mathcal{Y} \times \widehat{U}^\top)\| + \sqrt{K} \|(I - \Pi_{\widehat{V}}) \mathtt{m}_3(\mathcal{E} \times_1 \widehat{U}_1^\top)\|.
\end{aligned}$$

Since $\widehat{V}$ consists of $K$ leading left singular vectors of $\mathtt{m}_3(\mathcal{Y} \times_1 \widehat{U})$ and $\mathtt{m}_3(\mathcal{T}^* \times_1 \widehat{U}_1^\top)$ has rank $K$, we have $\|(I - \Pi_{\widehat{V}}) \mathtt{m}_3(\mathcal{Y} \times_1 \widehat{U}_1)\| = \sigma_{K+1}(\mathtt{m}_3(\mathcal{Y} \times_1 \widehat{U}_1)) \leqslant \|\mathtt{m}_3(\mathcal{E} \times \widehat{U}_1)\|$ by the Weyl inequality . It yields

$$\|\mathcal{W}^* \times_3 (I - \Pi_{\widehat{V}}) V^* \times_1 (\widehat{U}^\top U^*)\|_{\mathrm{F}} \leqslant 2\sqrt{K} \|\mathtt{m}_3(\mathcal{E} \times_1 \widehat{U}^\top)\|. \qquad (30)$$

Then, we bound the third term of (28). We have

$$\begin{aligned}
\|\mathcal{W}^* \times_3 V^* \times_1 (I - \Pi_{\widehat{U}}) U^*\|_{\mathrm{F}} &= \|\mathcal{W}^* \times_1 (I - \Pi_{\widehat{U}}) U^*\|_{\mathrm{F}} \\
&\leqslant \sigma_{\min}^{-1}(\widehat{V}_{T-1}^\top V^*) \|\mathcal{W}^*) \times_3 (\widehat{V}_{T-1}^\top V^*) \times_1 (I - \Pi_{\widehat{U}}) U^*\|_{\mathrm{F}} \\
&= \sigma_{\min}^{-1}(\widehat{V}_{T-1}^\top V^*) \|(I - \Pi_{\widehat{U}}) \mathtt{m}_1(\mathcal{T}^* \times_3 \widehat{V}_{T-1}^\top)\|_{\mathrm{F}}.
\end{aligned}$$

The matrix $\mathtt{m}_1(\mathcal{T}^* \times_3 \widehat{V}_{T-1}^\top) = \mathtt{m}_1(\mathcal{T}^*)(I_{d_2} \otimes \widehat{V}_{T-1})$ has rank at most $J$, so

$$\begin{aligned}
\|(I - \Pi_{\widehat{U}}) \mathtt{m}_1(\mathcal{T}^* \times_3 \widehat{V}_{T-1}^\top)\|_{\mathrm{F}} &\leqslant \sqrt{J} \|(I - \Pi_{\widehat{U}}) \mathtt{m}_1(\mathcal{T}^* \times_3 \widehat{V}_{T-1}^\top)\| \\
&\leqslant \sqrt{J} \|(I - \Pi_{\widehat{U}}) \mathtt{m}_1(\mathcal{Y} \times_3 \widehat{V}_{T-1}^\top)\| + \sqrt{J} \|(I - \Pi_{\widehat{U}}) \mathtt{m}_1(\mathcal{E} \times_3 \widehat{V}_{T-1}^\top)\|.
\end{aligned}$$

Since $\widehat{U}$ consists of $J$ leading left singular vectors of $\mathtt{m}_1(\mathcal{Y} \times_3 \widehat{V}_{T-1}^\top)$ and $\mathtt{m}_1(\mathcal{T}^* \times_3 \widehat{V}_{T-1}^\top)$ has the rank at most $J$, we have $\|(I - \Pi_{\widehat{U}}) \mathtt{m}_1(\mathcal{Y} \times_3 \widehat{V}_{T-1}^\top)\| = \sigma_{J+1}(\mathtt{m}_1(\widehat{\mathcal{Y}} \times_3 \widehat{V}_{T-1}^\top)) \leqslant \|\mathtt{m}_1(\mathcal{E}) \times_3 \widehat{V}_{T-1}^\top\|$ by the Weyl inequality. It implies

$$\|\mathcal{W}^* \times_3 V^* \times_1 (I - \Pi_{\widehat{U}}) U^*\|_{\mathrm{F}} \leqslant \frac{2\sqrt{J}}{\sigma_{\min}(\widehat{V}_{T-1}^\top V^*)} \|\mathtt{m}_1(\mathcal{E} \times_3 \widehat{V}_{T-1}^\top)\|.$$

Combining (28) with (29), (30) and the above display, we get

$$\begin{aligned}
\|\widehat{\mathcal{T}} - \mathcal{T}^*\|_{\mathrm{F}}^2 &\leqslant \|\mathcal{E} \times_3 \widehat{V}^\top \times_1 \widehat{U}^\top\|_{\mathrm{F}}^2 + 4K \|\mathtt{m}_3(\mathcal{E} \times_1 \widehat{U}^\top)\|^2 \\
&\quad + \frac{4J}{\sigma_{\min}^2(\widehat{V}_{T-1}^\top V^*)} \|\mathtt{m}_1(\mathcal{E} \times_3 \widehat{V}_{T-1}^\top)\| \\
&\leqslant \sup_{U \in \mathbb{O}_{d_1,J}, V \in \mathbb{O}_{d_2,K}} \|\mathcal{E} \times \times_3 V^\top \times_1 U^\top\|_{\mathrm{F}}^2 \\
&\quad + 4K \|\mathtt{m}_3(\mathcal{E} \times_1 \widehat{U}^\top)\|^2 + \frac{4J}{\sigma_{\min}^2(\widehat{V}_{T-1}^\top V^*)} \|\mathtt{m}_1(\mathcal{E} \times_3 \widehat{V}_{T-1}^\top)\|^2. \qquad (31)
\end{aligned}$$

**Step 2. Bounding $\sigma_{\min}(\widehat{V}_{T-1}^\top V^*)$, $\|\mathtt{m}_1(\mathcal{E} \times_3 \widehat{V}_{T-1}^\top)\|$, $\|\mathtt{m}_3(\mathcal{E} \times_1 \widehat{U}^\top)\|$.** To obtain the theorem, we need to bound $\sigma_{\min}(\widehat{V}_{T-1}^\top \times_3 \mathcal{E})$, $\|\mathtt{m}_1(\mathcal{E} \times_3 \widehat{V}_{T-1}^\top)\|$, $\|\mathtt{m}_3(\mathcal{E} \times_1 \widehat{U}^\top)\|$. We start with the latter two norms. We have

$$\|\mathtt{m}_3(\mathcal{E} \times_1 \widehat{U}^\top)\| = \|\mathtt{m}_3(\mathcal{E})(\widehat{U} \otimes I_{d_2})\| \leqslant \|\mathtt{m}_3(\mathcal{E})(\Pi_{U^*}\widehat{U} \otimes I_{d_2})\| + \|\mathtt{m}_3(\mathcal{E})((I - \Pi_{U^*})\widehat{U} \otimes I_{d_2})\|. \qquad (32)$$

Since $\Pi_{U*} = U^*(U^*)^\top$, the first term of the above is at most

$$\|\mathtt{m}_3(\mathcal{E})U^*((U^*)^\top\widehat{U} \otimes I_{d_2})\| = \|\mathtt{m}_3(\mathcal{E})(U^* \otimes I_{d_2})((U^*)^\top\widehat{U} \otimes I_{d_2})\|$$

$$\leqslant \|\mathtt{m}_3(\mathcal{E})(U^* \otimes I_{d_2})\|\|((U^*)^\top\widehat{U} \otimes I_{d_2})\|$$

$$\leqslant \|\mathtt{m}_3(\mathcal{E})(U^* \otimes I_{d_2})\|. \tag{33}$$

For the second term, we have

$$\|\mathtt{m}_3(\mathcal{E})((I - \Pi_{U*})\widehat{U} \otimes I_{d_2})\| \leqslant \|\mathtt{m}_3(\mathcal{E})(\frac{(I - \Pi_{U*})}{\|(I - \Pi_{U*})\widehat{U}\|} \otimes I_{d_2})\| \cdot \|(I - \Pi_{U*})\widehat{U}\|$$

$$\leqslant \sup_{\substack{V \in \mathbb{R}^{d_1 \times J}, \\ \|V\|=1}} \|\mathtt{m}_3(\mathcal{E})(V \otimes I_{d_2})\| \cdot \|(I - \Pi_{U*})\widehat{U}\|.$$

Then, we have

$$\|(I - \Pi_{U*})\widehat{U}\| = \|(I - \Pi_{U*})\Pi_{\widehat{U}}\| = \|(\Pi_{\widehat{U}} - \Pi_{U*})\Pi_{\widehat{U}}\| \leqslant \|\Pi_{\widehat{U}} - \Pi_{U*}\|,$$

where we used $\operatorname{Im}\widehat{U}^\top = \mathbb{R}^K$ and orthogonality of $\widehat{U}$ for the first equality. To bound the latter norm of the difference, we rely on the following standard proposition, which is proved

**Proposition F.1.** *For two orthogonal matrices $U_1, U_2 \in \mathbb{O}_{a,b}$, $a \geqslant b$, define the following semidistance*

$$\rho(U_1, U_2) = \inf_{O \in \mathbb{O}_{b,b}} \|U_1 - U_2 O\|.$$

*Then, we have*

$$\|\Pi_{U_1} - \Pi_{U_2}\| \leqslant 2 \cdot \rho(U_1, U_2).$$

The proposition implies

$$\|\mathtt{m}_3(\mathcal{E})((I - \Pi_{U*})\widehat{U} \otimes I_{d_2}\| \leqslant 2 \sup_{\substack{V \in \mathbb{R}^{d_1 \times J} \\ \|V\|=1}} \|\mathtt{m}_3(\mathcal{E})(V \otimes I_{d_2})\| \cdot \rho(\widehat{U}, U^*).$$

Combining the above with (32) and (33), we get

$$\|\mathtt{m}_3(\widehat{U} \times_1 \mathcal{E})\| \leqslant \|\mathtt{m}_3(\mathcal{E})(U^* \otimes I_{d_2})\| + 2 \sup_{\substack{V \in \mathbb{R}^{d_1 \times J} \\ \|V\|=1}} \|\mathtt{m}_3(\mathcal{E})(V \otimes I_{d_2})\| \cdot \rho(\widehat{U}, U^*). \tag{34}$$

Analogously, we have

$$\|\mathtt{m}_1(\widehat{V}_{T-1} \times_3 \mathcal{E})\| \leqslant \|\mathtt{m}_1(\mathcal{E})(I_{d_2} \otimes V^*)\| + 2 \sup_{\substack{V \in \mathbb{R}^{d_3 \times K} \\ \|V\|\leqslant 1}} \|\mathtt{m}_1(\mathcal{E})(I_{d_2} \otimes V)\| \cdot \rho(\widehat{V}_{T-1}, V^*). \tag{35}$$

Finally, we bound $\sigma_{\min}(\widehat{V}_{T-1}^\top V^*)$ below. We have

$$\sigma_{\min}^2(\widehat{V}_{T-1}^\top V^*) = \lambda_{\min}((V^*)^\top\widehat{V}\widehat{V}^\top V^*) = \lambda_K(\Pi_{V*}\Pi_{\widehat{V}_{T-1}}\Pi_{V*}),$$

where we used the fact that $V^*A(V^*)^\top$ has the same singular values as $A$ for any Hermitian $A \in \mathbb{R}^{K \times K}$. Since $\Pi_{V*}\Pi_{\widehat{V}}\Pi_{V*} = \Pi_{V*} - \Pi_{V*}(I - \Pi_{\widehat{V}_{T-1}})\Pi_{V*} = \Pi_{V*} - \Pi_{V*}(\Pi_{V*} - \Pi_{\widehat{V}_{T-1}})\Pi_{V*}$, the Weyl inequality implies

$$\lambda_K(\Pi_{V*}\Pi_{\widehat{V}_{T-1}}\Pi_{V*}) \geqslant \lambda_K(\Pi_{V*}) - \|\Pi_{V*}(\Pi_{V*} - \Pi_{\widehat{V}_{T-1}})\Pi_{V*}\| \geqslant 1 - \|\Pi_{\widehat{V}_{T-1}} - \Pi_{V*}\|.$$

Then, Proposition F.1 yields $\|\Pi_{\widehat{V}_{T-1}} - \Pi_{V*}\| \leqslant 2\rho(\widehat{V}_{T-1}, V^*)\}$, so

$$\sigma_{\min}(\widehat{V}_{T-1}^\top V^*) \geqslant \sqrt{1 - 2\rho(\widehat{V}_{T-1}, V^*)}, \tag{36}$$

provided $\rho(\widehat{V}_{T-1}, V^*) \leqslant 1/2$.

**Step 3. Bounding $\rho(\widehat{U}_t, U^*)$, $\rho(\widehat{V}_t, V^*)$ recursively.** We provide a recursive bound on $\rho(\widehat{U}_t, U^*)$ and $\rho(\widehat{V}_t, V^*)$. We widely use the following lemma, which is a weaker variant of the Wedin $\sin\Theta-$ theorem:

**Proposition F.2.** *Let $A, B$ be matrices, such that $A$ has rank $r$, and denote $B = A + E$. Let $L$ be left singular vectors of $A$ and $\widehat{L}$ be $r$ leading left singular vectors of $B$. Then*

$$\rho(L, \widehat{L}) \leqslant \frac{2\sqrt{2}\|E\|}{\sigma_r(A)}.$$

By Proposition F.2, we have

$$\rho(\widehat{U}_0, U^*) \leqslant \frac{2\sqrt{2}\|\mathtt{m}_1(\mathcal{E})\|}{\sigma_J(\mathtt{m}_1(\mathcal{T}^*))}. \tag{37}$$

To bound $\rho(\widehat{V}_t, V^*)$, we note the following. Since $\widehat{V}_t$ are leading $K$ left singular vectors of $\mathtt{m}_3(\mathcal{Y} \times_1 \widehat{U}_t^\top) = \mathtt{m}_3(\mathcal{T}^* \times_1 \widehat{U}_t^\top) + \mathtt{m}_3(\mathcal{E} \times_1 \widehat{U}_t^\top)$, and there exists an orthogonal matrix $O \in \mathbb{O}_{K,K}$ such that $V^*O$ are the left singular vectors of $\mathtt{m}_3(\mathcal{T}^* \times_1 \widehat{U}_t^\top) = V^*\mathtt{m}_3(\mathcal{W}^* \times_1 U^*)(\widehat{U}_t \otimes I_{d_2})$, by the definition of $\rho(\cdot, \cdot)$ and Proposition F.2, we have

$$\rho(\widehat{V}_0, V^*) \leqslant \frac{2\sqrt{2}\|\mathtt{m}_3(\mathcal{E} \times_1 \widehat{U}_0)\|}{\sigma_K(\mathtt{m}_3(\mathcal{T}^* \times \widehat{U}_0^\top))} \quad \text{and} \quad \rho(\widehat{V}_t, V^*) \leqslant \frac{2\sqrt{2}\|\mathtt{m}_3(\mathcal{E} \times_1 \widehat{U}_t)\|}{\sigma_K(\mathtt{m}_3(\mathcal{T}^* \times_1 \widehat{U}_t^\top))}$$

for $t = 1, \dots, T$. Let us bound $\rho(\widehat{V}_t, V^*)$ using $\rho(\widehat{U}_t, U^*)$. First, we have

$$\sigma_K(\mathtt{m}_3(\mathcal{T}^* \times_1 \widehat{U}_t^\top)) = \sigma_K(\mathtt{m}_3(\mathcal{T}^*)(\widehat{U}_t \otimes I_{d_2})) = \sigma_K(\mathtt{m}_3(\mathcal{T}^*)(U^* \otimes I_{d_2})((U^*)^\top \widehat{U} \otimes I_{d_2})) \tag{38}$$

$$\geqslant \sigma_K(\mathtt{m}_3(\mathcal{T}^*)(U^* \otimes I_{d_2}))\sigma_{\min}((U^*)^\top \widehat{U}_t) =$$

$$= \sigma_K(\mathtt{m}_3(\mathcal{T}^*)(\Pi_{U*} \otimes I_{d_2}))\sigma_{\min}((U^*)^\top \widehat{U}) \geqslant \sigma_K(\mathtt{m}_3(\mathcal{T}^*)) \cdot \sqrt{1 - 2\rho(\widehat{U}_t, U^*)},$$

provided $\rho(\widehat{U}_t, U^*) < 1/2$. Second, we bound $\|\mathtt{m}_3(\mathcal{E} \times_1 \widehat{U}_t^\top)\|$. Following the derivation of (34), we obtain

$$\|\mathtt{m}_3(\mathcal{E} \times_1 \widehat{U}_t^\top)\| = \|\mathtt{m}_3(\mathcal{E})(\widehat{U}_t \otimes I_{d_2})\|$$

$$\leqslant \|\mathtt{m}_3(\mathcal{E})(\Pi_{U*} \otimes I_{d_2})(\widehat{U}_t \otimes I_{d_2})\| + \|\mathtt{m}_3(\mathcal{E})((I - \Pi_{U*}) \otimes I_{d_1})(\widehat{U}_t \otimes I_{d_2})\|$$

$$\leqslant \|\mathtt{m}_3(\mathcal{E})(U^* \otimes I_{d_2})\| + \sup_{\substack{U \in \mathbb{R}^{d_1 \times J} \\ \|U\| \leqslant 1}} \|\mathtt{m}_3(\mathcal{E})(U \otimes I_{d_2})\| \cdot \|(I - \Pi_{U*})\widehat{U}_t\|.$$

Since $\widehat{U}_t$ is orthogonal, we have $\|(I - \Pi_{U*})\widehat{U}_t\| = \|(I - \Pi_{U*})\Pi_{\widehat{U}_t}\|$, so

$$\|(I - \Pi_{U*})\widehat{U}_t\| = \|(\Pi_{\widehat{U}_t} - \Pi_{U*})\Pi_{\widehat{U}_t}\| \leqslant \|\Pi_{\widehat{U}_t} - \Pi_{U*}\| \leqslant 2\rho(\widehat{U}_t, U^*),$$

due to Proposition F.1, and

$$\|\mathtt{m}_3(\mathcal{E} \times_1 \widehat{U}_t^\top)\| \leqslant \|\mathtt{m}_3(\mathcal{E})(U^* \otimes I_{d_2})\| + 2 \sup_{\substack{U \in \mathbb{R}^{d_1 \times J} \\ \|U\| \leqslant 1}} \|\mathtt{m}_3(\mathcal{E})(U \otimes I_{d_2})\| \cdot \rho(\widehat{U}_t, U^*). \tag{39}$$

Following the notation of the theorem, we get

$$\rho(\widehat{V}_t, V^*) \leqslant \frac{2\sqrt{2} \cdot \left(\alpha_V + 2\beta_V \cdot \rho(\widehat{U}_t, U^*)\right)}{\sigma_K(\mathtt{m}_3(\mathcal{T}^*))\sqrt{1 - 2\rho(\widehat{U}_t, U^*)}}. \tag{40}$$

Next, we will bound $\rho(\widehat{U}_t, U^*)$ using $\rho(\widehat{V}_{t-1}, V^*)$ for $t \geqslant 1$. Since $\widehat{U}_t$ are leading $J$ left singular vectors of $\mathtt{m}_1(\mathcal{Y} \times_3 \widehat{V}_{t-1}^\top) = \mathtt{m}_1(\mathcal{T}^* \times_3 \widehat{V}_{t-1}^\top) + \mathtt{m}_1(\mathcal{E} \times_3 \widehat{V}_{t-1}^\top)$, and there exists an orthogonal matrix $O \in \mathbb{O}_{J,J}$ such that $U^*O$ are the left singular vectors of $\mathtt{m}_1(\mathcal{T}^* \times_3 \widehat{V}_{t-1}^\top) = U^*\mathtt{m}_1(\mathcal{W}^* \times_3 V^*)(I_{d_2} \otimes \widehat{V}_{t-1})$, by Proposition F.2 and the definition of $\rho(\cdot, \cdot)$, we have

$$\rho(\widehat{U}_{t-1}, U^*) \leqslant \frac{2\sqrt{2}\|\mathtt{m}_1(\mathcal{E} \times_3 \widehat{V}_{t-1}^\top)\|}{\sigma_J(\mathtt{m}_1(\mathcal{T}^* \times_3 \widehat{V}_{t-1}))}.$$

Analogously to (38), we have

$$\sigma_J(\mathtt{m}_1(\mathcal{T}^* \times_3 \widehat{V}_{t-1})) \geqslant \sigma_J(\mathtt{m}_1(\mathcal{T}^*))\sqrt{1 - 2\rho(\widehat{V}_{t-1}, V^*)},$$

provided $\rho(\widehat{V}_{t-1}, V^*) < 1/2$. Analogously to (39), we have

$$\|\mathtt{m}_1(\mathcal{E} \times_3 \widehat{V}_{t-1})\| \leqslant \|\mathtt{m}_1(\mathcal{E})(I_{d_2} \otimes V^*)\| + 2 \sup_{\substack{V \in \mathbb{R}^{d_1 \times K} \\ \|V\| \leqslant 1}} \|\mathtt{m}_1(\mathcal{E})(I_{d_2} \otimes V)\| \cdot \rho(\widehat{V}_{t-1}, V^*). \quad (41)$$

Thus, using the notation of the theorem, we get

$$\rho(\widehat{U}_t, U^*) \leqslant \frac{2\sqrt{2}\left(\alpha_U + 2\beta_U \cdot \rho(\widehat{V}_{t-1}, V^*)\right)}{\sigma_J(\mathtt{m}_1(\mathcal{T}^*))\sqrt{1 - 2\rho(\widehat{V}_{t-1}, V^*)}}. \quad (42)$$

**Step 4. Solving the recursion.** We claim that for each $t = 0, \ldots, T$, we have

$$\rho(\widehat{U}_t, U^*) \leqslant 1/4 \quad \text{and} \quad \rho(\widehat{V}_t, V^*) \leqslant 1/4. \quad (43)$$

Let us prove it by induction. From (37) and conditions of the theorem, we have

$$\rho(\widehat{U}_0, U^*) \leqslant \frac{3\|\mathtt{m}_1(\mathcal{E})\|}{\sigma_J(\mathtt{m}_1(\mathcal{T}^*))} \leqslant \frac{1}{4}.$$

Suppose that we have $\rho(\widehat{U}_t, U^*) \leqslant 1/4$. Let us prove that $\rho(\widehat{V}_t, V^*) \leqslant 1/4$ and $\rho(\widehat{U}_{t+1}, U^*) \leqslant 1/4$. First, applying bound (40), we deduce

$$\rho(\widehat{V}_t, V^*) \leqslant \frac{2\sqrt{2}(\alpha_V + 2\beta_V \cdot \rho(\widehat{U}_t, U^*))}{\sigma_K(\mathtt{m}_3(\mathcal{T}^*))\sqrt{1 - 2\rho(\widehat{U}_t, U^*)}} \leqslant \frac{4(\alpha_V + \beta_V/2)}{\sigma_K(\mathtt{m}_3(\mathcal{T}^*))} \leqslant \frac{6\beta_V}{\sigma_K(\mathtt{m}_3(\mathcal{T}^*))} \leqslant \frac{1}{4},$$

where we used

$$\alpha_V = \|\mathtt{m}_3(\mathcal{E})(U^* \otimes I_{d_2})\| \leqslant \sup_{\substack{U \in \mathbb{R}^{d_1 \times J} \\ \|U\| \leqslant 1}} \|\mathtt{m}_3(\mathcal{E})(U \otimes I_{d_2})\| = \beta_V$$

and $\sigma_K(\mathtt{m}_3(\mathcal{T}^*)) \geqslant 24\beta_V$ due to conditions of the theorem. Similarly, from (40), we deduce

$$\rho(\widehat{U}_{t+1}, U^*) \leqslant \frac{2\sqrt{2}(\alpha_U + 2\beta_U \cdot \rho(\widehat{V}_t, V^*))}{\sigma_J(\mathtt{m}_1(\mathcal{T}^*))\sqrt{1 - 2\rho(\widehat{V}_t, V^*)}}$$

$$\leqslant \frac{4(\alpha_U + \beta_U/2)}{\sigma_J(\mathtt{m}_1(\mathcal{T}^*))} \leqslant \frac{6\beta_U}{\sigma_J(\mathtt{m}_1(\mathcal{T}^*))} \leqslant \frac{6\|\mathtt{m}_1(\mathcal{E})\|}{\sigma_J(\mathtt{m}_1(\mathcal{T}^*))} \leqslant \frac{1}{4},$$

by the conditions of the theorem and the definition of $\alpha_U, \beta_U$. Hence, for each $t = 0, \ldots, T$, we have $\rho(\widehat{U}_t, U^*) \leqslant 1/4$ and $\rho(\widehat{V}_t, V^*) \leqslant 1/4$.

Hence, we can simplify bounds (40),(42) as follows:

$$\rho(\widehat{V}_t, V^*) \leqslant \frac{4 \cdot \left(\alpha_V + 2\beta_V \cdot \rho(\widehat{U}_t, U^*)\right)}{\sigma_K(\mathtt{m}_3(\mathcal{T}^*))},$$

$$\rho(\widehat{U}_t, U^*) \leqslant \frac{4 \cdot \left(\alpha_U + 2\beta_U \cdot \rho(\widehat{V}_{t-1}, V^*)\right)}{\sigma_J(\mathtt{m}_1(\mathcal{T}^*))}.$$

We solve these recursive inequalities using the following proposition.

**Proposition F.3.** *Suppose that a sequence of numbers $(\rho_t, \eta_t)$ satisfies*

$$\rho_t \leqslant x_1 + x_2\eta_t,$$
$$\eta_t \leqslant y_1 + y_2\rho_{t-1}$$

*for some $x_1, y_1, x_2, y_2$ such that $x_2y_2 \leqslant 1/2$ and $x_2, y_2 \geqslant 0$. Then, we have*

$$\rho_t \leqslant 2(x_1 + x_2y_1) + x_2(x_2y_2)^t\eta_0,$$
$$\eta_t \leqslant 2(y_1 + x_1y_2) + (x_2y_2)^t\eta_0.$$

Applying Proposition F.3 to $\rho_t = \rho(\widehat{V}_t, V^*)$, $\eta_t = \rho(\widehat{U}_t, U^*)$, we obtain

$$
\rho(\widehat{V}_t, V^*) \leqslant \frac{8\alpha_V}{\sigma_K(\mathtt{m}_3(\mathcal{T}^*))} + \frac{16\beta_V\alpha_U}{\sigma_J(\mathtt{m}_1(\mathcal{T}^*))\sigma_K(\mathtt{m}_3(\mathcal{T}^*))}
$$
$$
+ \left(\frac{64\beta_V\beta_U}{\sigma_J(\mathtt{m}_1(\mathcal{T}^*))\sigma_K(\mathtt{m}_3(\mathcal{T}^*))}\right)^t \times \frac{24\beta_V\|\mathtt{m}_1(\mathcal{E})\|}{\sigma_K(\mathtt{m}_3(\mathcal{T}^*))\sigma_J(\mathtt{m}_1(\mathcal{T}^*))}, \tag{44}
$$

$$
\rho(\widehat{U}_t, U^*) \leqslant \frac{8\alpha_U}{\sigma_J(\mathtt{m}_1(\mathcal{T}^*))} + \frac{16\beta_U\alpha_V}{\sigma_J(\mathtt{m}_1(\mathcal{T}^*))\sigma_K(\mathtt{m}_3(\mathcal{T}^*))}
$$
$$
+ \left(\frac{64\beta_V\beta_U}{\sigma_J(\mathtt{m}_1(\mathcal{T}^*))\sigma_K(\mathtt{m}_3(\mathcal{T}^*))}\right)^t \times \frac{3\|\mathtt{m}_1(\mathcal{E})\|}{\sigma_J(\mathtt{m}_1(\mathcal{T}^*))}, \tag{45}
$$

where we used (37) to bound $\eta_0 = \rho(\widehat{U}_0, U^*)$.

**Step 4. Final bound.** Let us return to the bound (31). Using $\sqrt{\sum_i a_i} \leqslant \sum_i \sqrt{a_i}$ suitable for any positive numbers $a_i$, we get

$$
\|\widehat{\mathcal{T}} - \mathcal{T}^*\|_{\mathrm{F}} \leqslant \sup_{U\in\mathbb{O}_{d_1,J}, V\in\mathbb{O}_{d_2,K}} \|\mathcal{E} \times_3 V^\top \times_1 U^\top\|_{\mathrm{F}}
$$
$$
+ 2\sqrt{K}\|\mathtt{m}_3(\mathcal{E} \times_1 \widehat{U}^\top)\| + \frac{2\sqrt{J}}{\sigma_{\min}(\widehat{V}_{T-1}^\top V^*)}\|\mathtt{m}_1(\mathcal{E} \times_3 \widehat{V}_{T-1}^\top)\|.
$$

Combining (43) and (36), we obtain

$$
\|\widehat{\mathcal{T}} - \mathcal{T}^*\|_{\mathrm{F}} \leqslant \sup_{U\in\mathbb{O}_{d_1,J}, V\in\mathbb{O}_{d_2,K}} \|\mathcal{E} \times_3 V^\top \times_1 U^\top\|_{\mathrm{F}} + 2\sqrt{K}\|\mathtt{m}_3(\mathcal{E} \times_1 \widehat{U}^\top)\| + 3\sqrt{J}\|\mathtt{m}_1(\mathcal{E} \times_3 \widehat{V}_{T-1}^\top)\|.
$$

Then, applying (34) and (35), we get

$$
\|\widehat{\mathcal{T}} - \mathcal{T}^*\|_{\mathrm{F}} \leqslant \sup_{U\in\mathbb{O}_{d_1,J}, V\in\mathbb{O}_{d_2,K}} \|\mathcal{E} \times_3 V^\top \times_1 U^\top\|_{\mathrm{F}} + 2\sqrt{K}(\alpha_V + 2\beta_V\rho(\widehat{U}_T, U^*))
$$
$$
+ 3\sqrt{J}(\alpha_U + 2\beta_U \cdot \rho(\widehat{V}_{T-1}, V^*)).
$$

Then, we substitute bounds (45),(44) into above, and get

$$
\|\widehat{\mathcal{T}} - \mathcal{T}^*\|_{\mathrm{F}} \leqslant \sup_{U\in\mathbb{O}_{d_1,J}, V\in\mathbb{O}_{d_2,K}} \|\mathcal{E} \times_3 V^\top \times_1 U^\top\|_{\mathrm{F}} + 2\sqrt{K}(\alpha_V + v_1 + v_2)
$$
$$
+ 3\sqrt{J}(\alpha_U + u_1 + u_2),
$$

where

$$
v_1 = 2\beta_V \cdot \frac{16\beta_U\alpha_V}{\sigma_J(\mathtt{m}_1(\mathcal{T}^*))\sigma_K(\mathtt{m}_3(\mathcal{T}^*))},
$$
$$
v_2 = \frac{16\beta_V\alpha_U}{\sigma_J(\mathtt{m}_1(\mathcal{T}^*))} + \frac{6\beta_V\|\mathtt{m}_1(\mathcal{E})\|}{\sigma_J(\mathtt{m}_1(\mathcal{T}^*))} \times \left(\frac{64\beta_V\beta_U}{\sigma_J(\mathtt{m}_1(\mathcal{T}^*))\sigma_K(\mathtt{m}_3(\mathcal{T}^*))}\right)^T,
$$
$$
u_1 = 2\beta_U \cdot \frac{16\beta_V\alpha_U}{\sigma_J(\mathtt{m}_1(\mathcal{T}^*))\sigma_K(\mathtt{m}_3(\mathcal{T}^*))}
$$
$$
u_2 = \frac{16\beta_U\alpha_V}{\sigma_K(\mathtt{m}_3(\mathcal{T}^*))} + \left(\frac{64\beta_U\beta_V}{\sigma_J(\mathtt{m}_1(\mathcal{T}^*))\sigma_K(\mathtt{m}_3(\mathcal{T}^*))}\right)^T \|\mathtt{m}_1(\mathcal{E})\|.
$$

Since $\sigma_J(\mathtt{m}_1(\mathcal{T}^*)) \geqslant 24\|\mathtt{m}_1(\mathcal{E})\| \geqslant 24\beta_U$ and $\sigma_K(\mathtt{m}_3(\mathcal{T}^*)) \geqslant 24\beta_V$, we have $v_1 \leqslant \alpha_V$, $u_1 \leqslant \alpha_U/3$ and

$$
v_2 \leqslant \frac{16\beta_V\alpha_U}{\sigma_J(\mathtt{m}_1(\mathcal{T}^*))} + \left(\frac{64\beta_V\beta_U}{\sigma_J(\mathtt{m}_1(\mathcal{T}^*))\sigma_K(\mathtt{m}_3(\mathcal{T}^*))}\right)^T \|\mathtt{m}_1(\mathcal{E})\|.
$$

Combining the above, we obtain

$$
\|\widehat{\mathcal{T}} - \mathcal{T}^*\|_{\mathrm{F}} \leqslant \sup_{U\in\mathbb{O}_{d_1,J}, V\in\mathbb{O}_{d_2,K}} \|\mathcal{E} \times_3 V^\top \times_1 U^\top\|_{\mathrm{F}} + 4\sqrt{K}\alpha_V + 4\sqrt{J}\alpha_U + \Diamond_2 + r_T,
$$

where $\Diamond_2$ and $r_T$ are introduced in the statement of the theorem. $\qquad\square$

### F.1 Proof of Proposition F.1

*Proof.* For any matrix $O \in \mathbb{O}_{b,b}$, we have

$$\|\Pi_{\widehat{U}} - \Pi_{U*}\| = \|\widehat{U}\widehat{U}^\top - U^*(U^*)^\top\| = \|\widehat{U}\widehat{U}^\top - \widehat{U}O(U^*)^\top + \widehat{U}O(U^*)^\top - U^*(U^*)^\top\|$$
$$\leqslant \|\widehat{U}O(\widehat{U}O - U^*)^\top\| + \|(\widehat{U}O - U^*)(U^*)^\top\| \leqslant 2\|\widehat{U}O - U^*\|.$$

Taking the infimum over $O \in \mathbb{O}_{b,b}$, we obtain the proposition. $\qquad\square$

### F.2 Proof of Proposition F.2

*Proof of Proposition F.2.* For two subspaces $X, Y$ define:

$$\|\sin\Theta(X,Y)\| = \|(I - \Pi_X)\Pi_Y\|.$$

Then, the following theorem holds.

**Theorem F.4** (Wedin $\sin\Theta$-theorem (Wedin, 1972) )**.** *Let $P, Q$ be $\mathbb{R}^{a\times b}$ matrices. Fix $r \leqslant \min\{a,b\}$. Consider the SVD decomposition of $P = U_0\Sigma_0 V_0^\top + U_1\Sigma_1 V_1^\top$, $Q = \widetilde{U}_0\widetilde{\Sigma}_0\widetilde{V}_0^\top + \widetilde{U}_1\widetilde{\Sigma}_1\widetilde{V}_1^\top$, where $\Sigma_0, \widetilde{\Sigma}_0$ corresponds to the first $r$ singular values of $P, Q$ respectively. Suppose that $\sigma_{\min}(\widetilde{\Sigma}_0) - \sigma_{\max}(\Sigma_1) \geqslant \delta$. Then, we have*

$$\|\sin\Theta(\operatorname{Im}\widetilde{U}_0, \operatorname{Im}U_0)\| \leqslant \frac{1}{\delta}\max\{\|(P-Q)V_0^\top\|, \|U_0^\top(P-Q)\|\}.$$

To apply the above theorem, consider two cases. If $\sigma_r(A) \geqslant 2\|E\|$, then we apply the above theorem with $\delta = \sigma_r(A)/2$, $P = B$ and $Q = A$, and get

$$\|\sin\Theta(\operatorname{Im}L, \operatorname{Im}\widehat{L})\| \leqslant \frac{2\|E\|}{\sigma_r(A)}.$$

If $\sigma_r(A) \leqslant 2\|E\|$, then

$$\|\sin\Theta(\operatorname{Im}L, \operatorname{Im}\widehat{L})\| \leqslant 1 \leqslant \frac{2\|E\|}{\sigma_r(A)}.$$

Hence, in either case, we have

$$\|\sin\Theta(\operatorname{Im}L, \operatorname{Im}\widehat{L})\| \leqslant \frac{2\|E\|}{\sigma_r(A)}.$$

Finally, Lemma 1 of (Cai and Zhang, 2018) implies that

$$\rho(L, \widehat{L}) \leqslant \sqrt{2}\|\sin\Theta(\operatorname{Im}L, \operatorname{Im}\widehat{L})\| \leqslant \frac{2\sqrt{2}\|E\|}{\sigma_r(A)},$$

and the proposition follows. $\qquad\square$

### F.3 Proof of Proposition F.3

*Proof of Proposition F.3.* Combining the initial inequalities, we get

$$\eta_t \leqslant y_1 + y_2 x_1 + (x_2 y_2)\eta_{t-1}.$$

Iterating the above inequality $t - 1$ times, we get

$$\eta_t \leqslant (x_2 y_2)^t \eta_0 + (y_1 + y_2 x_1)\sum_{i=0}^{t-1}(x_2 y_2)^i \leqslant \frac{y_1 + y_2 x_1}{1 - x_2 y_2} + (x_2 y_2)^t \eta_0.$$

Using $x_2 y_2 \leqslant 1/2$, we obtain

$$\eta_t \leqslant 2(y_1 + y_2 x_1) + (x_2 y_2)^t \rho_0.$$

Combining the above with the bound $\rho_t \leqslant x_1 + x_2\eta_t$, we derive

$$\rho_t \leqslant x_1 + 2(y_1 x_2 + x_2 y_2 x_1) + x_2(x_2 y_2)^t \rho_0 \leqslant 2(x_1 + x_2 y_1) + x_2(x_2 y_2)^t \rho_0,$$

where we used $x_2 y_2 \leqslant 1/2$ again. $\qquad\square$

## G    PROOF OF THEOREM E.6

*Proof.* **Step 1. Reduction to the PAC-bayes inequality.** Let us rewrite the core expression, as a supremum of a certain empirical process. We have:

$$
\sup_{(A_1,A_2,A_3)\in\prod_{i=1}^3 \mathbb{S}_i} \langle \widehat{\mathcal{E}} \times_3 A_3^\top \times_1 A_1^\top, A_2 \rangle = \sup_{(A_1,A_2,A_3)\in\prod_{i=1}^3 \mathbb{S}_i} \langle A_2 \times_3 A_3^\top \times_1 A_1^\top, \widehat{\mathcal{E}} \rangle
$$

$$
= \sup_{(A_1,A_2,A_3)\in\prod_{i=1}^3 \mathbb{S}_i} \langle A_2 \times_3 A_3 \times_1 A_1, \widehat{\mathcal{E}} \rangle
$$

$$
= \sup_{(A_1,A_2,A_3)\in\prod_{i=1}^3 \mathbb{S}_i} \left\langle A_2 \times_3 A_3 \times_1 A_1, \sum_{i=1}^n \frac{1}{n}\mathcal{R}(\mathbf{X}_i\mathbf{X}_i^\top - \mathbb{E}(\mathbf{X}\mathbf{X}^\top)) \right\rangle
$$

$$
= \sup_{(A_1,A_2,A_3)\in\prod_{i=1}^3 \mathbb{S}_i} \left\langle \mathcal{R}^{-1}(A_2 \times_3 A_3 \times_1 A_1), \frac{1}{n}\sum_{i=1}^n \mathbf{X}_i\mathbf{X}_i^\top - \mathbb{E}(\mathbf{X}\mathbf{X}^\top) \right\rangle
$$

$$
= \sup_{(A_1,A_2,A_3)\in\prod_{i=1}^3 \mathbb{S}_i} \frac{1}{n}\sum_{i=1}^n \left\{ \mathbf{X}_i^\top \mathcal{R}^{-1}(A_2 \times_3 A_3 \times_1 A_1)\mathbf{X}_i \right.
$$

$$
\left. -\mathbb{E}\mathbf{X}^\top \mathcal{R}^{-1}(A_2 \times_3 A_3 \times_1 A_1)\mathbf{X} \right\}.
$$

Define the following functions:

$$
f_i(A_2 \times_3 A_3 \times_1 A_1) = \lambda\left\{ \mathbf{X}_i^\top \mathcal{R}^{-1}(A_2 \times_3 A_3 \times_1 A_1)\mathbf{X}_i - \mathbb{E}\mathbf{X}_i^\top \mathcal{R}^{-1}(A_2 \times_3 A_3 \times_1 A_1)\mathbf{X}_i \right\},
$$

$$
f_\mathbf{X}(A_2 \times_3 A_3 \times_1 A_1) = \lambda\left\{ \mathbf{X}^\top \mathcal{R}^{-1}(A_2 \times_3 A_3 \times_1 A_1)\mathbf{X} - \mathbb{E}\mathbf{X}^\top \mathcal{R}^{-1}(A_2 \times_3 A_3 \times_1 A_1)\mathbf{X} \right\},
$$

where the positive factor $\lambda$ will be chosen later. We will apply Lemma E.5 to the empirical process

$$
\sup_{(A_1,A_2,A_3)\in\prod_{i=1}^s \mathbb{S}_i} \frac{1}{n}\sum_{i=1}^n f_i(A_1, A_2, A_3)
$$

with the parameter space defined by the target spaces $L_i$ dimensionalities and the prior distribution $\mu$, constructed as a product of independent measures for each subspace separately. Choosing bases in $L_1, L_2, L_3$, we identify $A_1, A_2$ with corresponding matrices and $A_3$ with a corresponding tensor. Define linear spaces $\mathbb{L}_1 = \mathbb{R}^{d_1 \times l_1}, \mathbb{L}_2 = \mathbb{R}^{l_1 \times d_2 \times l_3}$ and $\mathbb{L}_3 = \mathbb{R}^{d_2 \times l_3}$, and consider distributions $\mathcal{D}_i$ over $\mathbb{L}_i$ defined as follows:

$$
\mathcal{D}_i = \begin{cases} \mathcal{N}(0, \sigma_i I_{l_i d_i}), & \text{if } l_i \cdot \mathbf{r}_i(\Sigma) \leqslant \log|\mathbb{S}_i|, \\ \text{Uniform}(\mathbb{S}_i), & \text{if } l_i \cdot \mathbf{r}_i(\Sigma) > \log|\mathbb{S}_i|, \end{cases}
$$

for some $\sigma_1, \sigma_2, \sigma_3$ to be chosen later, assuming that samples from the normal distribution have appropriate shapes. Then, we put

$$
\mu = \mathcal{D}_1 \otimes \mathcal{D}_2 \otimes \mathcal{D}_3.
$$

Consider random vectors $P, Q, R$ with mutual distribution $\rho_{A_1,A_2,A_3}$ such that $\mathbb{E}Q \times_3 R \times_1 P = A_2 \times_3 A_3 \times_1 A_1$. Since $f_i(A_1, A_2, A_3), f_\mathbf{X}(A_1, A_2, A_3)$ are linear in $A_2 \times_3 A_3 \times_1 A_1$, we have $\mathbb{E}_{\rho_{A_1,A_3,A_2}} f_i(P, Q, R) = f_i(A_1, A_2, A_3)$, so Lemma E.5 yields

$$
\sup_{\substack{A_1\in\mathbb{S}_1,\\ A_2\in\mathbb{S}_2, A_3\in\mathbb{S}_3}} \frac{1}{n}\sum_{i=1}^n f_i(A_1, A_2, A_3)
$$

$$
\leqslant \sup_{\substack{A_1\in\mathbb{S}_1,\\ A_2\in\mathbb{S}_2, A_3\in\mathbb{S}_3}} \left\{ \mathbb{E}_{\rho_{A_1,A_2,A_3}} \log \mathbb{E}_\mathbf{X} \exp f_\mathbf{X}(P, Q, R) + \frac{\mathcal{KL}(\rho_{A_1,A_2,A_3}, \mu) + \log(1/\delta)}{n} \right\} \tag{46}
$$

with probability at least $1 - \delta$. Then, we construct $\rho_{A_1,A_2,A_3}$ such that the right-hand side of the above inequality can be controlled efficiently.

**Step 2. Constructing $\rho_{A_1,A_2,A_3}$.** Suppose for a while that $\rho_{A_1,A_2,A_3}$-almost surely we have

$$
\lambda\|\Sigma^{1/2}\mathcal{R}^{-1}(Q \times_3 R \times_1 P)\Sigma^{1/2}\|_\mathrm{F} \leqslant 1/\omega. \tag{47}
$$

Then, Assumption 2.1 implies

$$
\begin{aligned}
\mathbb{E}_{\rho_{A_1,A_2,A_3}} &\log \mathbb{E}_{\mathbf{X}} \exp f_{\mathbf{X}}(P,Q,R) \\
&= \mathbb{E}_{\rho_{A_1,A_2,A_3}} \log \mathbb{E}_{\mathbf{X}} \exp \left\{ \lambda \left( \mathbf{X}^{\top} \mathcal{R}^{-1}(Q \times_3 R \times_1 P)\mathbf{X} \right. \right. \\
&\hspace{6em} \left. \left. - \mathbb{E} \mathbf{X}^{\top} \mathcal{R}^{-1}(Q \times_3 R \times_1 P)\mathbf{X} \right) \right\} \\
&\leqslant \lambda^2 \omega^2 \mathbb{E}_{\rho_{A_1,A_2,A_3}} \| \Sigma^{1/2} \mathcal{R}^{-1}(Q \times_3 R \times_1 P)\Sigma^{1/2} \|_{\mathrm{F}}^2.
\end{aligned}
\tag{48}
$$

So, to control the above and keep the left-hand side of (47) bounded, we do the following. Consider random matrices $G_1 \in \mathbb{R}^{d_1 \times l_1}, G_3 \in \mathbb{R}^{d_3 \times l_3}$ and a random tensor $G_3 \in \mathbb{R}^{l_1 \times d_2 \times l_3}$ such that

$$
\mathbf{vec}(G_i) \sim \begin{cases} \mathcal{N}(0, \sigma_i I_{d_i l_i}), & \text{if } \mathbf{r}_i(\Sigma) \leqslant \log |\mathbb{S}_i|, \\ \delta_0, & \text{if } l_i \cdot \mathbf{r}_i(\Sigma) > \log |\mathbb{S}_i|, \end{cases}
$$

where $\delta_0$ is the delta measure supported on $0 \in \mathbb{R}^{d_i l_i}$. Then, define a function $g : \mathbb{R}^{d_1 \times l_1} \times \mathbb{R}$

$$
g(u', v', w') = \| \Sigma^{1/2} \mathcal{R}^{-1}(v' \times_3 w' \times_1 u')\Sigma^{1/2} \|_{\mathrm{F}}^2.
\tag{49}
$$

Sequentially applying the triangle inequality for the Frobenius norm and using $(a+b)^2 \leqslant 2a^2 + 2b^2$, we obtain

$$
\begin{aligned}
f(A_1 + G_1, A_2 + G_2, A_3 + G_3) &\leqslant 2g(A_1, A_2 + G_2, A_3 + G_3) + 2g(G_1, A_2 + G_2, A_3 + G_3) \\
&\leqslant 4g(A_1, A_2, A_3 + G_3) + 4g(G_1, G_2, A_3 + G_3) \\
&\quad + 4g(A_1, G_2, A_3 + G_3) + 4g(G_1, A_2, A_3 + G_3) \\
&\leqslant 8g(A_1, A_3, A_2) + 8g(A_1, G_2, G_3) + 8g(A_1, A_3, G_3) + 8g(A_1, G_2, A_2) \\
&\quad + 8g(G_1, A_3, A_2) + 8g(G_1, G_2, G_3) + 8g(G_1, A_3, G_3) + 8g(G_1, G_2, A_2).
\end{aligned}
\tag{50}
$$

Then, we define the distribution $\rho_{A_1,A_2,A_3}$ of the random vector $(P, Q, R)$ as the distribution of $(A_1 + G_1, A_2 + G_2, A_3 + G_3)$ subject to the condition

$$
\begin{aligned}
(G_1, G_2, G_3) \in \Upsilon &= \{8g(a,b,c) \leqslant 8\mathbb{E}g(a,b,c) \mid (a,b,c) \in \Gamma\}, \quad \text{where} \\
\Gamma &= (\{A_1, G_1\} \times \{A_2, G_2\} \times \{A_3, G_3\}) \backslash \{(A_1, A_3, A_2)\}.
\end{aligned}
$$

Note that by the union bound and the Markov inequality, we have

$$
\begin{aligned}
\Pr\left((G_1, G_2, G_3) \notin \Upsilon\right) &\leqslant \sum_{(a,b,c) \in \Gamma} \Pr\left(f(a,b,c) > 8\mathbb{E}f(a,b,c)\right) \\
&\leqslant \sum_{(a,b,c) \in \Gamma} \frac{1}{8} = \frac{7}{8}.
\end{aligned}
\tag{51}
$$

Combining the definition of Upsilon with upper bound (50) implies the following bound on $g(P,Q,R)$:

$$
\begin{aligned}
g(P,Q,R) \leqslant 64 \big( &g(A_1, A_2, A_3) + \mathbb{E}g(A_1, A_2, G_3) + \mathbb{E}g(A_1, G_2, A_3) + \mathbb{E}g(A_1, G_2, G_3) \\
&+ \mathbb{E}g(G_1, A_2, A_3) + \mathbb{E}g(G_1, A_2, G_3) + \mathbb{E}g(G_1, G_2, A_3) + \mathbb{E}g(G_1, G_2, G_3) \big),
\end{aligned}
\tag{52}
$$

which holds $\rho_{A_1,A_2,A_3}$-almost surely.

Let us check that $\mathbb{E}_{\rho_{A_1,A_3,A_2}} Q \times_3 R \times_1 P = A_2 \times_3 A_3 \times_1 A_1$. Since both the Gaussian distribution and $\delta_0$ are centrally symmetric and the function $f$ does not change its value when multiplying any of its argument by $-1$, we have

$$
(P, Q, R) \overset{d}{=} (A_1 + \varepsilon_1(P - A_1), A_2 + \varepsilon_2(Q - A_2), A_3 + \varepsilon_3(R - A_3)),
\tag{53}
$$

where $\varepsilon_1, \varepsilon_2, \varepsilon_3$ are i.i.d. Rademacher random variables independent of $(P, Q, R)$. Then, we obtain

$$
\begin{aligned}
\mathbb{E}Q \times_3 R \times_1 P &= \mathbb{E}(A_2 + \varepsilon_2(Q - A_2)) \times_3 (A_3 + \varepsilon_3(R - A_3)) \times_1 A_1 \\
&\quad + \mathbb{E}(A_2 + \varepsilon_2(Q - A_2)) \times_3 (A_3 + \varepsilon_3(R - A_3)) \times_1 \varepsilon_1(P - A_1) \\
&= \mathbb{E}(A_2 + \varepsilon_2(Q - A_2)) \times_3 A_3 \times_1 A_1 + \mathbb{E}(A_2 + \varepsilon_2(Q - A_2)) \times_3 \varepsilon_3(R - A_3)) \times_1 A_1 \\
&= A_2 \times_3 A_3 \times_1 A_1 + \mathbb{E}\varepsilon_2(Q - A_2) \times_3 A_3 \times_1 A_1 = A_2 \times_3 A_3 \times_1 A_1.
\end{aligned}
$$

Hence, to satisfy the assumption (47) and use (48), it is enough to bound expectations $\mathbb{E}f(a, b, c)$ for $(a, b, c) \in \{A_1, G_1\} \times \{A_3, G_3\} \times \{A_2, G_2\}$.

**Step 3. Bounding expectations** $\mathbb{E}g(\cdot, \cdot, \cdot)$**.** Let us start with $g(A_1, A_3, A_2)$. From the definition (49), we have

$$
\begin{aligned}
g(A_1, A_2, A_3) &= \|\Sigma^{1/2}\mathcal{R}^{-1}(A_2 \times_3 A_3 \times_1 A_1)\Sigma^{1/2}\|_{\mathrm{F}}^2 \\
&\leqslant \|\Sigma\|^2 \|\mathcal{R}^{-1}(A_2 \times_3 A_3 \times_1 A_1)\|_{\mathrm{F}}^2 = \|\Sigma\|^2 \|A_2 \times_3 A_3 \times_1 A_1\|_{\mathrm{F}}^2 = \|\Sigma\|^2,
\end{aligned}
\tag{54}
$$

where we used the fact that $A_2$ has unit Frobenius norm and $\|A_1\| \leqslant 1$, $\|A_3\| \leqslant 1$ by the definition of $\mathbb{S}_i$.

In what follows, it will be useful to rewrite the function $f(A_1, A_2, A_3)$ in different notation. As in the proof of Lemma E.1, define tensors

$$
\begin{aligned}
\mathcal{S}_{p_1 q_1 r_1 p_2 q_2 r_2} &= \Sigma_{(p_1-1)qr+(q_1-1)r+r_1, (p_2-1)qr+(q_2-1)r+r_2} \\
\mathtt{A}^{(1)}_{p_2 p_3 j_1} &= (A_1)_{(p_2-1)p+p_3, j_1}, \quad \mathtt{A}^{(3)}_{r_2 r_3 k_1} = (A_3)_{(r_2-1)r+r_3, k_1}, \\
\mathtt{A}^{(2)}_{j_1 q_2 q_3 k_1} &= (A_3)_{j_1, (q_2-1)q+q_3, k_1}, \\
\mathcal{G}^{(1)}_{p_2 p_3 j_1} &= (G_1)_{(p_2-1)p+p_3, j_1}, \quad \mathcal{G}^{(3)}_{r_2 r_3 k_1} = (G_3)_{(r_2-1)r+r_3, k_1}, \\
\mathcal{G}^{(2)}_{j_1 q_2 q_3 k_1} &= (G_3)_{j_1, (q_2-1)q+q_3, k_1}.
\end{aligned}
$$

Then, we obtain

$$
\begin{aligned}
g(A_1, A_2, A_3) &= \|\Sigma^{1/2}\mathcal{R}^{-1}(A_2 \times_3 A_3 \times_1 A_1)\Sigma^{1/2}\|_{\mathrm{F}}^2 \\
&= \mathrm{Tr}\left(\Sigma\mathcal{R}^{-1}(A_2 \times_3 A_3 \times_1 A_1)\Sigma\mathcal{R}^{-\top}(A_2 \times_3 A_3 \times_1 A_1)\right) \\
&= \mathcal{S}_{p_1 q_1 r_1 p_2 q_2 r_2} \mathtt{A}^{(1)}_{p_2 p_3 j_1} \mathtt{A}^{(2)}_{j_1 q_2 q_3 k_1} \mathtt{A}^{(3)}_{r_2 r_3 k_1} \mathcal{S}_{p_3 q_3 r_3 p_4 q_4 r_4} \mathtt{A}^{(1)}_{p_1 p_4 j_2} \mathtt{A}^{(2)}_{j_2 q_1 q_4 k_2} \mathtt{A}^{(3)}_{r_1 r_4 k_2}.
\end{aligned}
\tag{55}
$$

Note that the above holds for any $A_i \in \mathbb{L}_i$, so the formula remains true when replacing $A_i$, $\mathtt{A}^{(i)}$ with $G_i$, $\mathcal{G}^{(i)}$ respectively.

Next, we bound $\mathbb{E}g(A_1, A_2, G_3)$. If $\mathbf{vec}(G_1) \sim \delta_0$, we have $\mathbb{E}g(A_1, A_2, G_3) = 0$, so it is enough to consider the case $\mathbf{vec}(G_3) \sim \mathcal{N}(0, \sigma_3 I_{d_3 l_3})$. Due to formula (55), it yields

$$
\begin{aligned}
\mathbb{E}g(A_1, A_2, G_3) &= \mathbb{E}\mathcal{S}_{p_1 q_1 r_1 p_2 q_2 r_2} \mathtt{A}^{(1)}_{p_2 p_3 j_1} \mathtt{A}^{(2)}_{j_1 q_2 q_3 k_1} \mathcal{G}^{(3)}_{r_2 r_3 k_1} \mathcal{S}_{p_3 q_3 r_3 p_4 q_4 r_4} \mathtt{A}^{(1)}_{p_1 p_4 j_2} \mathtt{A}^{(2)}_{j_2 q_1 q_4 k_2} \mathcal{G}^{(3)}_{r_1 r_4 k_2} \\
&= \sigma_3^2 \delta_{r_2 r_1} \delta_{r_3 r_3} \delta_{k_1 k_2} \mathcal{S}_{p_1 q_1 r_1 p_2 q_2 r_2} \mathtt{A}^{(1)}_{p_2 p_3 j_1} \mathtt{A}^{(2)}_{j_1 q_2 q_3 k_1} \mathcal{S}_{p_3 q_3 r_3 p_4 q_4 r_4} \mathtt{A}^{(1)}_{p_1 p_4 j_2} \mathtt{A}^{(2)}_{j_2 q_1 q_4 k_2} \\
&= \sigma_3^2 \mathcal{S}_{p_1 q_1 r_1 p_2 q_2 r_1} \mathtt{A}^{(1)}_{p_2 p_3 j_1} \mathtt{A}^{(2)}_{j_1 q_2 q_3 k_1} \mathcal{S}_{p_3 q_3 r_3 p_4 q_4 r_3} \mathtt{A}^{(1)}_{p_1 p_4 j_2} \mathtt{A}^{(2)}_{j_2 q_1 q_4 k_1}.
\end{aligned}
$$

Define matrices $\widetilde{A}^{(1,j)} \in \mathbb{R}^{p \times p}$, $\widetilde{A}^{(1,j,k)}$, $i = 1, 2$ and $j = 1, \dots, J$, by $\widetilde{A}^{(1,j)}_{p_2, p_3} = \mathtt{A}^{(1)}_{p_2 p_3 j_1}$ and $\widetilde{A}^{(2,j,k)}_{q_2, q_3} = \mathtt{A}^{(2)}_{j p_2 p_3 k}$. Then, we have

$$
\begin{aligned}
\mathbb{E}g(A_1, A_2, G_3) &= \sigma_3^2 \cdot \sum_{k_1 \in [l_3]} \mathrm{Tr}\left(\mathrm{Tr}_3(\Sigma) \sum_{j_1=1}^{l_1} \widetilde{A}^{(1,j_1)} \otimes \widetilde{A}^{(2,j_1,k_1)} \right. \\
&\qquad\qquad\qquad \left. \times \mathrm{Tr}_3(\Sigma) \sum_{j_2=1}^{l_1} (\widetilde{A}^{(1,j_2)} \otimes \widetilde{A}^{(2,j_2,k_1)})^\top \right) \\
&\leqslant \sigma_3^2 \sum_{k_1 \in [l_3]} \left\| \mathrm{Tr}_3(\Sigma) \cdot \sum_{j_1 \in [J]} \widetilde{A}^{(1,j_1)} \otimes \widetilde{A}^{(2,j_1,k_1)} \right\|_{\mathrm{F}}^2 \\
&\leqslant \sigma_3^2 \|\mathrm{Tr}_3(\Sigma)\|^2 \cdot \sum_{k_1 \in [l_3]} \left\| \sum_{j_1 \in [l_1]} \widetilde{A}^{(1,j_1)} \otimes \widetilde{A}^{(2,j_1,k_1)} \right\|_{\mathrm{F}}^2,
\end{aligned}
\tag{56}
$$

where we used the Cauchy–Schwartz inequality for the scalar product $\langle A, B \rangle = \text{Tr}(A^\top B) \leqslant \|A\|_F \|B\|_F$. Then, we introduce matrices $A'^{(2,k_1)}_{j_1,(q_2-1)q+q_3} = \mathtt{A}_{j_1 q_2 q_3 k_1}$, $k_1 \in [l_3]$, for which we have

$$\sum_{k_1 \in [l_3]} \| \sum_{j_1 \in [l_1]} \widetilde{A}^{(1,j_1)} \otimes \widetilde{A}^{(2,j_1,k_1)} \|_F^2 = \sum_{k_1 \in [l_3]} \|A_1^\top A'^{(2,k_1)}\|_F^2 \leqslant \sum_{k_1 \in [l_3]} \|A_1^\top\|^2 \|A'^{(2,k_1)}\|_F^2$$

$$\leqslant \sum_{k_1 \in [l_3]} \|A'^{(2,k_1)}\|_F^2 = \|A_2\|_F^2 \leqslant 1,$$

where we used $\|A_1\| \leqslant 1$ and $\|A_2\|_F \leqslant 1$. Substituting the above into (56) yields

$$\mathbb{E}g(A_1, A_2, G_3) \leqslant \sigma_3^2 \|\text{Tr}_3(\Sigma)\|^2. \tag{57}$$

Analogously, we obtain

$$\mathbb{E}g(G_1, A_2, A_3) \leqslant \sigma_1^2 \|\text{Tr}_1(\Sigma)\|^2 \tag{58}$$

Next, we study the term $\mathbb{E}g(A_1, G_2, A_3)$. Obviously, if $\mathbf{vec}(G_2) \sim \delta_0$, then $\mathbb{E}g(A_1, G_2, A_3) = 0$, so we consider the case then $\mathbf{vec}(G_2) \sim \mathcal{N}(0, \sigma_3 I_{d_2 l_2})$. Using (55) with $G_2$ in place of $A_2$ and defining a matrix $\widetilde{A}^{(3,k_1)} \in \mathbb{R}^{r \times r}$ as $\widetilde{A}^{(3,k_1)}_{r_2 r_3} = \mathtt{A}^{(3)}_{r_2 r_3 k_1}$, we obtain

$$\mathbb{E}g(A_1, G_2, A_3) = \mathbb{E}\mathcal{S}_{p_1 q_1 r_1 p_2 q_2 r_2} \mathtt{A}^{(1)}_{p_2 p_3 j_1} \mathcal{G}^{(2)}_{j_1 q_2 q_3 k_1} \mathtt{A}^{(3)}_{r_2 r_3 k_1} \mathcal{S}_{p_3 q_3 r_3 p_4 q_4 r_4} \mathtt{A}^{(1)}_{p_1 p_4 j_2} \mathcal{G}^{(2)}_{j_2 q_1 q_4 k_2} \mathtt{A}^{(3)}_{r_1 r_4 k_2},$$

$$= \sigma_2^2 \delta_{j_1 j_2} \delta_{q_1 q_2} \delta_{k_1 k_2} \mathcal{S}_{p_1 q_1 r_1 p_2 q_2 r_2} \mathtt{A}^{(1)}_{p_2 p_3 j_1} \mathtt{A}^{(3)}_{r_2 r_3 k_1} \mathcal{S}_{p_3 q_3 r_3 p_4 q_4 r_4} \mathtt{A}^{(1)}_{p_1 p_4 j_2} \mathtt{A}^{(3)}_{r_1 r_4 k_2}$$

$$= \sigma_2^2 \mathcal{S}_{p_1 q_1 r_1 p_2 q_1 r_2} \mathtt{A}^{(1)}_{p_2 p_3 j_1} \mathtt{A}^{(3)}_{r_2 r_3 k_1} \mathcal{S}_{p_3 q_2 r_3 p_4 q_2 r_4} \mathtt{A}^{(1)}_{p_1 p_4 j_1} \mathtt{A}^{(3)}_{r_1 r_4 k_1}$$

$$= \sigma_2^2 \sum_{j_1 \in [l_1], k_1 \in [l_3]} \text{Tr}\left( \text{Tr}_2(\Sigma) \cdot [\widetilde{A}^{(1,j_1)} \otimes \widetilde{A}^{(3,k_1)}] \cdot \text{Tr}_2(\Sigma) \cdot [\widetilde{A}^{(1,j_1)} \otimes \widetilde{A}^{(3,k_1)}]^\top \right)$$

$$\leqslant \sigma_2^2 \sum_{j_1 \in [l_1], k_1 \in [l_3]} \|\text{Tr}_2(\Sigma) \cdot [\widetilde{A}^{(1,j_1)} \otimes \widetilde{A}^{(3,k_1)}]\|_F^2,$$

where we used the Cauchy–Schwartz inequality on the last line. It yields

$$\mathbb{E}g(A_1, G_2, A_3) \leqslant \sigma_2^2 \|\text{Tr}_2(\Sigma)\|^2 \sum_{j_1 \in [l_1], k_1 \in [l_3]} \|\widetilde{A}^{(1,j_1)} \otimes \widetilde{A}^{(3,k_1)}\|_F^2$$

$$= \sigma_2^2 \|\text{Tr}_2(\Sigma)\|^2 \sum_{j_1 \in [l_1], k_1 \in [l_3]} \|\widetilde{A}^{(1,j_1)}\|_F^2 \|\widetilde{A}^{(3,k_1)}\|_F^2$$

$$= \sigma_2^2 \|\text{Tr}_2(\Sigma)\|^2 \|A_1\|_F^2 \|A_3\|_F^2 \leqslant \sigma_2^2 l_1 l_3 \|\text{Tr}_2(\Sigma)\|^2, \tag{59}$$

where we used $\|A_i\|_F^2 \leqslant l_i \|A_i\|^2 \leqslant l_i$ for $i = 1, 3$.

Next, we bound $\mathbb{E}g(A_1, G_2, G_3)$. If either $\mathbf{vec}(G_2) \sim \delta_0$ or $\mathbf{vec}(G_3) \sim \delta_0$, then $\mathbb{E}g(A_1, G_2, G_3) = 0$, so we consider the case when both $\mathbf{vec}(G_2) \sim \mathcal{N}(0, \sigma_2^2 I_{d_2 l_2})$ and $\mathbf{vec}(G_3) \sim \mathcal{N}(0, \sigma_3^2 I_{d_3 l_3})$. Using (55) with $G_2, G_3$ in place of $A_2, A_3$, we get

$$\mathbb{E}g(A_1, G_2, G_3) = \mathbb{E}\mathcal{S}_{p_1 q_1 r_1 p_2 q_2 r_2} \mathtt{A}^{(1)}_{p_2 p_3 j_1} \mathcal{G}^{(2)}_{j_1 q_2 q_3 k_1} \mathcal{G}^{(3)}_{r_2 r_3 k_1} \mathcal{S}_{p_3 q_3 r_3 p_4 q_4 r_4} \mathtt{A}^{(1)}_{p_1 p_4 j_2} \mathcal{G}^{(2)}_{j_2 q_1 q_4 k_2} \mathcal{G}^{(3)}_{r_1 r_4 k_2},$$

$$= \sigma_2^2 \sigma_3^2 \delta_{k_1 k_1} \mathcal{S}_{p_1 q_1 r_1 p_2 q_1 r_1} \mathtt{A}^{(1)}_{p_2 p_3 j_1} \mathcal{S}_{p_3 q_3 r_3 p_4 q_3 r_3} \mathtt{A}^{(1)}_{p_1 p_4 j_1}$$

$$= \sigma_2^2 \sigma_3^2 l_3 \sum_{j_1 = 1}^{l_1} \text{Tr}\left( \text{Tr}_{2,3}(\Sigma) \widetilde{A}^{(1,j_1)} \text{Tr}_{2,3}(\Sigma) (\widetilde{A}^{(1,j_1)})^\top \right)$$

$$\leqslant \sigma_2^2 \sigma_3^2 l_3 \sum_{j_1 = 1}^{l_1} \|\text{Tr}_{2,3}(\Sigma) \widetilde{A}^{(1,j_1)}\|_F^2 \leqslant \sigma_2^2 \sigma_3^2 l_3 \|\text{Tr}_{2,3}(\Sigma)\|^2 \sum_{j_1 = 1}^{l_1} \|\widetilde{A}^{(1,j_1)}\|_F^2$$

$$= \sigma_2^2 \sigma_3^2 l_3 \|\text{Tr}_{2,3}(\Sigma)\|^2 \|A_1\|_F^2.$$

Since $\|A_1\|_F^2 \leqslant l_1 \|A\|^2$, we obtain

$$\mathbb{E}g(A_1, G_2, G_3) \leqslant \sigma_2^2 \sigma_3^2 l_1 l_3 \|\text{Tr}_{2,3}(\Sigma)\|^2. \tag{60}$$

Analogously, we get

$$\mathbb{E}g(G_1, G_2, A_3) \leqslant \sigma_1^2 \sigma_2^2 l_1 l_3 \|\mathrm{Tr}_{1,2}(\Sigma)\|^2. \tag{61}$$

Then, we bound $\mathbb{E}g(G_1, A_2, G_3)$. Using (55) with $G_1, G_3$ in place of $A_1, A_3$, we get

$$
\begin{aligned}
\mathbb{E}g(G_1, A_2, G_3) &= \mathbb{E}\mathcal{S}_{p_1 q_1 r_1 p_2 q_2 r_2} \mathcal{G}^{(1)}_{p_2 p_3 j_1} \mathtt{A}^{(2)}_{j_1 q_2 q_3 k_1} \mathcal{G}^{(3)}_{r_2 r_3 k_1} \mathcal{S}_{p_3 q_3 r_3 p_4 q_4 r_4} \mathcal{G}^{(1)}_{p_1 p_4 j_2} \mathtt{A}^{(2)}_{j_2 q_1 q_4 k_2} \mathcal{G}^{(3)}_{r_1 r_4 k_2} \\
&= \sigma_1^2 \sigma_3^2 \delta_{p_1 p_2} \delta_{j_1 j_2} \delta_{r_1 r_2} \delta_{k_1 k_2} \delta_{p_3 p_4} \delta_{r_3 r_4} \\
&\quad \times \mathcal{S}_{p_1 q_1 r_1 p_2 q_2 r_2} \mathtt{A}^{(2)}_{j_1 q_2 q_3 k_1} \mathcal{S}_{p_3 q_3 r_3 p_4 q_4 r_4} \mathtt{A}^{(2)}_{j_2 q_1 q_4 k_2} \\
&= \sigma_1^2 \sigma_3^2 \mathcal{S}_{p_1 q_1 r_1 p_1 q_2 r_1} \mathtt{A}^{(2)}_{j_1 q_2 q_3 k_1} \mathcal{S}_{p_3 q_3 r_3 p_3 q_4 r_3} \mathtt{A}^{(2)}_{j_1 q_1 q_4 k_1} \\
&= \sigma_1^2 \sigma_2^2 \sum_{j_1 \in [l_1], k_1 \in [l_3]} \mathrm{Tr}\left(\mathrm{Tr}_{1,3}(\Sigma)\widetilde{A}^{(2,j_1,k_1)} \mathrm{Tr}_{1,3}(\Sigma)(\widetilde{A}^{(2,j_1,k_1)})^\top\right).
\end{aligned}
$$

By the Cauchy–Schwartz inequality for the matrix product, we obtain

$$
\begin{aligned}
\mathbb{E}g(G_1, A_2, G_3) &\leqslant \sigma_1^2 \sigma_3^2 \sum_{j_1 \in [l_1], k_1 \in [l_3]} \|\mathrm{Tr}_{2,3}(\Sigma)\widetilde{A}^{(2,j_1,k_1)}\|_{\mathrm{F}}^2 \\
&\leqslant \sigma_1^2 \sigma_3^2 \|\mathrm{Tr}_{2,3}(\Sigma) \sum_{j_1 \in [l_1], k_1 \in [l_3]} \|\widetilde{A}^{(2,j_1,k_1)}\|_{\mathrm{F}}^2 \\
&= \sigma_1^2 \sigma_3^2 \|\mathrm{Tr}_{2,3}(\Sigma)\|^2 \|A_2\|_{\mathrm{F}}^2 = \sigma_1^2 \sigma_3^2 \|\mathrm{Tr}_{2,3}(\Sigma)\|^2. \tag{62}
\end{aligned}
$$

Finally, we bound $\mathbb{E}g(G_1, G_2, G_3)$. If some $G_i$ is distributed according to $\delta_0$, then $\mathbb{E}g(G_1, G_2, G_3) = 0$, so it is enough to consider the case when $G_1, G_2, G_3$ are Gaussian. Using (55) with $A_i, \mathtt{A}^{(i)}$ replaced by $G_i, \mathcal{G}^{(i)}$, we obtain

$$
\begin{aligned}
\mathbb{E}g(G_1, G_2, G_3) &= \mathbb{E}\mathcal{S}_{p_1 q_1 r_1 p_2 q_2 r_2} \mathcal{G}^{(1)}_{p_2 p_3 j_1} \mathcal{G}^{(2)}_{j_1 q_2 q_3 k_1} \mathcal{G}^{(3)}_{r_2 r_3 k_1} \mathcal{S}_{p_3 q_3 r_3 p_4 q_4 r_4} \mathcal{G}^{(1)}_{p_1 p_4 j_2} \mathcal{G}^{(2)}_{j_2 q_1 q_4 k_2} \mathcal{G}^{(3)}_{r_1 r_4 k_2} \\
&= \sigma_1^2 \sigma_2^2 \sigma_3^2 \delta_{j_1 j_1} \delta_{k_1 k_2} \mathcal{S}_{p_1 q_1 r_1 p_1 q_1 r_1} \mathcal{S}_{p_3 q_3 r_3 p_3 q_3 r_3} \\
&= \sigma_1^2 \sigma_2^2 \sigma_3^2 l_1 l_3 \mathrm{Tr}(\Sigma)^2. \tag{63}
\end{aligned}
$$

We summarized obtained bounds on $\mathbb{E}g(\cdot, \cdot, \cdot)$ in Table 4.

| Quantity | Bound | Ref. |
|---|---|---|
| $g(A_1, A_2, A_3)$ | $\|\Sigma\|^2$ | (54) |
| $\mathbb{E}g(A_1, A_2, G_3)$ | $\sigma_3^2 \|\mathrm{Tr}_3(\Sigma)\|$ | (57) |
| $\mathbb{E}g(G_1, A_2, A_3)$ | $\sigma_1^2 \|\mathrm{Tr}_1(\Sigma)\|^2$ | (58) |
| $\mathbb{E}g(A_1, G_2, A_3)$ | $\sigma_2^2 l_1 l_3 \|\mathrm{Tr}_2(\Sigma)\|^2$ | (59) |
| $\mathbb{E}g(A_1, G_2, G_3)$ | $\sigma_2^2 \sigma_3^2 l_1 l_3 \|\mathrm{Tr}_{2,3}(\Sigma)\|^2$ | (60) |
| $\mathbb{E}g(G_1, G_2, A_3)$ | $\sigma_1^2 \sigma_2^2 l_1 l_3 \|\mathrm{Tr}_{1,2}(\Sigma)\|^2$ | (61) |
| $\mathbb{E}g(G_1, A_2, G_3)$ | $\sigma_1^2 \sigma_3^2 \|\mathrm{Tr}_{2,3}(\Sigma)\|^2$ | (62) |
| $\mathbb{E}g(G_1, G_2, G_3)$ | $\sigma_1^2 \sigma_2^2 \sigma_3^2 l_1 l_3 \mathrm{Tr}(\Sigma)^2$ | (63) |

Table 4: Bounds on $\mathbb{E}g(\cdot, \cdot, \cdot)$.

Combining (52) with bounds (54),(58)-(63) implies the following $\rho_{A_1, A_2, A_3}$-almost surely:

$$
\begin{aligned}
g(P, Q, R) \leqslant 64 \big( &\|\Sigma\|^2 + \sigma_1^2 \sigma_2^2 \sigma_3^2 l_1 l_3 \mathrm{Tr}(\Sigma)^2 \\
&+ \sigma_3^2 \|\mathrm{Tr}_3(\Sigma)\|^2 + \sigma_2^2 l_1 l_3 \|\mathrm{Tr}_2(\Sigma)\|^2 + \sigma_1^2 \|\mathrm{Tr}_1(\Sigma)\| \\
&+ \sigma_2^2 \sigma_3^2 l_1 l_3 \|\mathrm{Tr}_{2,3}(\Sigma)\|^2 + \sigma_1^2 \sigma_2^2 l_1 l_3 \|\mathrm{Tr}_{1,2}(\Sigma)\|^2 + \sigma_1^2 \sigma_3^2 \|\mathrm{Tr}_{2,3}(\Sigma)\|^2 \big).
\end{aligned}
$$

Finally, we choose $\sigma_1^2, \sigma_2^2, \sigma_3^2$ as follows:

$$\sigma_1 = \mathbf{r}_1^{-1}(\Sigma), \qquad \sigma_2 = \mathbf{r}_2^{-1}(\Sigma)/\sqrt{l_1 l_3}, \qquad \sigma_3 = \mathbf{r}_3^{-1}(\Sigma).$$

Then, $\rho_{A_1, A_2, A_3}$-almost surely, we have

$$\|\Sigma^{1/2} \mathcal{R}^{-1}(P \times_1 R \times_3 Q)\Sigma^{1/2}\|_{\mathrm{F}}^2 = f(P, Q, R) \leqslant 2^{12}\|\Sigma\|^2,$$

where we used $\|\mathrm{Tr}_S(\Sigma)\| \leqslant \|\Sigma\| \cdot \prod_{s \in S} \mathbf{r}_s(\Sigma)$ for any non-empty $S$. Hence, if $\lambda$ satisfies

$$2^6 \lambda \omega \|\Sigma\| \leqslant 1, \tag{64}$$

then (47) is fulfilled and, due to (48), we have

$$\mathbb{E}_{\rho_{A_1, A_2, A_3}} \log \mathbb{E}_{\mathbf{X}} \exp f_{\mathbf{X}}(P, Q, R) \leqslant 2^{12}\lambda^2\omega^2\|\Sigma\|^2. \tag{65}$$

**Step 4. Bounding the Kullback-Leibler divergence.** Define $I = \{i \in [3] \mid l_i \mathbf{r}_i(\Sigma) > \log|\mathbb{S}_i|\}$. Then, for $i \in I$, we have $\mathcal{D}_i = \mathrm{Uniform}(\mathbb{S}_i)$ and the density of $\rho_{A_1, A_2, A_3}$ is given by

$$\rho_{A_1, A_2, A_3}(a_1, a_2, a_3) = \prod_{i \in I} \delta_0(a_i - A_i) \times \prod_{i \in [3] \setminus I} \frac{\sigma_i^{-l_i d_i}}{(2\pi)^{l_i d_i/2}} \exp\left\{-\frac{1}{2\sigma_i^2}\|a_i - A_i\|_{\mathrm{F}}^2\right\}$$

$$\times \frac{\mathbb{1}\{(a_1 - A_1, a_2 - A_2, a_3 - A_3) \in \Upsilon\}}{\mathrm{Pr}((G_1, G_2, G_3) \in \Upsilon)}.$$

By the definition of $\Upsilon$, $\rho_{A_1, A_2, A_3}$ can be decomposed into product of the truncated Gaussian $\rho_{-I}$ and delta measures $\bigotimes_{i \in I} \delta_{A_i}$. Hence, we have

$$\mathcal{KL}(\rho_{A_1, A_2, A_3}, \mu) = \mathcal{KL}(\rho_{-I} \otimes \bigotimes_{i \in I} \delta_{A_i}, \mathcal{D}_1 \otimes \mathcal{D}_2 \otimes \mathcal{D}_3)$$

$$= \mathcal{KL}(\rho_{-I}, \bigotimes_{i \in [3] \setminus I} \mathcal{D}_i) + \sum_{i \in I} \mathcal{KL}(\delta_{A_i}, \mathrm{Uniform}(\mathbb{S}_i))$$

$$= \mathcal{KL}(\rho_{-I}, \bigotimes_{i \in [3] \setminus I} \mathcal{D}_i) + \sum_{i \in I} \log|\mathbb{S}_i|. \tag{66}$$

Recap that for $i \in [3] \setminus I$, distribution $\mathcal{D}_i$ is the centered Gaussian with the covariance matrix $\sigma_i^2 I_{d_i l_i}$ up to the reshaping, so the density of $\bigotimes_{i \in I} \mathcal{D}_i$ is given by

$$\mu_{-I}((a_i)_{i \in [3] \setminus I}) = \prod_{i \in [3] \setminus I} \frac{\sigma_i^{-d_i l_i}}{(2\pi)^{d_i l_i/2}} \exp\left(-\frac{1}{2\sigma_i^2}\|a_i\|_{\mathrm{F}}^2\right).$$

Hence, we have

$$\mathcal{KL}(\rho_{-I}, \otimes_{i \in [3] \setminus I} \mathcal{D}_i) = \int_{\prod_{i \in [3] \setminus I} \mathbb{L}_i} \rho_{-I}((a_i)_{i \in [3] \setminus I})$$

$$\times \log\left[\frac{\prod_{i \in [3] \setminus I} \exp\left(\|a_i\|_{\mathrm{F}}^2/2\sigma_i^2 - \|a_i - A_i\|_{\mathrm{F}}^2/2\sigma_i^2\right)}{\mathrm{Pr}((G_1, G_2, G_3) \in \Upsilon)}\right] \prod_{i \in [3] \setminus I} \mathrm{d}a_i$$

$$= \log\frac{1}{\mathrm{Pr}((G_1, G_2, G_3) \in \Upsilon)} - \sum_{i \in [3] \setminus I} \frac{1}{2\sigma_i^2}\|A_i\|_{\mathrm{F}}^2 + \sum_{i \in [3] \setminus I} \frac{1}{\sigma_i^2}\langle\mathbb{E}\boldsymbol{\xi}^i, A_i\rangle,$$

where $\boldsymbol{\xi}^i$ is distributed as the $i$-th marginal of $(P, Q, R) \sim \rho_{A_1, A_2, A_3}$. Using (53), we get $\mathbb{E}\boldsymbol{\xi}^i = A_i$, so bound (51) implies

$$\mathcal{KL}(\rho_{-I}, \otimes_{i \in [3] \setminus I} \mathcal{D}_i) \leqslant \log 8 + \sum_{i \in [3] \setminus I} \frac{1}{2\sigma_i^2}\|A_i\|_{\mathrm{F}}^2$$

$$\leqslant \log 8 + \frac{1}{2}\sum_{i \in [3] \setminus I} l_i \mathbf{r}_i^2(\Sigma),$$

where we used the definition of $\sigma_i$ and the fact that $\|A_i\|_{\mathrm{F}}^2 \leqslant l_i \|A_i\|^2 \leqslant l_i$ for $i = 1, 3$. Then, bound (66) implies

$$\mathcal{KL}(\rho_{A_1,A_2,A_3}, \mu) \leqslant \log 8 + \frac{1}{2} \sum_{i \in [3] \setminus I} l_i \mathbf{r}_i^2(\Sigma) + \sum_{i \in I} \log |\mathbb{S}_i|$$

$$\leqslant \log 8 + \sum_{i=1}^{3} \min\{\mathbf{r}_i^2(\Sigma) \cdot l_i, \log |\mathbb{S}_i|\}. \tag{67}$$

**Step 5. Final bound.** Then, we substitute bounds (65),(67) into (46). It yields

$$\sup_{\substack{A_1 \in \mathbb{S}_1, \\ A_2 \in \mathbb{S}_2, A_3 \in \mathbb{S}_3}} \frac{1}{n} \sum_{i=1}^{n} \langle \widehat{\mathcal{E}} \times_3 A_3^\top \times_1 A_1^\top, A_2 \rangle \leqslant 2^{12} \lambda \omega^2 \|\Sigma\|^2$$

$$+ \frac{\log 8 + \sum_{i=1}^{3} \min\{\mathbf{r}_i(\Sigma) \cdot l_i, \log |\mathbb{S}_i|\} + \log \frac{1}{\delta}}{\lambda n}$$

with probability at least $1 - \delta$, provided $2^6 \lambda \omega \|\Sigma\| \leqslant 1$. Since $n \geqslant \sum_{i=1}^{3} \min\{\mathbf{r}_i^2(\Sigma) \cdot l_i, \log |\mathbb{S}_i|\} + \log(8/\delta)$, we can choose $\lambda$ as

$$\lambda = \frac{1}{2^6 \omega \|\Sigma\|} \sqrt{\frac{\sum_{i=1}^{3} \min\{\mathbf{r}_i^2(\Sigma) \cdot l_i, \log |\mathbb{S}_i|\} + \log(8/\delta)}{n}}.$$

It implies

$$\sup_{\substack{A_1 \in \mathbb{S}_1, \\ A_2 \in \mathbb{S}_2, A_3 \in \mathbb{S}_3}} \frac{1}{n} \sum_{i=1}^{n} \langle \widehat{\mathcal{E}} \times_3 A_3^\top \times_1 A_1^\top, A_2 \rangle \leqslant 2^7 \omega \|\Sigma\| \sqrt{\frac{\sum_{i=1}^{3} \min\{\mathbf{r}_i^2(\Sigma) \cdot l_i, \log |\mathbb{S}_i|\} + \log(8/\delta)}{n}}$$

with probability at least $1 - \delta$. This completes the proof. $\qquad\square$

# H ADDITIONAL EXPERIMENTS

## H.1 TENSOR-PRLS PSEUDOCODE

In this section, we give pseudocode for our version of PRLS adopted to order-3 tensors. See Algorithm 2.

---
**Algorithm 2:** PRLS Thresholding Algorithm

---
**Require:** Tensor $\mathcal{X} \in \mathbb{R}^{d_1 \times d_2 \times d_3}$, regularization parameters $\lambda_1$, $\lambda_2$
**Ensure:** Soft-thresholded tensor $\widehat{\mathcal{X}}$
    **Step 1: Mode-1 Unfolding and Thresholding**
1: Reshape initial tensor into matrix: $\mathcal{X}_{(1)} = \mathtt{m}_1(\mathcal{X})$
2: Perform SVD of matricization: $U, S, V^\top = \mathrm{SVD}(\mathcal{X}_{(1)})$
3: Apply soft-thresholding: $S' = \max(S - \lambda_1/2, 0)$
4: Combine soft-thresholded SVD into a matrix: $\widehat{\mathcal{X}}_{(1)} = U \cdot \mathrm{diag}(S') \cdot V^\top$
5: Reshape back into tensor: $\mathcal{X}' = \mathtt{m}_1^{-1}(\widehat{\mathcal{X}}_{(1)})$
    **Step 2: Mode-3 Unfolding and Thresholding**
6: Reshape new approximation into matrix: $\mathcal{X}_{(3)} = \mathtt{m}_3(\mathcal{X}')$
7: Perform SVD of matricization: $U, S, V^\top = \mathrm{SVD}(\mathcal{X}_{(3)})$
8: Apply soft-thresholding: $S' = \max(S - \lambda_2/2, 0)$
9: Combine soft-thresholded SVD into a matrix: $\widehat{\mathcal{X}}_{(3)} = U \cdot \mathrm{diag}(S') \cdot V^\top$
10: Set $\widehat{\mathcal{X}} = \mathtt{m}_3^{-1}(\widehat{\mathcal{X}}_{(3)})$

---

Table 5: Performance comparison of tensor decomposition algorithms for $n = 4000$. Relative errors were averaged over 16 repeats of the experiment, empirical standard deviation is given after $\pm$ sign. Best results are boldfaced.

| Metric | Algorithm | | |
|---|---|---|---|
| | Sample Mean | TT-HOSVD | HardTTh |
| Relative Error | $0.430 \pm 0.007$ | $0.105 \pm 0.008$ | $\mathbf{0.054 \pm 0.002}$ |
| Time (seconds) | $0.0039 \pm 0.0015$ | $0.64 \pm 0.15$ | $3.2 \pm 3.3$ |

| Metric | Algorithm | | |
|---|---|---|---|
| | Tucker | Tucker+HOOI | PRLS |
| Relative Error | $0.105 \pm 0.007$ | $\mathbf{0.054 \pm 0.002}$ | $0.217 \pm 0.015$ |
| Time (seconds) | $30.7 \pm 3.9$ | $51.5 \pm 3.9$ | $0.8 \pm 1.1$ |

## H.2    EXTRA EXPERIMENTS ON COVARIANCE ESTIMATION

Here we study the performance of tensor decomposition algorithms in the setup of Section 3. First, we repeat experiments of Section 3 for $n = 4000$, see Table 5.

Second, we study the dependence of $\sin\Theta$-distance of estimated singular subspaces to singular subspaces of matricizations of $\mathcal{T}^*$ on the number of iterations $T$ and the sample size $n$. Matrices $\widehat{U}_0, \widehat{U}_T, \widehat{V}_0, \widehat{V}_T$ are defined in Algorithm 1. As before, the number of additional iterations is taken 10. The results are presented in Table 6.

Table 6: The study of $\sin\Theta$-distance from estimated singular subspaces to singular subspaces of matricizations of $\mathcal{R}(\Sigma)$. Average errors and standard deviations are obtained after 16 repeats of the experiment. The setup is defined in Section 3.

| | $n = 500$ | $n = 2000$ | $n = 5000$ | $n = 6000$ | $n = 7000$ |
|---|---|---|---|---|---|
| $\sin\Theta(\operatorname{Im}\widehat{U}_0, \operatorname{Im}U^*)$ | $1.0 \pm 0.0$ | $1.0 \pm 0.0$ | $0.8 \pm 0.3$ | $0.8 \pm 0.2$ | $0.6 \pm 0.3$ |
| $\sin\Theta(\operatorname{Im}\widehat{V}_0, \operatorname{Im}V^*)$ | $1.0 \pm 0.0$ | $1.0 \pm 0.0$ | $1.0 \pm 0.0$ | $0.90 \pm 0.14$ | $0.9 \pm 0.2$ |
| $\sin\Theta(\operatorname{Im}\widehat{U}_T, \operatorname{Im}U^*)$ | $1.0 \pm 0.0$ | $0.33 \pm 0.08$ | $0.17 \pm 0.04$ | $0.13 \pm 0.03$ | $0.13 \pm 0.02$ |
| $\sin\Theta(\operatorname{Im}\widehat{V}_T, \operatorname{Im}V^*)$ | $1.0 \pm 0.0$ | $0.46 \pm 0.17$ | $0.21 \pm 0.03$ | $0.18 \pm 0.05$ | $0.17 \pm 0.02$ |

For scalability study we increase the number of parameters from $10^6$ to $7.4 \cdot 10^8$ for 1000 samples. One can see that our methods scales successfully, even winning comparison with Tucker+HOOI. The results are shown in Table 7. Next, we increase number of parameters up to $4 \cdot 10^9$ for 1000 and 2000 samples. Unfortunately, Tucker+HOOI does not show ability for scaling due to enormous time overhead, so results in Table 8 are provided excluding it.

We provide ablation study on the effect of ranks on the error rate. We expect that large increase of ranks leads to broken spectral gap condition, thus, models takes part of the noise as vital information. Large decrease leads to loss of vital information, since relevant singular values may be erased. Despite that, small perturbation in ranks may lead to better bias-variance tradeoff, thus, decreasing error overall. See Figure 2 for details.

## H.3    EXPERIMENTS ON TENSOR ESTIMATION

This section is devoted to experiments that did not have enough space in the main text. In particular, we numerically study the impact of additional iterations of Algorithm 1 in the tensor estimation problem. We do not consider the misspecified case, and, given $(J, K)$ and $p, q, r$, generate $\mathcal{T}^*$ as follows. First, we generate matrices $U_j, W_{jk}, V_k$ from model (5) according to the matrix initialize method - random, random symmetric, symmetric with special spectrum decay (i.e. inverse quadratic,

Table 7: Performance comparison of tensor decomposition algorithms for $n = 1000$, $p = q = r = 30$. Best results are boldfaced.

| Metric | Algorithm | | |
| --- | --- | --- | --- |
| | Sample Mean | TT-HOSVD | HardTTh |
| Relative Error | 4.448 | 0.216 | **0.065** |
| Time (seconds) | 3.504 | 867.756 | 1007.069 |

| Metric | Algorithm | | |
| --- | --- | --- | --- |
| | Tucker | Tucker+HOOI | PRLS |
| Relative Error | 0.192 | 0.110 | 4.422 |
| Time (seconds) | 14601.442 | 31256.665 | 703.230 |

Table 8: Performance comparison of tensor decomposition algorithms for $p = q = r = 40$. Best results are boldfaced.

| Metric (n = 1000) | Algorithm | | | | |
| --- | --- | --- | --- | --- | --- |
| | Sample Mean | TT-HOSVD | HardTTh | Tucker | PRLS |
| Relative Error | 6.86 | 0.21 | **0.055** | 0.19 | 6.82 |
| Time (seconds) | 20.94 | 5095.54 | 6873.25 | 84360.27 | 5872.09 |

| Metric (n = 2000) | Algorithm | | | | |
| --- | --- | --- | --- | --- | --- |
| | Sample Mean | TT-HOSVD | HardTTh | Tucker | PRLS |
| Relative Error | 4.87 | 0.19 | **0.038** | 0.1845 | 4.83 |
| Time (seconds) | 20.63 | 5839.20 | 6889.42 | 84476.38 | 5825.16 |

exponential, linear, etc.). We will refer to these matrices $U_j, W_{jk}, V_k$ as sub-components of matrix

$$S = \sum_{j=1}^{J} \sum_{k=1}^{K} U_j \otimes W_{jk} \otimes V_k \in \mathbb{R}^{pqr \times pqr},$$

and reshape it to a tensor $\mathcal{T}^* = \mathcal{R}(S)$. It is ease to see that such procedure is equivalent to the direct assignment of TT factors, due to Equation (8). Then, choosing a noise level $\sigma$, we generate a noise tensor $\widehat{\mathcal{E}}$ as a random normal with $\sigma$ as its standard deviation and compute

$$\mathcal{Y} = \mathcal{T}^* + \widehat{\mathcal{E}}.$$

Our code supports some other testing regimes: one can choose the $S$ structure directly (block-Toeplitz, structure (1), etc.) supporting misspecification case, and rank selection method (via hard thresholding, effective rank, absolute error). For more information on rank selection see display (13).

For the specific experiment, we vary the algorithms to test, as well as the actual ranks and sizes of the components $U_j, W_{jk}, V_k$. For PRLS algorithm, due to its special setup, we tune $\lambda_1, \lambda_2$ parameters on a log-scale. In the Table 9 one can see, that our method also shows less variance, compared to the previous algorithms, such as sample mean or Algorithm 2 with noise variance equal to 0.3.

Now consider the case of a low SNR setting (high-noise regime, fast spectrum decay). This case violates the assumptions of Theorem 2.2. It can be seen that the methods perform poorly and do not restore the signal (the relative error remains at the level of 0.3), thus, demonstrating the necessity of theorem's conditions. The experiment below was conducted for the case when sub-components of $S$ spectra decrease as inverse square sequence (see Table 10 for details).

It may be useful to examine the spectrum of matrix $S$ and matricizations in order to understand how the behavior of algorithms varies in different scenarios. Figure 3 illustrates this. These plots were

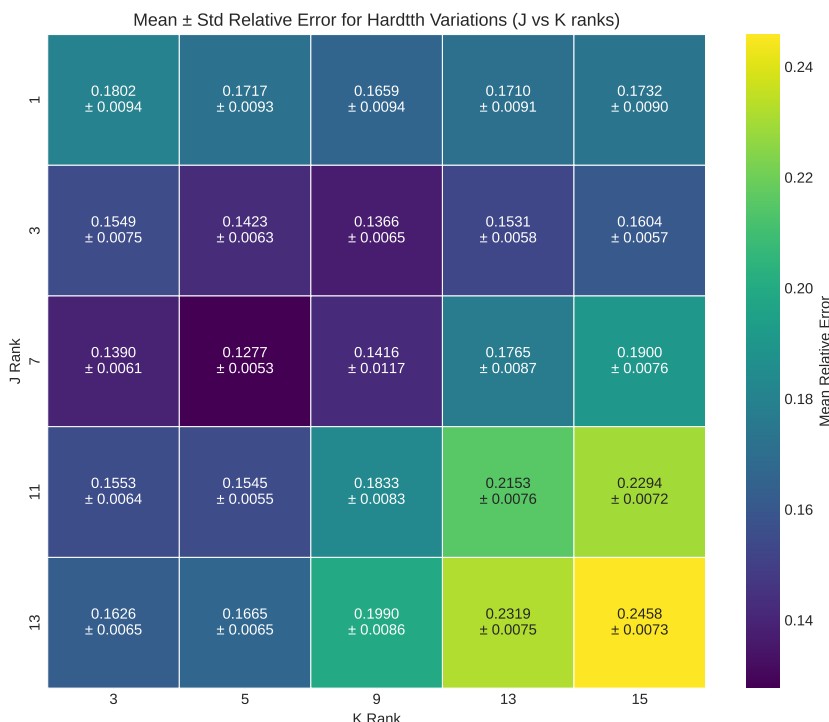

Figure 2: Rank ablation study for covariance with parameters $(J, K) = (7, 9)$, $p = q = r = 10$, averaged by 32 runs.

Table 9: Performance comparison of tensor decomposition algorithms under medium noise conditions. The best results are boldfaced.

| Metric | Algorithm | | |
|---|---|---|---|
| | Sample Mean | TT-HOSVD | HardTTh |
| Relative Error | $0.3643 \pm 0.0135$ | $0.0449 \pm 0.0018$ | $\mathbf{0.0357 \pm 0.0015}$ |
| Time (seconds) | $0.0204 \pm 0.0096$ | $4.4732 \pm 1.8079$ | $7.5522 \pm 2.1386$ |

| Metric | Algorithm | | |
|---|---|---|---|
| | Tucker | Tucker+HOOI | PRLS |
| Relative Error | $0.0439 \pm 0.0016$ | $\mathbf{0.0357 \pm 0.0015}$ | $0.1130 \pm 0.0037$ |
| Time (seconds) | $56.7830 \pm 16.3132$ | $106.5766 \pm 25.2531$ | $0.7076 \pm 0.1160$ |

constructed for tensor-train rank $(J, K)$ pairs of 7 and 9, respectively, with sub-components having a size of $10 \times 10$. The total matrix size was $1000 \times 10000$, composed of these sub-components.

To experimentally confirm the necessity of the conditions of our theorem, we plotted the relationship between singular values and noise levels, as well as the relative error and noise levels. Our findings indicate that, after a certain threshold, our algorithm no longer effectively mitigate noise but instead overfit to it, resulting in inferior performance compared to one-step methods such as TT-HOSVD (see Figure 4).

Table 10: Performance of tensor decomposition algorithms under inverse quadratic decay of spectrum. In case of low SNR we observe that iterative methods perform worse than one-shot and both do not restore signal. The best result is boldfaced.

| Metric | Algorithm | | |
|--------|-----------|---|---|
| | Sample Mean | TT-HOSVD | HardTTh |
| Relative Error | $0.3508 \pm 0.0004$ | $\mathbf{0.0251 \pm 0.0001}$ | $0.0279 \pm 0.0003$ |
| Time (seconds) | $0.0509 \pm 0.0166$ | $13.9748 \pm 4.1845$ | $282.7375 \pm 145.8327$ |

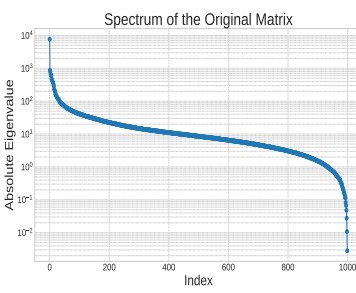

(a) Matrix $S$ spectrum

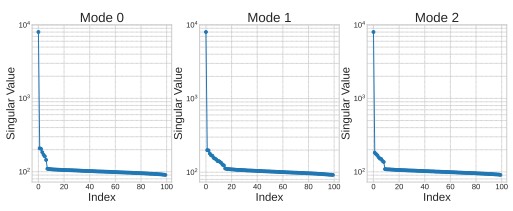

(b) Singular values of matricizations

Figure 3: Spectrum of the objectives in case of random sub-components. As one can see, dense spectrum of matrix $S$ with noise become separable for matricizations.

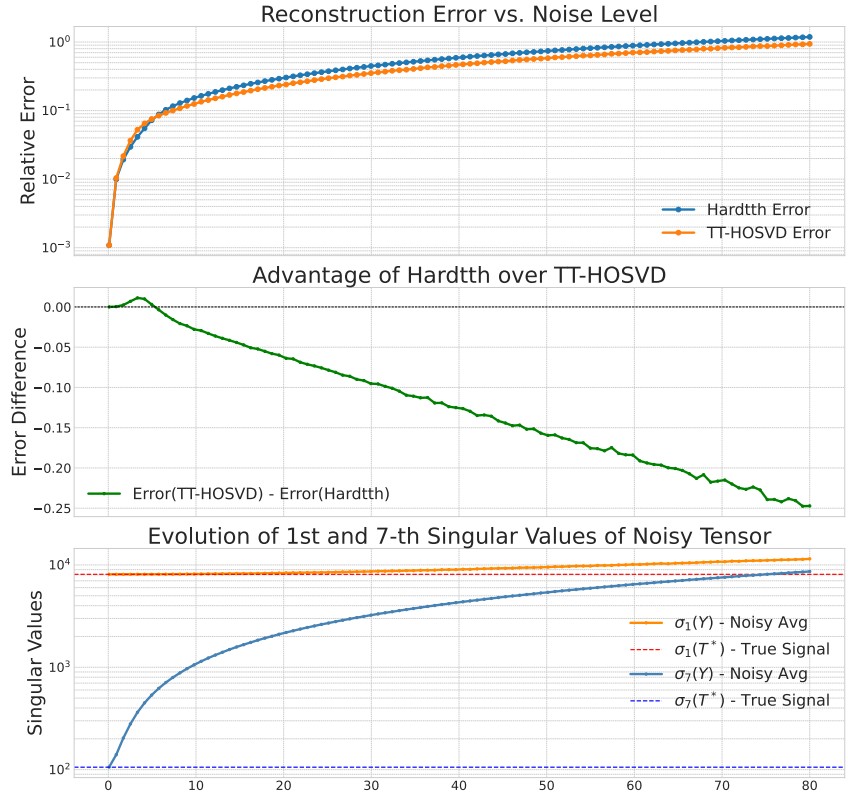

Figure 4: Performance of tensor decomposition algorithms and spectrum behavior under noise increase.

