# OpenReview forum: "Structured covariance estimation via tensor-train decomposition"
_ICLR.cc/2026/Conference — Submitted to ICLR 2026_

### Official Review · Reviewer_XWb1 · 2025-10-28

**Soundness:** 2
**Presentation:** 1
**Contribution:** 2
**Rating:** 2
**Confidence:** 3

**Summary:**

This paper extends previous approaches based on low-rank tensor-structured models for covariance matrix estimation. Specifically, it introduces a Tucker-2 structured covariance matrix model, as per (5), which has a higher degree of flexibility than the existing sum-of-Kronecker-products model (2).

It then resorts to a quite standard high-order orthogonal iteration (HOOI) algorithm (Algorithm 1) initialized by a truncated high-order SVD (HOSVD), with respect to two modes (in line with the Tucker-2 structure).

Its main result (Theorem 2.2) is then a bound on the quadratic estimation error (with respect to the true covariance) made by Algorithm 1. This bound and the conditions of Theorem 2.2 are stated without reference to the ambient space dimension, but rather involve three introduced quantities ($r_1, r_2, r_3$) which are said to play the role of "effective dimensions".

A set of numerical experiments is described in the paper, but only involving synthetic, randomly generated covariance matrices having the postulated structure by construction.

**Strengths:**

The proposal of new, richer tensor model structures for learning high-dimensional covariance matrices is certainly of some interest, as well as deriving dimension-free guarantees regarding the approximation error delivered by a simple approximation scheme.

**Weaknesses:**

1) My main concern is that, in spite of the interest of its proposal, this paper suffers from serious readability and clarity issues:
- When referring to the model, there is a constant ambiguity between Tensor Train (TT) and Tucker-2 structure, with the terminology going back-and-forth along the paper, which does not help. I don't see why one should use the former terminology, since the model is quite clear a Tucker-2 model with two matrix factors and a core tensor. This is how it is written in (9) and how Algorithm 1 is devised. If the authors wish to point to a connection with the TT model, I suggest that they use some remark to do it, but keep the discussion focused on the Tucker-2 structure for clarity.
- The claim that the CP is a special case of (5) is far-fetched, since in (5) if one eliminates one of the summations (say, setting $J=1$), then one gets a special CP in which the first vector is repeated in all terms. Moreover, this claim is somewhat misleading in the sense that it might lead the reader to think that an algorithm capable of estimating a model with the structure (5) would also be able to handle the CP as a special case. The TT model cannot be said to be a more general model than CP, it is simply a different one. In fact, it turns out that several different tensor models can be thought each one as a special case of the other (CPD, Tucker, BTD, ...), but in terms of actually estimating these models, the algorithms are specialized, and the properties and issues faced in practice are not the same.
- The multilinear transformation (such as in (9)) is not denoted as usual (with the core tensor coming first, and then the transforming matrices).
- Line 240 refers to a randomized truncated SVD, while the operator SVD$_J$ is defined before as a standard truncated SVD. This causes confusion, since the paper never discusses the use of a randomized SVD, and possible errors due to randomization do not seem to be taken into account in the analysis.
- The statement of Theorem 2.2 is very cumbersome and hard to parse and interpret. I believe that further work is needed to simplify/distillate as much as possible that statement and thereby render it more accessible/useful.
- What is "the variance part" of $\tilde{r}_T$ mentioned in line 305?
- Why is homoscedastic used to mean i.i.d. in line 336, when it actually means "of equal variance"?
- It is hard to conclude anything from the discussion on the choice of $J$ and $K$. Are those given estimators practical? How can $C'$ be inferred?
- Why are singular values referred to as "singular numbers"? This terminology sounds very unusual to me.

2) Algorithm 1 is not novel at all, and cannot be claimed to be proposed by the authors, as done in the Conclusion. This algorithm is in fact just a standard high-order orthogonal iteration (HOOI) applied to a Tucker-2 model for estimating its two factors and core, and initialized in a very usual way by means of a truncated high-order SVD.

3) The contents of the numerical experiments are insufficient in my view, mainly because only random synthetic examples matching the postulated model are considered. This is of course useful for measuring the error with respect to the ground truth, but the whole motivation discussed in the Introduction is to introduce model (5) for greater flexibility with respect to previous approaches. Hence, a comparison with other models using real-world sample covariance matrices should be given.

4) Finally, a careful revision is needed regarding English usage. Some examples: "yielding rough variance proxy..." (line 324); "impact of ... to the perturbation" (lines 337-341); "for $n$ one has bounds" (line 360); "which diagonal" (line 395).

**Questions:**

1) How should the condition stated in (11) be compared to that in your result?

2) I do not understand the claim "This highlights the difference between low-rank tensor estimation problem and low-rank matrix estimation problem, since for the latter there is no significant difference between soft-thresholding and hard-thresholding estimation." Estimators relying on soft- and hard-thresholding singular values can behave quite differently depending on the example at hand. What do you mean exactly here?

---

> ### Author Response · Authors · 2025-11-21
>
> *1. How should the condition stated in (11) be compared to that in your result?*
>
> Our condition can be milder if effective dimensions are much smaller than ambient dimensions, see added Proposition 2.3 and the discussion below. But they are incomparable in general. However, one can hope to develop a procedure of polynomial complexity that works under milder conditions on the spectral gap than ours.
>
> *2. I do not understand the claim "This highlights the difference between low-rank tensor estimation problem and low-rank matrix estimation problem, since for the latter there is no significant difference between soft-thresholding and hard-thresholding estimation." Estimators relying on soft- and hard-thresholding singular values can behave quite differently depending on the example at hand. What do you mean exactly here?*
>
> For the low-rank matrix estimation, both soft-thresholding and hard-thresholding achieve minimax optimal rates. For our model, one should expect that replacing hard-thresholding with soft-thresholding dramatically affects the performance of the algorithm on average. One should expect this type of behavior for any HOOI-like procedure.
>
> *3. Algorithm 1 is not novel at all, and cannot be claimed to be proposed by the authors, as done in the Conclusion. This algorithm is in fact just a standard high-order orthogonal iteration (HOOI) applied to a Tucker-2 model for estimating its two factors and core, and initialized in a very usual way by means of a truncated high-order SVD.*
>
> In the revised version, we do not insist on particular novelty of our approach and focus on theoretical contribution of our paper.
>
> *4. The contents of the numerical experiments are insufficient in my view, mainly because only random synthetic examples matching the postulated model are considered. This is of course useful for measuring the error with respect to the ground truth, but the whole motivation discussed in the Introduction is to introduce model (5) for greater flexibility with respect to previous approaches. Hence, a comparison with other models using real-world sample covariance matrices should be given.*
>
> Our motivation was to introduce a statistical model, which, on one hand, was convenient for theoretical analysis, and, on the other hand, included the widespread CANDECOMP/PARAFAC model as a particular case. Then dimension-free bounds for our model automatically yield similar bounds for CANDECOMP/PARAFAC. We would like to emphasize previous dimension-free bounds in tensor estimation were valid only for simple rank-one tensors. For this reason, our paper contributes significantly to the field of high-dimensional statistics.
>
> *5. Finally, a careful revision is needed regarding English usage. Some examples: "yielding rough variance proxy..." (line 324); "impact of ... to the perturbation" (lines 337-341); "for $n$ one has bounds" (line 360); "which diagonal" (line 395).*
>
> Thank you for pointing out the clarity issues. We have made the corresponding corrections.
>
> *6. My main concern is that, in spite of the interest of its proposal, this paper suffers from serious readability and clarity issues: \newline \newline
> When referring to the model, there is a constant ambiguity between Tensor Train (TT) and Tucker-2 structure, with the terminology going back-and-forth along the paper, which does not help. I don't see why one should use the former terminology, since the model is quite clear a Tucker-2 model with two matrix factors and a core tensor. This is how it is written in (9) and how Algorithm 1 is devised. If the authors wish to point to a connection with the TT model, I suggest that they use some remark to do it, but keep the discussion focused on the Tucker-2 structure for clarity.*
>
> Thank you for the remark. We do agree that Tucker-2 structure coincides with the TT decomposition in the 3-dimensional case. We chose the TT decomposition because we believe that the general audience is more familiar with this name, which helps position our work for them. However, to avoid confusion, we suggest making the title more specific by emphasizing the 3-dimensional case: "Structured covariance estimation via tensor-train decomposition in three dimensions".

---

> ### Author Response · Authors · 2025-11-21
>
> *The claim that the CP is a special case of (5) is far-fetched, since in (5) if one eliminates one of the summations (say, setting ), then one gets a special CP in which the first vector is repeated in all terms. Moreover, this claim is somewhat misleading in the sense that it might lead the reader to think that an algorithm capable of estimating a model with the structure (5) would also be able to handle the CP as a special case. The TT model cannot be said to be a more general model than CP, it is simply a different one. In fact, it turns out that several different tensor models can be thought each one as a special case of the other (CPD, Tucker, BTD, ...), but in terms of actually estimating these models, the algorithms are specialized, and the properties and issues faced in practice are not the same.*
>
> To derive the proper CP decomposition one should take $W_{jk}$ as zero matrices whenever $j \neq k$. Through this derivation, we aimed to show that our proposed model allows the result from Theorem 2.2 to be applied to the CP decomposition as well, thereby deriving dimension-free bounds for it, too.
>
> *The multilinear transformation (such as in (9)) is not denoted as usual (with the core tensor coming first, and then the transforming matrices).*
>
> We have made the respective changes in the revision.
>
> *Line 240 refers to a randomized truncated SVD, while the operator SVD is defined before as a standard truncated SVD. This causes confusion, since the paper never discusses the use of a randomized SVD, and possible errors due to randomization do not seem to be taken into account in the analysis.*
>
> We changed the discussion of algorithm complexity by replacing the randomized SVD complexity with that of exact deterministic SVD. We also mentioned the possible usage of approximate algorithms in practice.
>
> *The statement of Theorem 2.2 is very cumbersome and hard to parse and interpret. I believe that further work is needed to simplify/distillate as much as possible that statement and thereby render it more accessible/useful.*
>
> We replaced the bias term with a less accurate but simpler upper bound, see line 272 in the revision.
>
> *What is "the variance part" of $\tilde  r_T$ mentioned in line 305?*
>
> That had been corrected, see lines 300-305 in the revision.
>
> *Why is homoscedastic used to mean i.i.d. in line 336, when it actually means "of equal variance"?*
>
>  We removed the adjective ``homoscedastic''.
>
> *It is hard to conclude anything from the discussion on the choice of J and K. Are those given estimators practical? How can C' be inferred?*
>
> $C'$ can be inferred from the concentration inequalities provided in the appendix, see Lemmas E.1 and E.2. In our case, $C'$ will be roughly several thousands. However, theoretical analysis usually overestimates absolute constants.
>
> Overestimation of $J$ could lead to much worse theoretical guarantees involving the product of effective dimensions in the convergence rate, and $\widehat{J}$ prevent this under the spectral gap assumptions. However, applying HardTTh with $J = J' \le \widehat{J}$ can result in a better bias-variance tradeoff. We conducted rank ablation studies (see Figure 2 on page 40) and the results confirmed that.
>
> *Why are singular values referred to as "singular numbers"? This terminology sounds very unusual to me.*
>
> We are grateful for your attention and comments, which have been addressed in the revision.

---

### Official Review · Reviewer_GRns · 2025-11-01

**Soundness:** 4
**Presentation:** 4
**Contribution:** 3
**Rating:** 8
**Confidence:** 3

**Summary:**

The paper studies high-dimensional covariance estimation under a flexible structural assumption that the covariance matrix can be well approximated by a sum of double Kronecker products arranged in a tensor-train (TT) format. Concretely, the model writes $\Sigma$ as a sum over factors $U_j\otimes V_{jk}\otimes W_k$ with TT ranks $(J, K)$, which strictly generalizes classical Kronecker-product and CP covariance models and allows richer cross-mode interactions. The authors show that $\Sigma$ can be rearranged into a third-order tensor $\mathcal{R}(\Sigma)$ that admits a Tucker-2 structure, and they propose a polynomial-time estimator based on iterative truncated SVDs on the mode-1 and mode-3 unfoldings. The algorithm, named HardTTh (Hard Tensor-Train Thresholding), alternates subspace updates via rank-$J$ and rank-$K$ truncated SVDs and then projects the observed tensor onto the estimated subspaces.

On the theoretical side, under a mild sub-exponential-type assumption on the standardized data (a Hanson–Wright-style moment generating function bound), the paper derives non-asymptotic, high-probability, dimension-free Frobenius error bounds for the estimator. The bound separates into a bias term (capturing model misspecification), a leading variance term that scales as $\omega||\Sigma||$ times a sum of effective dimensions divided by $n$, and smaller remainder terms that decay with the number of iterations. The effective dimensions are defined via norms of partial traces of $\Sigma$ and capture intrinsic complexity of the covariance structure, analogously to effective rank in unstructured covariance estimation. The analysis also improves on prior Kronecker-sum results by expressing the variance in terms of $||\Sigma||$, and it explicitly treats misspecification. Experiments on synthetic data show that HardTTh substantially improves over one-shot TT-HOSVD when spectral gaps are reasonable and the sample size is moderate-to-large, and it attains Tucker+HOOI accuracy with noticeably lower runtime. The results also illustrate that when spectral conditions fail (e.g., small $n$ or weak signal), iterative refinement does not help, which matches the theory’s requirements.

**Strengths:**

Originality. The paper introduces a well-motivated TT-structured covariance model that subsumes widely used Kronecker and CP models while enabling richer multiway dependencies. It proposes an efficient HOOI-like estimator specialized to Tucker-2/TT covariance structure, and it develops a nontrivial analysis that yields dimension-free guarantees based on partial-trace effective dimensions. This combination of modeling, algorithmic design, and analysis is novel and meaningful.

Quality. The theoretical development is careful and modular. The authors first establish a deterministic perturbation bound for their iterative SVD procedure and then derive high-probability error bounds using a PAC–Bayes approach tailored to the structured noise arising from sample covariance tensors. The bias–variance decomposition is explicit, misspecification is handled cleanly, and the variance term scales with interpretable effective dimensions. The algorithmic complexity is analyzed, showing practical scaling with truncated randomized SVDs. The experiments are designed to probe when additional iterations help, how performance varies with sample size, and why soft-thresholding (PRLS) can be suboptimal in multi-mode settings.

Clarity. The paper uses heavy notation, but it is consistent and well-documented. The rearrangement operator and mode-wise unfoldings are clearly defined, and the key algebraic identities are stated. The main theorem is presented with a helpful collection of ancillary terms. The proof sketch guides the reader through the argument, and the appendices provide the necessary technical details.

Significance. The work bridges an important gap by providing a computationally efficient covariance estimator for multiway structured models with non-asymptotic, dimension-free guarantees beyond simple rank-one settings. It contributes a practical alternative to CP/nuclear-norm approaches that face computational hardness, and it extends effective-dimension ideas from unstructured covariance estimation to a richer tensor setting. Given the prevalence of multiway data (e.g., spatio-temporal, multi-sensor, MIMO), the proposed framework has the potential for notable impact.

**Weaknesses:**

1. The main theoretical guarantees rely on lower bounds on the $J$-th and $K$-th singular values of the mode-1 and mode-3 matricizations of $\mathcal{R}(\Sigma)$. While these conditions are standard in tensor estimation, they may be restrictive in some realistic regimes. The paper acknowledges this statistical–computational gap, but it would benefit from more actionable guidance on how practitioners can assess these conditions from data.

2. The analysis indicates that the noise structure induced by sample covariance is heteroscedastic, and that an ideal version of the algorithm would include a debiasing step before applying SVD. The present work leaves such debiasing to future research. Even a pragmatic approximate debiasing scheme could strengthen the practical contribution.

3. The proposed data-driven selection of $J$ and $K$ depends on $\omega$ and on concentration of partial traces, and the paper defers establishing the needed concentration bounds. An empirical study examining the stability and accuracy of the rank selection rule would make the method more turnkey.

4. All experiments are synthetic. Although they are informative, adding at least one real-data example would improve the credibility and applicability of the method and help illustrate interpretability of the learned structure.

5. The effective dimensions $r_i(\Sigma)$ are insightful but somewhat abstract. The paper could provide more intuition through examples where $r_i(\Sigma)$ is demonstrably much smaller than ambient dimensions, helping readers anticipate when the dimension-free rates will be most favorable.

**Questions:**

1. How robust is HardTTh to mild violations of the spectral gap conditions? It would be helpful to include experiments where the signal-to-noise ratio is near the threshold and to quantify the degradation of the subspace estimates and final error.

2. For the rank selection scheme, $\omga$ may be unknown. Do you suggest estimating $\omega$ from the data (e.g., via Gaussian calibration of quadratic forms or via residual-based methods), or using data splitting or bootstrap techniques to tune the thresholds? An empirical comparison between fixed ranks and the proposed selection (reporting rank recovery rates and sensitivity to n) would be very helpful.

3. The bias term involves suprema over orthonormal $U$ and $V$ of the projected misspecification energy. Can you provide more interpretable sufficient conditions that translate into explicit bias rates, such as tail bounds on singular values of the mode unfoldings or decay conditions on TT cores?

4. In practice, when should one prefer the proposed TT covariance model over CP or pure Kronecker models? Brief guidelines contrasting modeling flexibility, sample complexity, and computational demands across these alternatives would aid practitioners.

5. Do your methods and theory extend cleanly to orders higher than three (e.g., four-way TT covariance)? Which parts of the analysis (partial trace definitions, PAC–Bayes steps, recursion) generalize directly, and where would new technical work be needed?

6. In the Gaussian setting, can sharper constants or reduced logarithmic factors be achieved using exact Hanson–Wright or Gaussian isoperimetric inequalities? If so, would this change the iteration complexity needed for the remainder terms to be dominated?

7. On the empirical side, it would strengthen the paper to add a real-data case study (even small-scale), report measured proxies of the effective dimensions, and provide ablations on the number of iterations, randomized SVD settings, and misspecified ranks to guide practical deployment.

---

> ### Author Response · Authors · 2025-11-21
>
> *1. How robust is HardTTh to mild violations of the spectral gap conditions? It would be helpful to include experiments where the signal-to-noise ratio is near the threshold and to quantify the degradation of the subspace estimates and final error.*
>
> We provide such experiments in Appendices H.2 and H.3 (see Table 6 and Figure 4).
>
> *2. For the rank selection scheme, $\omega$
> may be unknown. Do you suggest estimating $\omega$ from the data (e.g., via Gaussian calibration of quadratic forms or via residual-based methods), or using data splitting or bootstrap techniques to tune the thresholds? An empirical comparison between fixed ranks and the proposed selection (reporting rank recovery rates and sensitivity to n) would be very helpful.*
>
> We do not suggest estimating $\omega$, this parameter is primarily used for quantification of the estimation error. Our rank selection procedure works if we have some prior information about $\omega$. Without any additional information about data distribution, it is very hard to select proper ranks $J$ and $K$. This concerns not only our method but many other procedures.
>
> *3. The bias term involves suprema over orthonormal $U$ and $V$ of the projected misspecification energy. Can you provide more interpretable sufficient conditions that translate into explicit bias rates, such as tail bounds on singular values of the mode unfoldings or decay conditions on TT cores?*
>
> We replaced the bias term with a less accurate but simpler upper bound, see line 272 in the revision.
>
> *4. In practice, when should one prefer the proposed TT covariance model over CP or pure Kronecker models? Brief guidelines contrasting modeling flexibility, sample complexity, and computational demands across these alternatives would aid practitioners.*
>
> The best choice heavily depends on a particular task. Your question concerns the problem of model selection which goes beyond the scope of the present submission. We focus on rigorous theoretical guarantees on estimation accuracy once a model is fixed.
>
> *5. Do your methods and theory extend cleanly to orders higher than three (e.g., four-way TT covariance)? Which parts of the analysis (partial trace definitions, PAC–Bayes steps, recursion) generalize directly, and where would new technical work be needed?*
>
> Our technique can be straightforwardly generalized for higher-order Tucker decomposition, assuming stronger conditions on spectral gaps. The case of TT-format is more sophisticated, and it is hard to justify which parts of our theory can be used.
>
> *6. In the Gaussian setting, can sharper constants or reduced logarithmic factors be achieved using exact Hanson–Wright or Gaussian isoperimetric inequalities? If so, would this change the iteration complexity needed for the remainder terms to be dominated?*
>
> Gaussian isoperimetric inequalities may lead to sharper constants but it is unlikely that they will lead to significant improvements in the iteration complexity.
> Nevertheless, it is an interesting question for future research.
>
> *7. On the empirical side, it would strengthen the paper to add a real-data case study (even small-scale), report measured proxies of the effective dimensions, and provide ablations on the number of iterations, randomized SVD settings, and misspecified ranks to guide practical deployment. All experiments are synthetic. Although they are informative, adding at least one real-data example would improve the credibility and applicability of the method and help illustrate interpretability of the learned structure.*
>
> We have added experiments on image denoising, compared to other tensor-like methods. In addition, we computed effective dimensions for network on MNIST dataset, see discussion below. One should mention that several pipelines use covariance estimation as a step inside, but estimating the total quality changing only one step is not fair - additional block tuning is needed. Nevertheless, one can find works related to Fisher matrix or tensor completion below [Chekalina, et al., 2025], [Han, Willett, Zhang, 2022].
>
> [Chekalina, et al., 2025] Chekalina, Viktoriia, et al. "Generalized Fisher-Weighted SVD: Scalable Kronecker-Factored Fisher Approximation for Compressing Large Language Models." arXiv preprint arXiv:2505.17974 (2025).
>
> [Han, Willett, Zhang, 2022] Han, Rungang, Rebecca Willett, and Anru R. Zhang. "An optimal statistical and computational framework for generalized tensor estimation." The Annals of Statistics 50.1 (2022): 1-29.

---

> ### Author Response · Authors · 2025-11-21
>
> *The main theoretical guarantees rely on lower bounds on the J-th and K-th singular values of the mode-1 and mode-3 matricizations of $\mathcal{R}(\Sigma)$. While these conditions are standard in tensor estimation, they may be restrictive in some realistic regimes. The paper acknowledges this statistical–computational gap, but it would benefit from more actionable guidance on how practitioners can assess these conditions from data.*
>
> The spectral gap condition is a fundamental limitation. Its violation leads to dramatic performance decrease. Once the ranks $J, K$ are fixed, one can estimate the spectral gap via empirically obtained singular values.
>
> *The analysis indicates that the noise structure induced by sample covariance is heteroscedastic, and that an ideal version of the algorithm would include a debiasing step before applying SVD. The present work leaves such debiasing to future research. Even a pragmatic approximate debiasing scheme could strengthen the practical contribution.*
>
> One could try to estimate the mean of $\mathtt{m}_1(\mathcal{E}) \mathtt{m}_1(\mathcal{E}) ^\top$ in the decomposition of $\mathtt{m}_1(\mathcal{Y}) \mathtt{m}_1(\mathcal{Y})^\top$ by
>
> $S = \frac{1}{n^2} \sum_{i = 1}^n \mathtt{m}_1(\mathcal{R}(\widehat{\Sigma} - \mathbf{X}_i \mathbf{X}_i^\top)) \mathtt{m}_1(\mathcal{R}(\widehat{\Sigma} - \mathbf{X}_i \mathbf{X}_i^\top))^\top$, and take the truncated SVD of
>
> $\mathtt{m}_1(\mathcal{\mathcal{Y}}) \mathtt{m}_1(\mathcal{\mathcal{Y}}) ^\top - S$ to estimate the left singular vectors of $\mathtt{m}_1(\mathcal{X})$. However, we conducted several experiments and did not obtain any advantage compared to the algorithm without debiasing.
>
> *The proposed data-driven selection of J and K depends on $\omega$ and on concentration of partial traces, and the paper defers establishing the needed concentration bounds. An empirical study examining the stability and accuracy of the rank selection rule would make the method more turnkey.*
>
> We have added experiments that can be found on the page 40 (see Figure 2).
>
> *The effective dimensions are insightful but somewhat abstract. The paper could provide more intuition through examples where is demonstrably much smaller than ambient dimensions, helping readers anticipate when the dimension-free rates will be most favorable.*
>
> We added Proposition 2.3 that bounds the effective dimensions using effective ranks of the Kronecker factors. Moreover, we conducted experiments with effective ranks. We considered a 3-layer model with ReLU activation and 5070 parameters on MNIST. After training it for 10 epochs, we constructed its Fisher matrix over entire training dataset, which consists of 60000 objects. Assuming $p,q,r= 13, 15, 26$ we have computed effective ranks and have noticed, that for dimension 5070, which is equal to the parameter size, effective dimensions are nearly $0.67, 5, 20$ respectively.

---

### Official Review · Reviewer_nFb5 · 2025-11-01

**Soundness:** 3
**Presentation:** 3
**Contribution:** 2
**Rating:** 4
**Confidence:** 3

**Summary:**

The paper develops a new structured covariance estimation framework based on tensor-train (TT) decomposition, extending Kronecker and CP-type models. It proposes a polynomial-time algorithm, HardTTh, and provides dimension-free non-asymptotic error bounds under assumption 2.1, in particular under sub-Gaussian assumptions. Experiments on synthetic data show performance improvements over Tucker/HOOI and PRLS baselines.

**Strengths:**

**Originality:**

Combining TT decomposition with covariance modeling is mathematically elegant.

Rigorous derivation of dimension-free Frobenius error bounds, uncommon in tensor-structured covariance literature.

**Quality:**

Theoretical analysis is rigorous, clearly distinguishing bias and variance contributions to the estimation error.

**Clarity:**

Notation is heavy but consistent; theoretical results are clearly labeled and contextualized.

Numerical results are clearly tabulated, showing systematic improvement with increasing sample size.

**Significance:**

Provides the first dimension-free rates for a tensor-train covariance estimator.

The balance between statistical guarantees and computational feasibility could influence future research on structured covariance learning.

**Weaknesses:**

All experiments are toy synthetic, with modest dimensions (≈ 10³ parameters). There is no demonstration on real, large-scale, or practical ML datasets.

The contribution is primarily theoretical. While interesting for statistical estimation, it lacks connection to modern ML problems (representation learning, deep generative models, etc.).

Unproven scalability:
Complexity analysis is asymptotic, there is no empirical scaling or memory footprint evaluation for large d. This makes it hard to evaluate its practicability for $d>10^4$

In the experiments, the algorithm assumes known TT-ranks, the proposed selection rule (Eq. 13) is untested experimentally.

The theoretical guarantees rely on sub-Gaussian assumptions and large spectral gaps. No robustness analysis for real conditions is provided.

**Questions:**

Can the approach handle non-Gaussian or heavy-tailed data in practice?

Have you attempted any real data (e.g., spatiotemporal or vision tensors)?

How sensitive is performance to the choice of ranks J,K?

How does HardTTh scale when the tensor shapes exceed a few hundred?

Could this framework integrate into neural-network covariance or feature-space modeling (feature covariance regularization in deep embeddings, low-rank approximations of neural Fisher information matrices, structured priors in Gaussian processes or diffusion models or covariance-aware adaptation in federated or meta-learning)?

---

> ### Author Response · Authors · 2025-11-21
>
> *1. All experiments are toy synthetic, with modest dimensions ($\approx 10^3$ parameters). There is no demonstration on real, large-scale, or practical ML datasets. Have you attempted any real data (e.g., spatiotemporal or vision tensors)?*
>
> We conducted experiments up to $4\cdot 10^9$ parameters, see Tables 7, 8 in the revision for more details. As once can see, HardTTh shows ability of scaling, whereas other methods do not. Additional experiments on image denoising were conducted, see page 10 in the revision. We highlight the fact that our work is primary theoretical, so experiments were suggested as an illustration for the theorems. Since HardTTh is adaptation of HOOI for the Tucker-2 structure, one can check relevant papers [Han, Willett, Zhang, 2022], [Liu, Shi, Liao, 2023], [Xu, 2015] for applications.
>
> [Han, Willett, Zhang, 2022] Han, Rungang, Rebecca Willett, and Anru R. Zhang. "An optimal statistical and computational framework for generalized tensor estimation." The Annals of Statistics, 50(1): 1-29, 2022.
>
> [Liu, Shi, Liao, 2023] Liu S, Shi X, Liao Q. Rank-Adaptive Tensor Completion Based on Tucker Decomposition. Entropy 25(2):225, 2023.
>
> [Xu, 2015] Xu, Yangyang. "On the convergence of higher-order orthogonality iteration." arXiv preprint arXiv:1504.00538 (2015).
>
> *2. The contribution is primarily theoretical. While interesting for statistical estimation, it lacks connection to modern ML problems (representation learning, deep generative models, etc.). Could this framework integrate into neural-network covariance or feature-space modeling (feature covariance regularization in deep embeddings, low-rank approximations of neural Fisher information matrices, structured priors in Gaussian processes or diffusion models or covariance-aware adaptation in federated or meta-learning)?*
>
> Similar approaches were applied to large-scale neural network training (see, for instance, [Eschenhagen, et al., 2023], [Chekalina, et al., 2025], [Zhou, et al., 2022]). For this reason, we do not see any obstacles for applying our method for this problem. However, this goes far beyond the scope of the present paper and is left for the future research.
>
> [Eschenhagen, et al., 2023] Eschenhagen, Runa, et al. "Kronecker-factored approximate curvature for modern neural network architectures." Advances in Neural Information Processing Systems 36 (2023): 33624-33655.
>
> [Chekalina, et al., 2025] Chekalina, Viktoriia, et al. "Generalized Fisher-Weighted SVD: Scalable Kronecker-Factored Fisher Approximation for Compressing Large Language Models." arXiv preprint arXiv:2505.17974 (2025).
>
> [Zhou, et al., 2022] Zhou, Yuchen, et al. "Optimal high-order tensor SVD via tensor-train orthogonal iteration." IEEE transactions on information theory 68.6 (2022): 3991-4019.

---

> ### Author Response · Authors · 2025-11-21
>
> *3. Unproven scalability: Complexity analysis is asymptotic, there is no empirical scaling or memory footprint evaluation for large $d$. This makes it hard to evaluate its practicability for $d > 10^4$. How does HardTTh scale when the tensor shapes exceed a few hundred?*
>
> According to the results reported in Tables 2 and 3, HardTTH is much faster than widely used HOOI and sufficiently better in terms of quality than PRLS. Additionally we provide experiments on a larger matrices, up to 700 million parameters and might conclude that our method scales well, even outperforming Tucker+HOOI baseline (further scaling is complicated due to enormous slowness of baselines, see Tables 7 for comparison).
>
> *4. In the experiments, the algorithm assumes known TT-ranks, the proposed selection rule (Eq. 13) is untested experimentally. How sensitive is performance to the choice of ranks J,K?*
>
> Additional experiments on sensitiveness of rank selection can be found in Figure 2.
>
> *5. The theoretical guarantees rely on sub-Gaussian assumptions and large spectral gaps. No robustness analysis for real conditions is provided. Can the approach handle non-Gaussian or heavy-tailed data in practice?*
>
> Yes, the approach can handle non-Gaussian data, because it provably works for the large class of sub-Gaussian distributions. Sub-Gaussian distributions are also quite common for real-world tasks.
>
> The main obstacle in covariance estimation with heavy-tailed observations is to keep logarithmic dependence on $(1/\delta)$. Usually, statistically optimal algorithms are not polynomial (and, thus, are not practical, see, for instance, [Mendelson, Zhivotovskiy, 2020], [Abdalla, Zhivotovskiy, 2025]). For this reason, extension to the case of heavy-tailed data is extremely challenging.
>
> HardTTh extensively relies on singular-value decompositions of unfoldings. The quality of SVD-type procedures decreases dramatically when the spectral gap condition is violated. For this reason, this condition is crucial for optimal rates of convergence.
>
> [Abdalla, Zhivotovskiy, 2025] P. Abdalla, N. Zhivotovskiy. Covariance Estimation: Optimal Dimension-free Guarantees for Adversarial Corruption and Heavy Tails, Journal of the European Mathematical Society, 2025.
>
> [Mendelson, Zhivotovskiy, 2020] S. Mendelson, N. Zhivotovskiy. Robust covariance estimation under $L_4$-$L_2$ moment equivalence. The Annals of Statistics, 2020.

---

### Official Review · Reviewer_43Kt · 2025-11-02

**Soundness:** 4
**Presentation:** 3
**Contribution:** 2
**Rating:** 4
**Confidence:** 4

**Summary:**

The paper studies covariance estimation from i.i.d. samples under a structured model for the covariance. Namely, the authors assume that the covariance is the sum of a small number of terms, with each of them given by the tensor product of three matrices.
Under suitable rearrangement of the entries, such a matrix can be written as a low rank tensor of order three. Hence the matrix estimation problem can be recast as a low-rank tensor estimation problem.
The author study an algorithm based on tensor unfolding and tensor power iteration, and derive a dimension-free bound on the estimation error of this algorithm under a condition on the singular values of the components of the covariance.

**Strengths:**

The paper is clear.
The results seem to be solid although the assumption on the effective dimensions is possibly suoptimal.

**Weaknesses:**

1. The model is a generalization of earlier models for matrices with tensor product structure. The new model is obviously more expressive than earlier ones,  but expressivity could have been achieved in other ways. No substantive motivation is provided for studying this specific model.
2. The mapping onto low-rank tensor denoising is obvious and once this is done, the application of tensor estimation method is straightforward (of course bounding the estimation error is non-trivial).

**Questions:**

Related to the above.
1) Is there a substantive motivation for studying this model?
2) Is there a difference in the estimation algorithm with respect to earlier work in tensor estimation?
3) It would be useful to describe the proof strategy in the main text.

---

> ### Author Response · Authors · 2025-11-21
>
> *1. The model is a generalization of earlier models for matrices with tensor product structure. The new model is obviously more expressive than earlier ones, but expressivity could have been achieved in other ways. No substantive motivation is provided for studying this specific model. Is there a substantive motivation for studying this model?*
>
> First, the considered model includes CANDECOMP/PARAFAC as a special case (see our discussion on page 2, the paragraph between eqs. (6) and (7)). While a sufficient number of papers studies its practical aspects and applications, only a few papers perform theoretical analysis of CANDECOMP/PARAFAC. Furthermore, we are not aware of any papers providing dimension-free bounds for this model when the canonical rank is greater than 1. Second, tensor-train decompositions look prospective in the context of LLM adaptation and compression. For this reason, theoretical analysis of such models is of considered interest.
>
> *2. The mapping onto low-rank tensor denoising is obvious and once this is done, the application of tensor estimation method is straightforward (of course bounding the estimation error is non-trivial).*
>
> We agree that construction of the rearrangement operator mapping a matrix with Tucker-2 structure into a low-dimensional tensor is obvious. However, it is not a contribution of the present paper. Our main result is a dimension-free non-asymptotic upper bound on estimation error.
>
> *3. Is there a difference in the estimation algorithm with respect to earlier work in tensor estimation?*
>
> We adapt the higher-order orthogonal iteration algorithm (HOOI) for our specific setup, where the covariance matrix has a Tucker-2 structure. According to the results reported in Tables 2 and 3, our method is much faster, than the standard HOOI.
>
> *4. It would be useful to describe the proof strategy in the main text.*
>
> We described the proof strategy in Appendix D. Unfortunately, we cannot move it to the main text because of page limits.

---

### Author Response · Authors · 2025-11-21

We sincerely thank the reviewers for their constructive and insightful feedback. We have addressed all comments point by point and incorporated the corresponding revisions into the manuscript.

---

### Author Response · Authors · 2025-11-28

Dear reviewers,

We wanted to kindly ask if there are any remaining questions or points that you would like us to clarify following our rebuttal.
We would be happy to provide further details if needed.

---

### Author Response · Authors · 2025-12-03

During the review process, we addressed several key areas regarding the motivation for our statistical model, the scope of our experiments, and the robustness of the proposed method. Below is a summary of the main concerns and how we resolved them in the revision.

**Theoretical Motivation and Model Choice**

Reviewers requested a clearer motivation for studying our specific tensor model. We clarified that our framework generalizes the widely used CANDECOMP/PARAFAC (CP) decomposition, treating it as a special case. This distinction is critical because, to date, dimension-free bounds in tensor estimation have been established primarily for rank-1 tensors. Our work contributes to high-dimensional statistics by deriving dimension-free non-asymptotic bounds for tensors of arbitrary rank, thereby offering theoretical guarantees that were previously unavailable for the CP model when the canonical rank exceeds one.

**Empirical Validation and Scalability**

To address the observation that our initial experiments relied heavily on synthetic data, we significantly expanded the empirical evaluation. We conducted real-world experiments involving image denoising and analyzed the effective dimensions of the Fisher matrix for a neural network trained on MNIST. Furthermore, we demonstrated the scalability of our approach, HardTTh, by conducting benchmarks on matrices containing up to 700 million parameters, showing that it outperforms standard baselines in computational efficiency.

**Robustness and Assumptions**

Regarding the method's sensitivity to theoretical assumptions, we emphasized that our guarantees hold for a broad class of sub-Gaussian distributions, ensuring applicability beyond strictly Gaussian settings. Additionally, we added new ablation studies (Figure 2) to demonstrate the stability of the method with respect to rank selection and address concerns about practical hyperparameter tuning.

**Clarity and Presentation**

Finally, we improved the readability of the manuscript by simplifying the statement of our main theorem (Theorem 2.2), making the bound more interpretable. We also addressed minor clarity issues and corrected typos throughout the text, as suggested by reviewers.

---

### Meta-Review · Area_Chair_PySn · 2026-01-06

**Summary:**

This paper addresses the important problem of covariance estimation from i.i.d. samples of high-dimensional random vectors and proposes a new estimation algorithm based on tensor-train (TT) decomposition.

The reviews are mixed, ranging from strong accept (8) to reject (2). Overall, I view the work as technically solid and the proposed approach as an interesting contribution. At the same time, several reviewers raised concerns about the assumptions and, more importantly, about how clearly the paper positions its relevance to machine learning. This lack of clarity likely contributes to the divergence in reviewer evaluations.

In the rebuttal, the authors made an effort to address these points. They added ablation studies to demonstrate robustness with respect to the assumptions, and they strengthened the empirical relevance by applying the method to estimating a Fisher information matrix for a neural network trained on MNIST. This is a promising direction, particularly given the connection to uncertainty quantification (UQ). The authors also demonstrated scalability, including experiments up to 700 million parameters, which further strengthens the practical appeal.

Despite these improvements, the paper remains borderline for acceptance, largely due to the remaining uncertainty about its fit and the mixed reviewer feedback. While I appreciated the revised manuscript and believe it improved the submission, I do not think it fully resolves the broader concerns about positioning and relevance. For this reason, I recommend rejection at this time, but I strongly encourage the authors to continue developing the work, either by targeting a more theoretical venue or by further strengthening the relevance for ML (e.g., demonstrating downstream value for UQ in large-scale neural networks).

**Reviewer Concerns:**

Overall, the authors effectively addressed a range of concerns, including the point that Algorithm 1 is not necessarily novel. The main contribution of the paper lies in its theoretical insights. In the rebuttal, the authors clarified the motivation and provided additional ablation studies and empirical results to strengthen the evaluation. However, it remains unclear whether the new experiments fully resolve some of the reviewers’ concerns. In addition, the manuscript would benefit from further revision to improve readability and better align the presentation with the expectations of this venue.

**Reviewer Scores:**

I do not expect the discussion phase would have moved the strongly positive or strongly negative scores by much.

---

### Decision · Program_Chairs · 2026-01-26

Reject